# 3D multiple point geostatistical simulation of joint subsurface redox and geological architectures

Rasmus Bødker Madsen[1], Hyojin Kim[1], Anders Juhl Kallesøe[1], Peter B.E. Sandersen[1], Troels Norvin Vilhelmsen[2], Thomas Mejer Hansen[2], Anders Vest Christiansen[2], Ingelise Møller[1], Birgitte Hansen[1]

[1]Geological Survey of Denmark and Greenland, Groundwater and Quaternary Geology Mapping, Aarhus, 8000, Denmark
[2]Aarhus University, Department of Geoscience, Aarhus, 8000, Denmark

*Correspondence to:* Rasmus Bødker Madsen (rbm@geus.dk)

**Abstract.** Nitrate contamination of subsurface aquifers is an ongoing environmental challenge due to nitrogen (N) leaching from intensive N fertilization and management on agricultural fields. The distribution and fate of nitrate in aquifers are primarily governed by geological, hydrological and geochemical conditions of the subsurface. Therefore, we propose a novel approach to model both geology and redox architectures simultaneously in high resolution 3D (25m x 25m x 2m) using multiple point geostatistical simulation (MPS). Data consists of 1) mainly resistivities of the subsurface mapped with towed transient electromagnetic measurements (tTEM), and 2) lithologies from borehole observations, 3) redox conditions from colors reported in borehole observations, and 4) chemistry analyses from water samples. Based on the collected data and supplementary surface geology maps and digital elevation models, the simulation domain was subdivided into geological elements with similar geological traits and depositional history. The conceptual understandings of the geological and redox architectures of the study system were introduced to the simulation as training images for each geological element. On the basis of these training images and conditioning data, independent realizations were jointly simulated of geology and redox inside each geological element and stitched together into a larger model. The joint simulation of geological and redox architectures, which is one of the strengths of the MPS simulations compared to other geostatistical methods, secures that the two architectures in general show coherent patterns. Despite the inherent subjectivity of interpretations of the training images and geological element boundaries, they enable an easy and intuitive incorporation of qualitative knowledge of geology and geochemistry in quantitative simulations of the subsurface architectures. Altogether, we conclude that our approach effectively simulates the consistent geological and redox architectures of the subsurface that can be used for hydrological modelling with nitrogen (N)-transport, which may lead to a better understanding of N-fate in the subsurface and to future more targeted regulation of agriculture.

## 1 Introduction

The loss of reactive nitrogen (N) from agricultural soils results in adverse environmental and human health impacts (Schullehner et al., 2018; Temkin et al., 2019), including eutrophication of freshwater and marine ecosystems and nitrate contamination of groundwater and drinking water (Schullehner and Hansen, 2014). In Denmark, since the 1980s N-regulations of intensive agriculture at national or regional scales have succeeded in lowering the N-impact on the aquatic environment (Dalgaard et al., 2014; Hansen et al., 2017). However, further actions are still required to improve the state of the aquatic ecosystems to meet the requirements of e.g. the EU Water Framework Directive (European Commission, 2018; Hansen et al., 2019; Kallis and Butler, 2001). Moreover, this must be achieved in a cost-effective manner for the society and the agricultural industry. This creates a demand for new knowledge and new solutions for more efficient future N-regulation of the agricultural sector both in Denmark and in other countries with intensive agriculture. The proposed direction is to introduce more targeted N-regulation depending on the site-specific conditions at field level. The targeted N-regulations require detailed knowledge about the subsurface hydrogeological and biogeochemical conditions because nitrate, which is the dominant form of N in aquatic environments, is transported predominantly with water flow and undergoes reduction in reducing zones of the subsurface. Thus, it has now become increasingly important to have detailed knowledge of the subsurface geology and redox architectures.

In a simple case with only vertical infiltration, nitrate concentrations in aquifers decrease with an increasing depth along three sequential redox zones (Kim et al., 2019; Wilson et al., 2018):

1) Oxic zone: Nitrate concentrations are equal to the leaching from the soil because of the oxic conditions preventing reduction

2) N-reducing zone: Nitrate decrease with increasing depth due to ongoing reduction of nitrate

3) Reduced zone: Nitrate free zone due to complete reduced redox conditions

The redox conditions of the subsurface has been widely investigated using various approaches focusing on different redox sensitive chemical compounds in groundwater such as nitrate, iron, sulphate, arsenic, uranium, and some organic contaminants. Modeling approaches have included: 1) process-based approaches (e.g. Abbaspour et al., 2007; Hansen et al., 2014a,2016a; Lee et al., 2008), 2) geostatistical methods (e.g., Kriging; Ernstsen et al. 2008; Goovaerts et al. 2005; Lin, 2008) and 3) machine learning (Close et al., 2016; Koch et al., 2019; Nolan et al., 2015; Ransom et al., 2017; Rosecrans et al., 2017; Tesoriero et al., 2015; Wilson et al., 2018). However, many of these approaches require large sets of data of especially groundwater chemistry, and it is costly and time consuming to collect sufficient volumes of data. Furthermore, ancillary data to spatially extrapolate the water chemistry, for instance soil types, topography, land use, surface slopes, only provide information about the near surface conditions (i.e., topsoil layer); therefore, predicting the redox conditions below the topsoil layer using these data may be inadequate. Particularly under geologically heterogeneous settings such as glacial terrain, the redox architecture can be complex (e.g. Hansen et al., 2021; Kim et al., 2019) with many shifts in redox state with depth at the same location. Upscaling of the point scale measurements of redox conditions into the 3D space would benefit from more detailed spatial information of the subsurface geological architecture.

In Denmark, the uppermost 100 to 200 meters of the subsurface generally consist of unconsolidated sediments reworked or deposited by glacial processes, making the subsurface architecture complex (Høyer et al., 2015; Jørgensen et al., 2015). Through the National Groundwater Mapping Program, Denmark is extensively covered with airborne electromagnetic measurements (AEM) (Møller et al., 2009; Thomsen et al., 2004) and together with borehole data, 3D geological mapping of Denmark has predominantly been carried out as cognitive modeling (see e.g. Høyer et al., 2015). In cognitive modeling, an experienced geologist combines all available subsurface data (e.g. boreholes, electromagnetic data, and seismic data) with preexisting geological background knowledge and performs interpretations through either manual (e.g. Jørgensen et al., 2013)

or semi-automatic approaches (e.g. Gulbrandsen et al., 2017; Jørgensen et al., 2015). Complex geological settings, however, pose a challenge for 3D modeling and interpretations between geological point data may lead to large uncertainties (Wellmann and Caumon, 2018).

The subsurface information itself contains uncertainties from sources that include measurement errors (Malinverno and Briggs, 2004), errors from using approximate physics (Hansen et al., 2014b; Madsen and Hansen, 2018), bias from interpolation methods (Wellmann and Caumon, 2018), and processing errors when handling geophysical data (Claerbout et al., 2004; Madsen et al., 2018; Viezzoli et al., 2013). Even geological knowledge cannot be considered uncertainty free (Bond, 2015; Lindsay et al., 2012; Sandersen, 2008; Wellmann et al., 2018; Wilson et al., 2019) and may rely on the training and experience

of the interpreter (Alcalde et al., 2017). These subjective biases are seen by some as one of the weak points of cognitive geological modeling (Bond, 2015; Wycisk et al., 2009), but is also argued to not imply a lack of scientific rigor (Curtis, 2012). It is difficult to fully incorporate the various uncertainties related to the subsurface information in cognitive modelling, and even more difficult to propagate these uncertainties through to subsequent analysis such as hydrological modelling.

In recent years, some studies have adopted geostatistical simulation methods for geological mapping of the substratum in order to quantify and possibly account for some of these uncertainties. A few examples exist of multiple-point geostatistical simulation (MPS) utilized for mapping 3D geology with AEM data (Barfod et al., 2018; He et al., 2014b; Høyer et al., 2017; Jørgensen et al., 2015; Vilhelmsen et al., 2019). However, AEM data provide structural information of the deeper subsurface (100-200 m) at a coarser resolution (Sørensen and Auken, 2004), and hence may not be adequate to provide structural

information for simulations of N-transport at catchment level occurring mainly within the upper 30 m. A newly developed towed transient electromagnetic method (tTEM) (Auken et al., 2019) provides data at much higher resolution but with a lower penetration depth than AEM. tTEM is, therefore, ideal for high-resolution mapping when focusing on the uppermost 50 to 70 m of the subsurface. None of the previous studies has investigated the geological and redox architecture simultaneously although these two are related and sometimes coevolved (Grenthe et al., 1992; Hansen et al., 2016a; Wilkin et al., 1996; Yan

et al., 2016).

The development of redox zones in the subsurface is dependent of several factors including 1) infiltration of atmospheric oxygen in geologic time; 2) anthropogenic leaching of nitrate; 3) the amount and reactivity of geogenic reducing minerals as pyrite or organic matter; and 4) the hydrogeological flow conditions. We propose a novel way to combine the available

information about hydrogeology and redox conditions (boreholes, electromagnetic data, geological maps and digital elevation maps) by estimating a quantified uncertainty at unsampled locations in modeling using geostatistical simulation. We specifically use MPS simulation to describe the spatial uncertainty in our models through a series of realizations of the subsurface that describe a quantified posterior distribution (Mariethoz and Caers, 2015). Using a bivariate training image (TI) of both geology and redox, we jointly simulate both redox and geology to ensure these will be consistent in the realizations.

TIs are created using expert knowledge combined with the available data to directly incorporate prior expert geological information. In addition to our proposed efforts of combining redox and geology modeling, we have also utilized data and geological knowledge to subdivide the simulation volume into smaller volumes based on different geological characteristics and the depositional environment. We refer to such smaller volumes as 'geological elements' (e.g. He et al., 2015; Høyer et al., 2015). Individual TIs are created with cognitive voxel modeling for each geological element such that they can be simulated

independently and subsequently stitched together. Geological interpretation of the depositional environment and the age of the sediments will help create an event chronology that supports the independence of the individual geological elements.

The aim of this paper is to demonstrate and review the proposed methodology of jointly simulating and determining the distribution of redox and geology using MPS. This is, to our knowledge, the first study of simultaneous modeling of geology and redox architectures in a geostatistical high-resolution 3D model. The novelty of this paper is hence the presentation of the complete practical framework and steps needed to apply MPS for redox and geology modeling. These steps include quantifying spatial variability in TIs, quantifying conditional information and accounting for major geological depositional events via geological elements. This may be fundamental to better understanding N-retention within the subsurface and important for future more targeted N-regulation and management of agriculture for protection of vulnerable surface waters and groundwater. Thus, providing stakeholders with a powerful tool based on integrated expert knowledge and quantified estimates of structural uncertainty through probabilistic predictions of the complex interplay between redox and geological architecture.

## 2 The study area

The study area is a small Danish agricultural first-order hydrological catchment to Horndrup Bæk called LOOP3 with an area
of approximately 550 ha. The area is located at the Jutland peninsula in Denmark, with a coastal temperate climate (Figure 1).
The dominant soil types are classified as sand-mixed clay (70%) and clay sand (24%). Forest accounts for 18% of the catchment
area, the rest being used for agricultural purposes except for a limited area taken up by buildings and roads. The catchment has
been part of the Danish National Environmental Monitoring Program since 1989 aiming at evaluating the effect of the Danish
N-regulation of agriculture on the aquatic environment (Hansen et al., 2019). During the last almost 30 years, the N-
concentrations in soil water, drainage, shallow groundwater and streams have been measured regularly at several stations in
the agricultural fields (Blicher-Mathiesen et al., 2019). Therefore, the site is ideal for testing new subsurface mapping
techniques of geological and redox architectures.

The study area is located in a hilly glaciated landscape in the eastern part of Jutland just east of the highest point in Denmark
(Figure 1). The highest elevations reach 170 meters above sea level (m a.s.l.) in the southwestern part and slopes down to
around 40 m a.s.l. in the northeast (Figure 1a). To the north of the study area, a system of open tunnel valleys forms a low-
lying area with several lakes. The catchment is dominated by glacial till deposits from the latest glaciation and the orientation
of the hills generally show former ice push directions from the northeast. In the lowest parts of the terrain, occurrences of
meltwater sand are also found. Occurrences of postglacial freshwater deposits can be found locally in smaller topographical
lows (Jakobsen and Tougaard, 2020). Several buried valleys have been mapped outside the study area (Sandersen and
Jørgensen (2016); www.buriedvalleys.dk (2020); Figure 1). The buried valleys were formed as elongated tunnel valleys
underneath the ice sheets, they are generally between 1 and 2 km wide and some of them have depths of more than 100 m
(Jørgensen and Sandersen, 2006; Sandersen and Jørgensen, 2017). These valleys are generally filled with younger Quaternary
sediments. In this region, the valleys mostly have two preferred orientations, one around WNW-ESE/NW-SE and the other
around SW-NE/WSW-ENE (Figure 1), with the first mentioned clearly visible in the present-day topography.

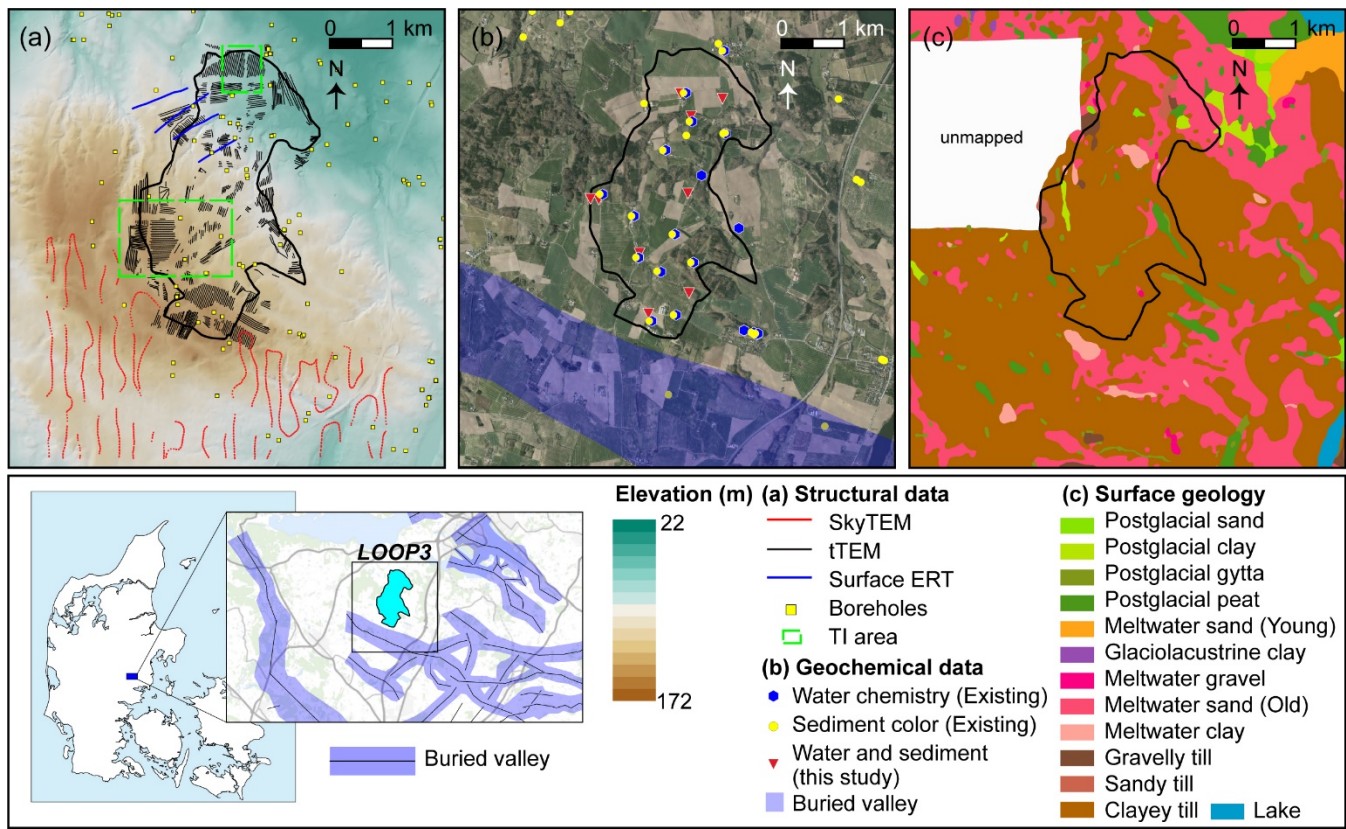

Figure 1: The study area and available data, where a) display digital terrain model, geophysical data and outlined TI areas, b) an orthophoto (from Geodatastyrelsen, ortophoto forår, WMS Service, March 2021) and geochemical data, and c) a surface geology map (1 m below surface (Jakobsen and Tougaard, 2020)). Insets with map of Denmark and regional view of the study site with mapped buried valleys (www.buriedvalleys.dk (2020)).

## 3 Materials

Some data are specifically gathered for this study (tTEM and new boreholes, see Figure 1) while other existing data are freely available through the Danish borehole database "Jupiter" (Hansen and Pjetursson, 2011) and the Danish geophysical database for onshore data "GERDA" (Møller et al., 2009). All available data are shown in Figure 1 along with the terrain and outline of the study area.

### 3.1 Geological and topographical data

The digital elevation map presented in Figure 1a is available from Styrelsen for Dataforsyning og Effektivisering (2016). The elevation map is resampled on a 25m x 25m grid such that adjustment with interpreted surfaces is seamless. The geological surface map (Figure 1c) of the surficial cover of Denmark is compiled from small pristine sediment samples collected at c. 1 m depth using a so-called spear-auger. The mapping geologists interpret the origin and type of the sediment in the field and classify a sediment-type following the current terminology described by Jakobsen and Tougaard (2020). Samples are taken with a distance of 100-200 m to map the transitions between the different sediment types. Afterwards the surface geology-symbols are transferred to a master map, contoured and color-coded resulting in a geological map sheet on a scale of 1:25.000 with a resolution of ± 100 m (Figure 1c, https://eng.geus.dk/products-services-facilities/data-and-maps/maps-of-denmark/ (2020))

Borehole lithological information (Figure 1a) is gathered from the Jupiter database to which lithological sample descriptions have been reported since 1912. The borehole lithological samples are described and interpreted by geologists following standards outlined by Gravesen and Fredericia (1984), including interpretation of depositional environment and chronostratigraphy and thereby resulting in sediment types similar to those used in the geological mapping.

In our study site, a total of 18 specific sediment types are found in borehole descriptions and on the geological surface map combined. To lower the number of variables in the geostatistical modelling and potentially later on in hydrological simulations, the sediment types are grouped into five categories focusing on their hydrological properties and depositional environment (Table 1). For instance, the two till groups have vastly different hydrological properties because of the overall grain size difference between clay tills and sand/gravelly tills. The partly organic postglacial sediments may show variable hydrological properties. However, they are hugely important in terms of redox potential because of organic content; therefore, they are categorized in one group.

Table 1: Lithology groups in the study area used in the geostatistical simulation. The sediment type abbreviations in the right column represent the Danish sediment characterization standards.

| Lithology groups – study area | | |
|---|---|---|
| *no.* | *Group name* | *Sediment type* |
| 1 | Clay till | ML, (L) |
| 2 | Meltwater sand/gravel | DS, DS-DG, DG, G, S, TS, (O) |
| 3 | Meltwater clay/silt | DL, DI, DV, (FL) |
| 4 | Sandy till | MS, MG |

| 5 | Postglacial (partly org.) | FP, FT, FS |

## 3.2 Geophysical data

The tTEM (ground-based) and SkyTEM (airborne) are transient electromagnetic systems used for mapping subsurface resistivity variations (Auken et al. 2019, Auken and Sørensen 2004). The SkyTEM system carries the instrument, transmitter loop and receiver coil in a sling load under a helicopter and is designed to map resistivity to several hundred meters depth. The tTEM system applies the transient electromagnetic method in an offset-loop configuration which for the present study is configured using a 2 m by 4 m transmitter loop and a receiver coil in a distance of 9 m, towed by an all-terrain vehicle (Auken

et al., 2019). The tTEM system is designed to resolve resistivity from 2-3 m depth to c. 70 m depth. Processing and inversion of tTEM data follow in general the scheme for SkyTEM, described by Auken et al. (2009). The inversion of the data is based on local 1D forward responses and spatial constraints between the model parameter forming a pseudo 3D model space (Auken et al., 2015; Viezzoli et al., 2008).

The tTEM dataset has been collected in 2018. Although the coverage is rather patchy (< 50 % of the model area in Figure 1a), it provides valuable information on the geological setting. The final tTEM information used in the geostatistical modelling is the pseudo 3D model space moved to the closest grid node. Together with borehole lithological logs, tTEM represents the basis for modelling the geology. A few deep boreholes are used for the correlation between resistivities and lithologies.

Although located outside the study area, the SkyTEM-data (Figure 1a) adds valuable information on the geological connections to neighboring areas. A small survey of surface electrical resistivity tomography (ERT) (e.g. Loke et al. (2013)) gathered from the GERDA database supplements the tTEM survey in the northern part of the study area.

## 3.3 Geochemical data

Redox conditions can be defined both by sediment colors, concentrations of redox sensitive elements such as dissolved oxygen, nitrate, iron, and sulfate in water (Ernstsen and von Platen, 2014; Hansen et al., 2016a, 2021; Kim et al., 2019) and the sediment fraction of ferrous iron ($Fe_{FA}^{II}$) of the formic acid extractable Fe ($Fe_{FA}^{II}+ Fe_{FA}^{III}$) (Hansen et al., 2021). In this study, the sediment color was the primary indicator to define redox conditions, and the water chemistry was used to supplement the sediment color interpretations. The sediment colors may be the resultant of the cumulative effects of the redox structure evolution since the

deglaciation while the water chemistry may display a snapshot of the short-term redox chemistry, which may be temporally variable. Therefore, we postulate that the redox conditions interpreted from the sediment colors may be more coherent with the geological structure than that of the water chemistry. In addition, the sediment colors provide 1D profile information of the redox conditions and more data points are available compared to water chemistry which provide point scale information. The sediment color and water chemistry data were extracted from the Jupiter database and the 9 new boreholes that were

drilled in this study (Figure 1b).

Based on the sediment colors, oxic conditions are defined by red, orange, yellow and combinations of these colors. Gray, olive and blue colors represent reduced conditions. Mixed colors between oxic and reduced colors (e.g., yellowish gray) are defined as N-reducing conditions. Within the catchment boundary, the sediment color data were available at 14 boreholes in the Jupiter

database and for the 9 new boreholes. Based on water chemistry, oxic is defined by dissolved oxygen greater than 1mg/L, N-reducing is dissolved oxygen less than 1mg/L and nitrate greater than 1mg/L, and reduced is both dissolved oxygen and nitrate below 1 mg/L and iron greater than 0.2 mg/L (Hansen et al., 2021). Based on sediment chemistry, the fraction of $Fe_{FA}^{II}$ over $Fe_{FA}^{II}+ Fe_{FA}^{III}$ is close to 0 in oxic conditions and close to 1 fraction in reduced conditions (Hansen et al., 2021). The values in

between are interpreted as N-reducing conditions (Hansen et al., 2021). The water and sediment chemistry data were available

at 22 and 9 locations, respectively (13 in the Jupiter database and 9 in this study, see Figure 1b).

## 4 Methods

### 4.1 MPS modelling

In this paper, we adopt a MPS simulation approach for quantifying the spatial uncertainty of the subsurface. Geostatistical
simulation generally provides a way of quantifying the spatial uncertainty through different possible realizations of the
subsurface architecture. These realizations are generated using stochastic modeling that accounts for the spatial dependency
between the model parameters. We choose MPS simulation over e.g. a two-point geostatistical approach because it is generally
more capable of producing realizations with geological realism in terms of correlation and coherency of geological features
(Journel and Zhang, 2006; Madsen et al., 2021; Mariethoz and Caers, 2015). Effectively reproducing coherent layers is key
for a successful subsequent hydrological modeling. The expected subsurface variability is portrayed in one or more training
images (TIs). MPS simulation is then able to utilize these TIs to generate different realizations of the portrayed subsurface
through a stochastic sampling process. In total, these realizations, stemming from the MPS algorithm plus TI, together
represent the quantified prior information of the system. In our case, the intuitive aspect of a TI, as opposed to a mathematical
prior, is helpful for collaboration between mapping experts and geostatisticians.

Many MPS algorithms exist today (Gravey and Mariethoz, 2019; Guardiano and Srivastava, 1993; Hansen et al., 2016b;
Hoffimann et al., 2017; Mariethoz et al., 2010; Straubhaar et al., 2011; Strebelle, 2002; Tahmasebi et al., 2012). In the current
study we use direct sampling (Mariethoz et al., 2010) as implemented in the software package DeeSse (Straubhaar, 2019). The
main reason is its ability to utilize a bivariate training image that allows for joint simulation of geology and redox.

Simulations can be forced to match observational data creating conditional realizations (Chilès and Delfiner, 2012; Journel
and Huijbregts, 1978). Additional data not portrayed in the TI enters the simulation setup as either hard or soft data. Hard data
corresponds to information not allowed to change between different realizations and is placed directly in the simulation grid.
Information from some boreholes can often be considered as hard data because it is fixed in space and can have a relatively
high resolution and accuracy. Hard data, in most cases, offers the first conditioning nodes and patterns to be matched during
simulation, depending of course on the number of conditional points used. Consequently, hard data usually plays a significant
role in lowering the entropy of the final simulations. If data is not reliable enough (too uncertain) to be deemed hard data, they
can instead be treated as uncertain information (soft data), quantified through probability distributions. In DeeSse, soft data
probabilities are handled by introducing a penalty proportional to the soft data probabilities, such that it becomes difficult to
find a match for a given lithology group or redox condition if the probability is low and conversely easier if the probability is
high. (Mariethoz et al., 2015).

### 4.2 Ensemble statistics

We introduce the mode and entropy as summary statistics for the ensemble of possible models of the subsurface. For a discrete
probability distribution, the mode represents the most probable category in each voxel. The entropy, H, of a discrete probability
distribution with $K$ outcomes is explicitly calculated as (Shannon, 1948):

$$\mathrm{H} = -\sum_{k=1}^{K} \log_K\big(p(k)\big)\, p(k), \tag{1}$$

where $p(k)$ is the probability of the $k$th outcome. In our case, the entropy is calculated in each voxel where $p(k)$ is the number
of times a certain category appears in the realizations divided by the number of realizations. The entropy reveals insights to
the variability and hence the certainty of a specific outcome of each voxel. For H=0 we have full certainty (maximum

information content) of the voxel category and conversely for H=1 (Hansen, 2021). The mode and entropy are hence comparable to the mean and variance in Gaussian statistics.

**5 Modelling setup**

In the following, we present the methodology progressing through the modeling workflow of the study area. The workflow consists of three phases: 1) Preparing input data, 2) data analysis and setup including delineation of geological elements, construction of training images, preparing hard and soft data as well as setting up the simulation grid, and 3) running the MPS simulation algorithm. A schematic overview of the workflow is seen in Figure 2. The following sections primarily describe

phase 2.

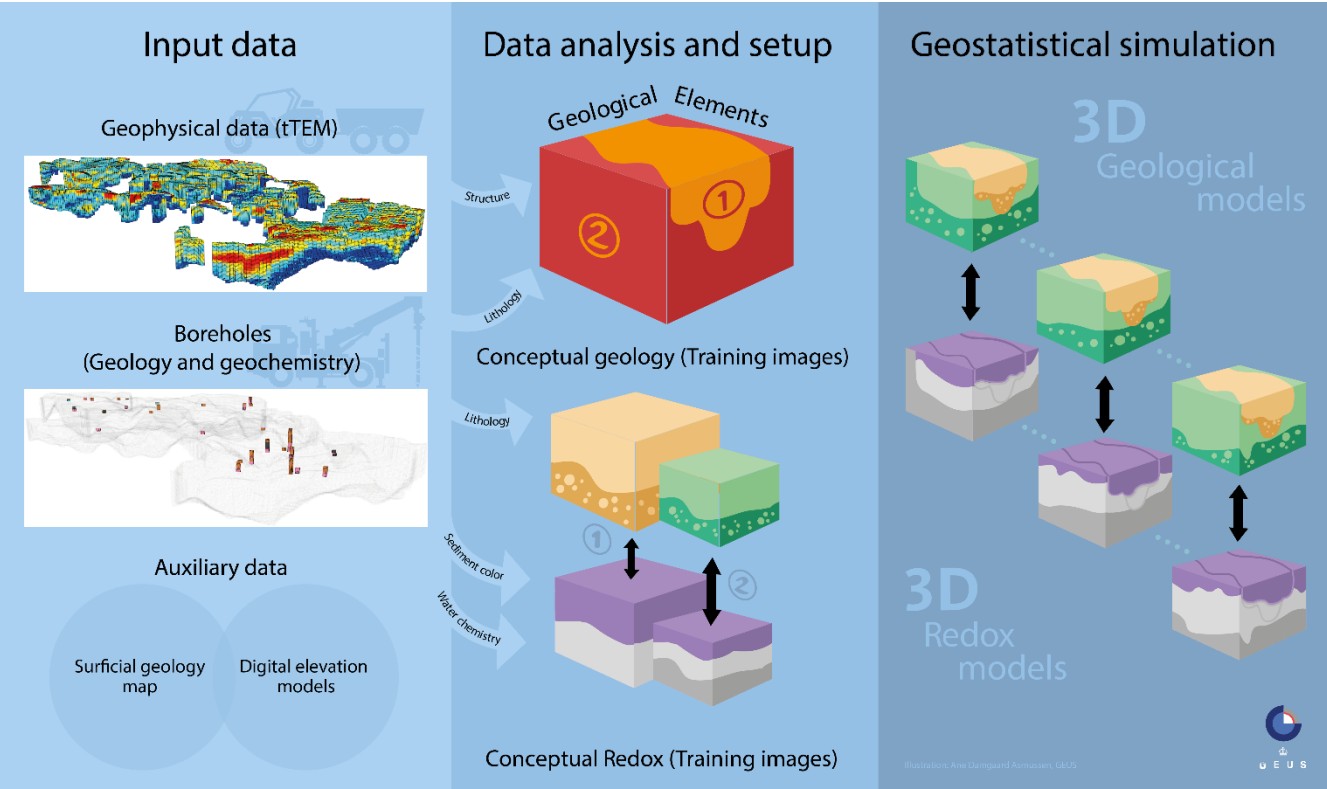

**Figure 2: Schematic overview of the proposed workflow from input data (left) through data analysis and simulation setup (middle) and geostatistical simulation (right).**

**5.1 Simulation grid**

The simulation grid is discretized with a voxel resolution of 25m x 25m x 2m. The digital elevation model constitutes the top of the simulation grid, whereas both the bottom boundary and the internal subdivision into subvolumes are delineated by the geological elements (see below for details). The resulting simulation grid is shown in Figure 3b and the total number of voxels in the simulation grid is listed in Table 2.


**Table 2: Summary of number of voxels for simulation grid and TIs. The relative sizes of the TIs are calculated as the ratio between the number of voxels in the TI and the number of grid voxels.**

|  | Number of voxels | Number of voxels in TI | Relative size of TI |
|---|---|---|---|
| Quaternary sequence (Element 1) | 143698 voxels | 54258 voxels | 37.76% |
| Buried valley (Element 2) | 57015 voxels | 12449 voxels | 21.83% |
| Total | 200713 voxels | 66707 voxels | 33.24% |

## 5.2 Geological elements

The modeling domain is split into geological elements in order to subdivide the subsurface into separated volumes based on sediment heterogeneity and geological event chronology. In this way, smaller volumes with different lithology and structure can be treated separately in the geostatistical simulations. The geologist interprets and delineates the geological elements of the subsurface using the geological, geophysical and topographical input data. Three distinct geological elements are identified in the study area, see Figure 3a; (1) An upper Quaternary succession of sediments with an erosional boundary to the pre-

Quaternary sediments below, (2) A large, deeply eroded, buried tunnel valley and (3) Pre-Quaternary Paleogene clays defining the bottom of the groundwater system. The simulation grid is chosen to include only the Geological Elements 1 and 2 (see Figure 3b). The third geological element, the Paleogene clay, constitute a thick non-penetrable layer, and as its top defines the lower hydrological boundary of the area, geostatistical simulation has not been performed on this geological element. We find it reasonable to do so because the Paleogene clays are homogeneous and very thick. This type of clay is generally found as a

good electrical conductor in Denmark, and because the TEM method is sensitive to good conductors, the depth to the top of the layer can be determined with low uncertainty (e.g. Danielsen et al. (2003)). Delineation of the Paleogene clay surface from the tTEM data is therefore straightforward as long as it can be found within the depth of investigation of the tTEM method (Vest Christiansen and Auken, 2012). Furthermore, in the study of Barfod et al. (2018), Paleogene clays were given a discrete value in the MPS simulation but showed only little variability in the spatial extent.


We assume independence between the two uppermost geological elements because they appear to represent different geological events. The buried valley to the north is apparently incised into both the Quaternary sequence and the pre-Quaternary clay below, and the infill is clearly different compared to the Quaternary sediments to the south. The buried valley (Geological Element 2) has a more complex infill with individual layers of limited extent compared to the Quaternary layers

of Geological Element 1, which show less complexity and more pronounced stratification more or less undeformed by the glaciations. The geological events that formed each element are therefore considered different although they contain the same lithology groups, and this justifies the assumption of independence from a geological point of view. The buried valley to the north takes up roughly a quarter of all voxels whereas the Quaternary sequence occupies the main part of the simulation grid (Table 2).

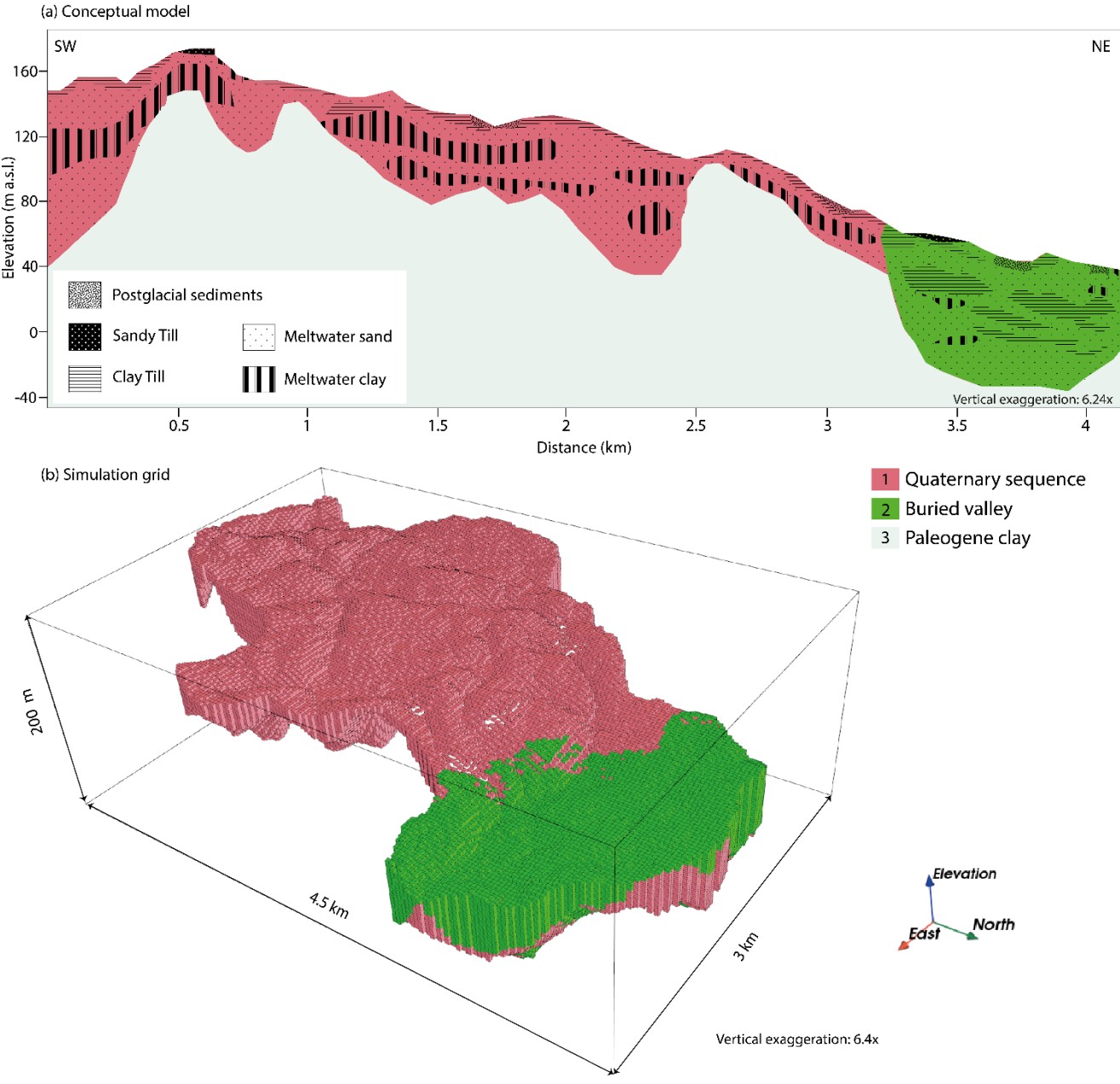

**Figure 3: a) Conceptual drawing of SW/NE profile through the study area. b) Simulation grid showing the two main geological elements used for the geostatistical simulation.**

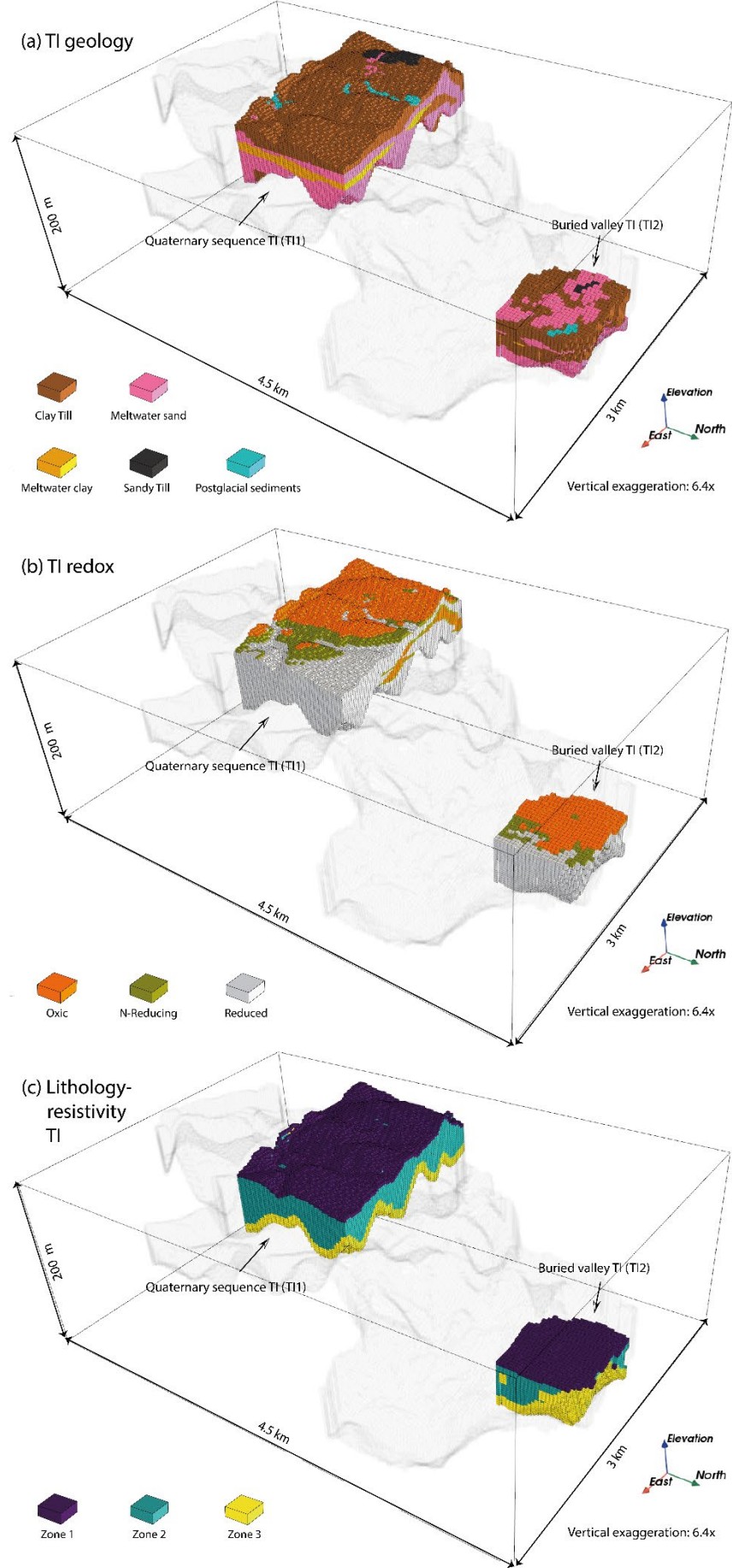

**Figure 4: a) Geology TIs overprinted on the simulation grid. b) Redox TIs overprinted on the simulation grid. c) Zones in TIs used for resistivity-lithology relationship inference**

## 5.3 Training images

The TIs providing information about the geology and redox conditions within the geological elements are designed in a sequential workflow (Figure 4a,b). At first, two geology TIs are generated, one within each of the two geological elements. The first step is to appoint a smaller part of the simulation area for detailed geological characterization and interpretation using a voxel modelling approach, see Figure 4a. The lithological population of the voxels is based on the conceptual understanding of the geological event chronology, glacial processes forming the area, and an interpretation combining borehole information, digital elevation maps, surface geology maps and the spatially distributed geophysics using regional geological understanding. The criteria for TI area selection in this specific study were dense data coverage of geophysics and especially the availability of boreholes that penetrate the entire modeling domain with good quality lithological descriptions. For despite having a better geophysical data coverage in the southernmost part of study area according to Figure 1a, the TI in the Quaternary element is chosen based on sufficient geophysical data coverage and having the two main boreholes within its borders. The TI section needs also to represent the expected variability in geology, both in vertical and horizontal extent, which is another selection criterion. In reality, it is not possible to capture the total variability and heterogeneity in the TI, due to its finite size, but the important features must be represented. In TI1, smooth glaciotectonic deformation of the Quaternary units due to ice push from the northeast, is modelled. Likewise, smaller incised buried valleys in the Eocene clay with mostly sandy infill is included based on the tTEM spatial data coverage, see Figure 3a. TI2 represents the sedimentary infill in a large buried valley (geological element 2), where also more regional information from nearby buried valleys of the same generation was taken into account (see Figure 1b). This information combined with the tTEM data coverage, two boreholes within the valley north of the study area, and the surface geology maps, has been the basis for the voxel modelling of TI2. The complexity in the infill of the buried valley is represented by individual layers of limited extent as seen in Figure 3a.

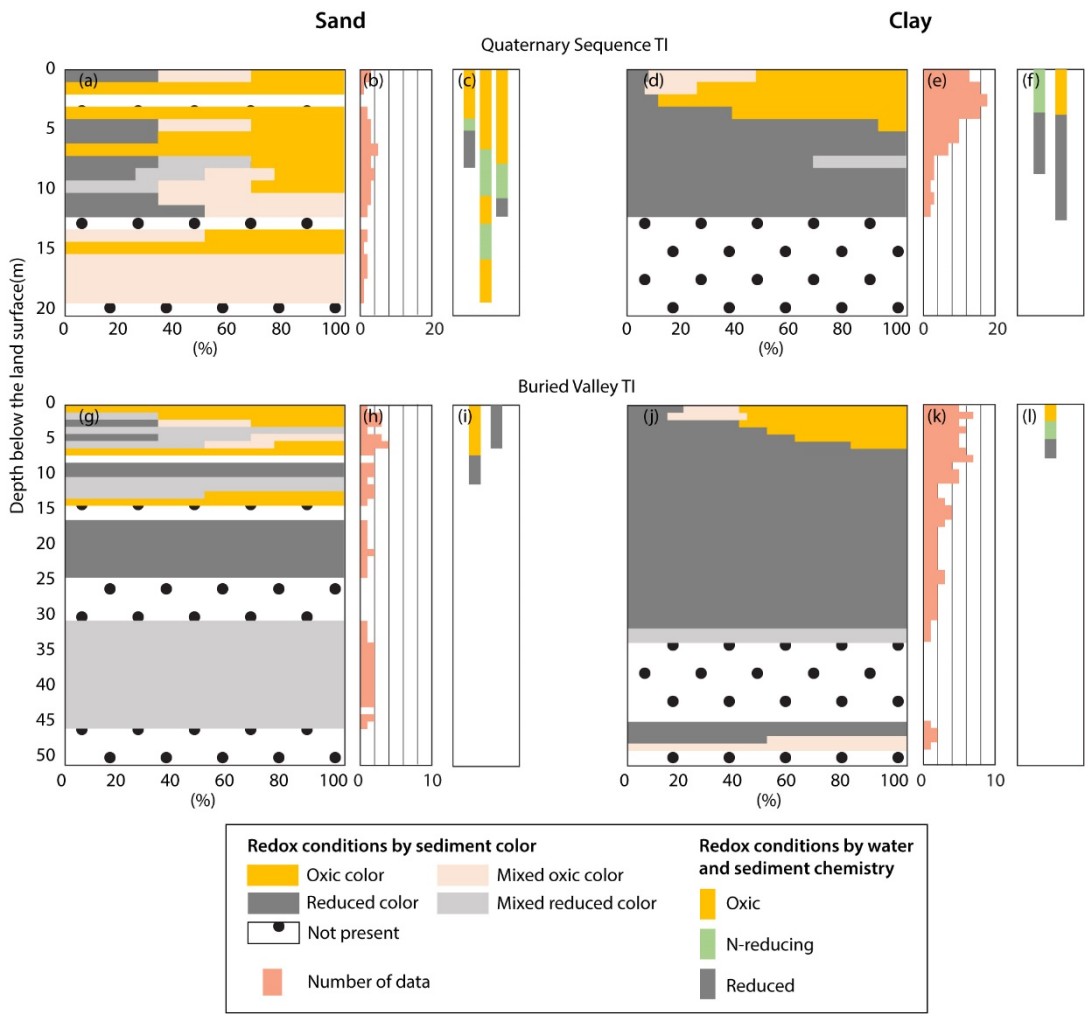

**Figure 5:** Profiles of redox conditions for sand and clay for each TI area based on geochemical observations. The redox interpretation based on the sediment colors are done separately for sand (a) and clay (d) of the Quaternary sequence TI and of the Buried valley TI (g and j, respectively), and the number of boreholes used in the interpretations are shown in b, e, h and k, respectively. The number of boreholes of each redox color category (oxic, mixed oxic, mixed reduced and reduced colors) are normalized and shown in %. The redox interpretations based on water and sediment chemistry of sand and clay of the Quaternary sequence and buried valley TIs are shown in c, f, i, and l, respectively.

350

The geological training images are then translated into redox TIs by upscaling the redox interpretations of the sediment colors and water chemistry as described in section 3.3 (Figure 4b and Figure 5). The sediment color data were first discretized into a 1m-interval and then the redox condition for each interval was assigned according to the sediment color. The interpreted data were summed up separately for sand and clay for each TI area to produce depth profiles of redox conditions (Figure 5a, 5d, 5g and 5j). For the Quaternary sequence and buried valley TIs, 13 and 7 boreholes are available with sediment color descriptions (Figure 5b, 5e, 5h and 5k), respectively, and 5 and 3 boreholes are available with water and sediment chemistry (Figure 5c, 5f, 5i and 5l), respectively.

The redox interpretation revealed that in the Quaternary sequence, oxic conditions are much deeper in sand (at least 20 meters; Figure 5a and 5c) than in clay (4-6 meters; Figure 5d and 5f). We postulated that the Quaternary sequence is the geological window type of redox architecture proposed by Kim et al. (2019): the sandy units exposed to the surface act as 'geological windows', which allow transporting oxidants (i.e., oxygen and nitrate) via gas and water into the deeper subsurface, resulting in development of a deep oxic zone below a reduced clay layer. In the Quaternary sequence area, all the boreholes for the water and sediment chemistry were collected in these geological windows, which are predominantly in oxic conditions, confirming our interpretations. In the buried valley, the oxic layer was relatively shallow compared to that of the Quaternary sequence. This shallower oxic layer may be attributed to a shallower and temporally invariant groundwater table in this area compared with the Quaternary sequence. A secondary oxic layer below the first oxic layer is not expected, due to the clay-dominant conditions of the surface geology (mainly clay-till; Figure 1c) and subsurface structure. We concluded that in the buried valley, oxidants are delivered either vertically via water infiltration or gas diffusion or the top oxic layer (4-6 meters below the land surface) from the Quaternary sequence, resulting in the planar type of redox architecture (Kim et al. 2019).

Based on these interpretations, we assigned each lithology group with a probability of belonging to each of the three redox conditions at the surface (Table 3). 80% of the meltwater sand in the geological windows (sand units connected to the surface) of TI1 were assigned to be oxic down to 20 meters, and the rest was equally distributed between N-reducing and reduced conditions, respectively, to allow variability in simulations. These N-reducing and reduced conditions were mainly located in lower elevations because of the higher possibility of water saturated conditions. For the connected sand, with increasing depth, the fraction of oxic voxels was assumed to be reduced by 10% compared to that of the overlying layer for the 20-30-meter interval and by 20% for depth below 30 meters. The sand voxels that are not connected to the land surface was assumed to be reduced. The N-reducing conditions is always located at the boundary of oxic conditions in the profiles the fraction was limited to 10 % of the total sandy voxels of each layer in the TIs. The rest was assigned to reduced conditions. For clay till and meltwater clay of TI1, 60%, 20%, and 20% of the first layer voxels (Table 3) were attributed to oxic, N-reducing, and reduced conditions in the order of elevation (lower elevation = reduced condition) due to proximity to streams. With increasing depth, the fractions of N-reducing and reduced conditions were assumed to be increased by 10% and 20%, respectively up to 6 meters below the land surface. Below 6 meters, clay was always reduced.

385

For meltwater sand and sandy till of TI2, 80%, 10%, and 10% of the top layer (Table 3) were attributed to oxic, N-reducing, and reduced conditions in the order of elevation. Below the first layer, the fractions of the oxic and N-reducing voxels then were assumed to be decreased by 70% and 50%, respectively, and the rest was assigned to reduced conditions. Clay till and

meltwater clay of the buried valley TI, 60%, 20% and 20% of the top layer (Table 3) were attributed to oxic, N-reducing, and reduced conditions in the order of elevation. The fractions of oxic and N-reducing voxels were assumed to be lowered by 60% and 20%, respectively, compared to those of the overlying layer with increasing depth. The rest was assigned reduced conditions.

Postglacial sediments, which are freshwater deposits often rich in organic material (Jakobsen and Tougaard, 2020), were assigned almost exclusively with reducing conditions (90% probability). Like the sandy and clayey sediments, we distributed the remaining 10% probability equally among the other two redox conditions to allow for some variability.

**Table 3: The probabilities for redox conditions (oxic, N-reducing and reduced) based on geochemical observations in Figure 5 for the top (surface) layer of the training images. The probabilities for each lithology group sums to one.**

| Lithology group | Oxic | N-Reducing | Reduced |
|---|---|---|---|
| Clay till | 0.6 | 0.2 | 0.2 |
| Meltwater sand/gravel | 0.8 | 0.1 | 0.1 |
| Meltwater clay/silt | 0.6 | 0.2 | 0.2 |
| Sandy till | 0.8 | 0.1 | 0.1 |
| Postglacial | 0.05 | 0.05 | 0.9 |

The described sequential workflow of geology and redox TI construction ensures consistency between the two training images in the joint simulation of the two variables. Approximately one-fourth of 1 reaches outside of the study area whereas the whole of TI2 is located within. We intentionally do this to ease the construction of TI1 as the surrounding area to the west shows similar geological variability to the Quaternary sequence and therefore provide helpful information during the creation of TI1. Additionally, this information is independent and allows more possible matching configurations in the TI during simulation. TI1 is about one third of the size of Quaternary sequence element and that of TI2 is one fifth of the buried valley element (Table 2). The TIs used in this study have different statistical properties depending on the location, i.e. they are non-stationary. For instance, visually it is easy to confirm that the probability of finding an oxic redox condition in the lower part of the TI is much different than in the top. A non-stationary TI is not unacceptable but can have some unwanted effects when combined with MPS algorithms expecting a stationary TI and will be discussed later.

**5.4 Conditioning data**

**5.4.1 Hard data**

The geological surface map and the borehole data (both lithology and redox) were treated as hard data in the simulation grid and are shown in Figure 6.

The sediment types that were grouped into lithologies (Table 3) were placed at the top voxel in the simulation grid, corresponding to the surface. We do not explicitly use the entire geological map as hard data. The borders between the lithology polygons of the surface geology map were originally delineated based on sediment samples, geomorphology, and topography(Jakobsen and Tougaard, 2020). In general, it means that the closer you are to the center of a polygon, the more certain you are of the correct lithology. Conversely, the boundaries between polygons represent the least certain parts of the map. A buffer zone is therefore adapted between the polygons to express the uncertainty of the geological surface map. The

buffer zone is simply created by checking all neighboring voxels for each voxel in the surface map. If the current voxel shares a value with all surrounding voxels it is likely situated safely within a polygon and is kept as hard data. Conversely, if one of the neighboring voxels provides a mismatch, the current voxel is likely close to a polygon boundary and is not included as hard data. Alternatively, a negative buffer around each polygon could be adopted.

The redox conditions are grouped into the three main redox categories; oxic, N-reducing, and reduced. The wells indicate that the area is dominated by reduced conditions. Oxic conditions are mainly present in the upper meters of the simulation domain, and only one well displays the reverse trend with an oxic part below reduced conditions due to heterogenous geology.

### 5.4.2 Soft data

We use the geological surface map (Figure 1c) as a soft data indicator of lithology in the buffer zone. Geological complexity is one of the main drivers of uncertainty in geological mapping along with the amount, quality and spatial distribution of data (Keefer, 2007). Accessibility is an important factor to consider in terms of both amount and spatial distribution of data (Keaton and Degraff, 1996). In Denmark, however, neither terrain nor private property poses a major issue when mapping surface geology. On the level of investigation, the geology in the study area is relatively simple, alleviating some of the uncertainty due to complexity. The main source of uncertainty in the surface geology maps comes from interpretations of sediment types from the small samples and the final shape and size of polygons. We generally consider the surface geology as very certain data and thus provide all values 0.7 probability of being true. The last 0.3 probability is split equally between the four other lithologies, which reflect the uncertainty level of misinterpretations. Regardless, because much of the geological surface map is used directly as hard data, the quantified uncertainty only affects the buffer zone as outlined earlier. For the redox domain, we translate the geological surface map to soft redox data using the probabilities provided in Table 3.

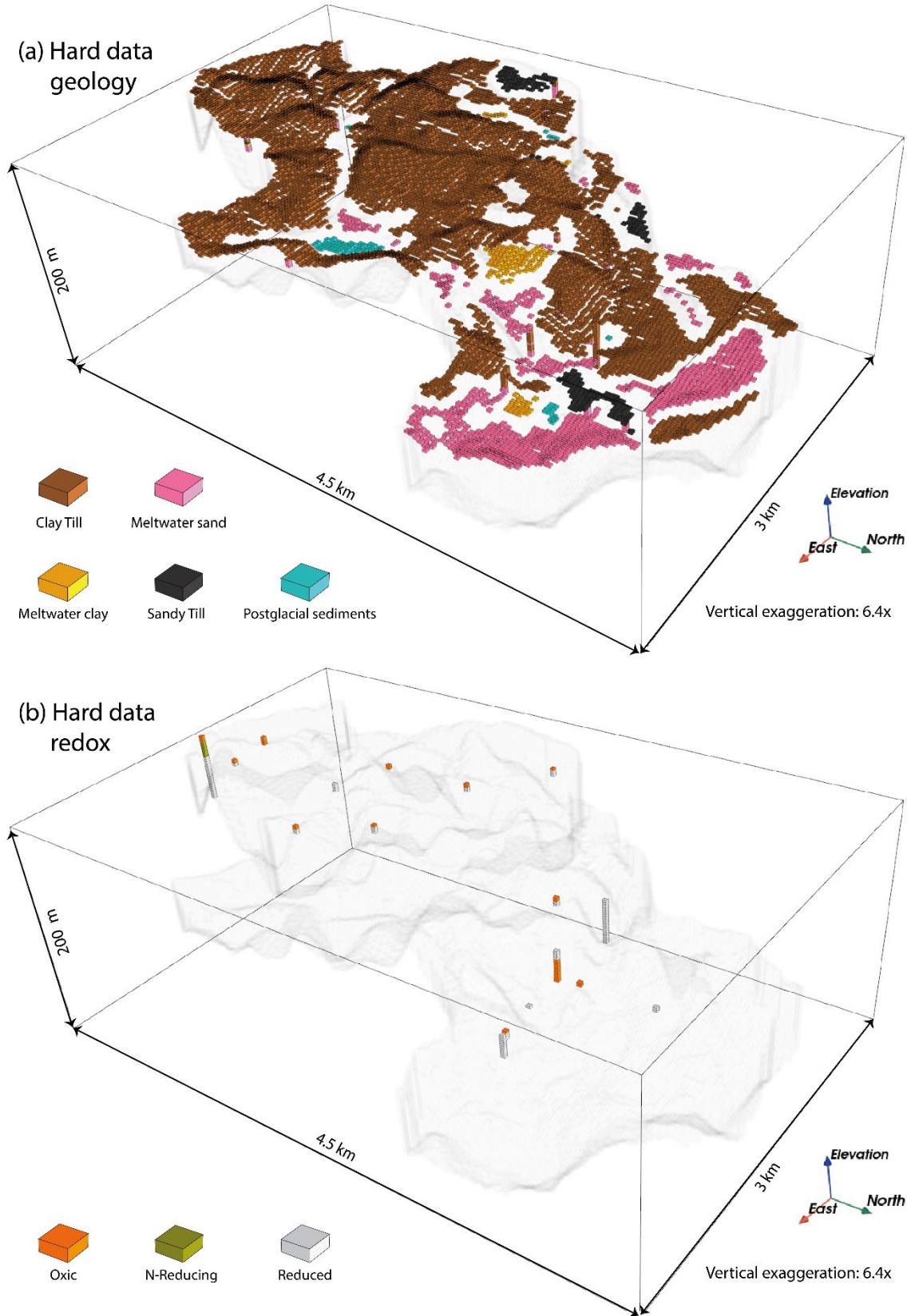

**Figure 6: a) The geology surface map along with the geology wells placed on the simulation grid as hard data. b) The redox wells on the simulation grid as hard data.**

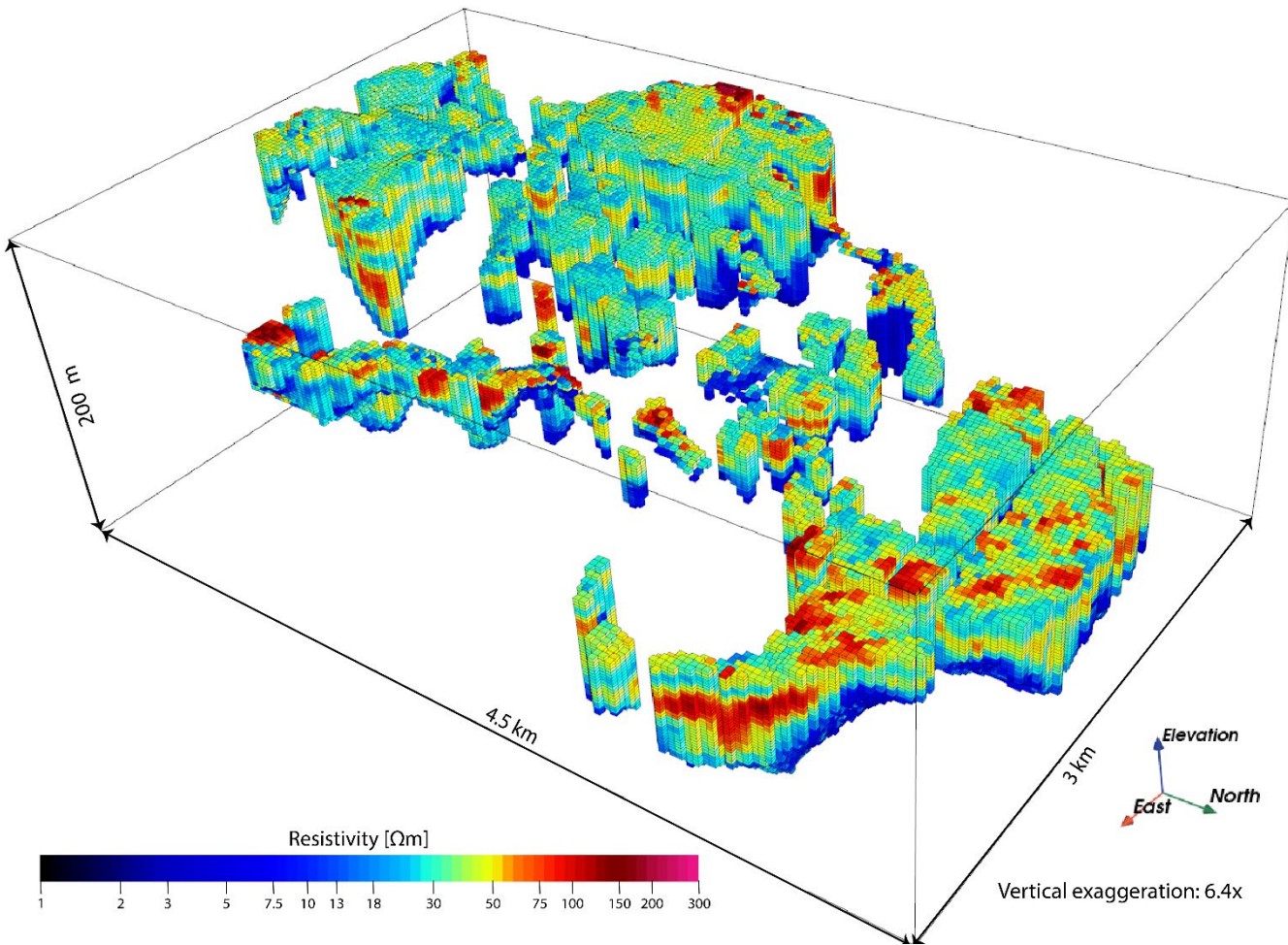

**Figure 7: 3D resistivity grid from the tTEM model results in a grid equal to the simulation grid.**

The tTEM 3D resistivities in the simulation grid contains 87547 voxels covering 43.5% of the simulation grid (Figure 7). The tTEM 3D resistivity grid is converted into soft data probabilities of geology. This requires a known lithology-resistivity relationship, which here is established in two parts.

Firstly, because the geological TIs are based on interpretations of resistivity data in combination with geological information,
many voxels in the TIs have a corresponding resistivity value in the resistivity grid (Figure 7). Local histograms for the study area are built for each lithological group by collecting all the resistivity values in the two geology TIs. We divide the TIs into three zones to account for some of the non-stationarity with depth that affects this relationship (Figure 4c). The upper 4 meters make up zone 1 and is the only place where we expect postglacial sediments and sandy till. Because both of these lithology classes contain so few counts in the TIs they would otherwise get underrepresented in a relationship covering the entire TI.
Zone 2 covers the bulk part of the TIs from 4 m below surface and down to zone 3 covering the last 10 m of the TIs. Zone 3 contains very low resistivities from the underlying conductive Paleogene clay that are "smeared" into the resistivities of the above lying material due to averaging during inversion (dark blue colors in Figure 7). This smearing effect happens at large contrasts in the subsurface resistivity and generally increases with depth as the resolution of the data decreases (Vignoli et al., 2015). This affects the inference of the lithology-resistivity by lowering the overall resistivity of meltwater sand/gravel that
mainly constitutes the lower parts of the study area. By separating the last 10 meters in a disconnected zone from the bulk zone, we minimize the effect of these low resistivities on the overall lithology-resistivity relationship in zone 2. The final pooled histograms for the two TIs are shown in Figure 8a-c for each of the respective zones. For all zones, relatively low resistivities are attributed to clay-rich deposits whereas relatively high resistivities are attributed to sandy lithologies, although meltwater sand/gravel accounts for many of the lower resistivity counts in the zone 3 relationship due to the smearing effect.
Generally, the resistivity of clay till is so high that it corresponds to much of the meltwater sand/gravel resistivities. Meltwater clay/silt is the most distinctive lithology group tending towards rather low resistivity values. The histograms confirms the

common issue of lithologies overlapping in the resistivity domain (Barfod et al., 2016; Schamper et al., 2014). The histogram with the best separation is seen in zone 2, which indicate the importance of detaching the low resistive meltwater sand in zone 3. The sandy till in zone 1 is associated with some of the highest resistivity values found in the TI area, whereas the postglacial sediments cover a large spectrum within the most ambiguous resistivity values. For each bin in a histogram, we summarize the size of each lithology group and stack them. If we then normalize with the total number of counts within that resistivity bin, we get a cumulative distribution of the lithologies (Figure 8d-f).

Secondly, because there are very few counts for the low and high resistivities, here defined as < 0.5% of the total counts for each zone, we let an a priori established relationship govern these values. We assume that low resistivities are associated with clay till and meltwater clay/silt, whereas high resistivities are associated with sandy till and meltwater sand/gravel. This is based on our general understanding of the lithology-resistivity relationship in the area and supported by Barfod et al. (2016) and Schamper et al. (2014). The proportion between e.g. the two low-resistive lithology groups are found by retrieving the proportion between clay till and meltwater clay/silt in the respective zone of the TIs (Figure 4). For instance, there is no meltwater clay/silt in zone 1 of the TIs and hence we expect that low resistivities are only attributed to clay till (Figure 8d), while meltwater clay/silt covers approximately 25% among the two low-resistive lithology groups in zone 2 (Figure 8e). To smooth the transition between the relationship inferred from the TIs and the a priori distribution, we weight the adjacent 10 bins between the two relationships. The weights are distributed linearly such that below the cut-off of 0.5% only the a priori relationship is used and 10 bins from there the relationship relies solely on the inferred relationships from Figure 8a-c. Regardless, the effect of the a priori relationship is miniscule as in all zones approximately 95% of all resistivities in the simulation grid are supported solely or at least partially by the relationship inferred relationship. The remaining 5% is supported solely by the a priori established relationship as seen in the total distribution of all resistivities in the simulation grid Figure 8c.

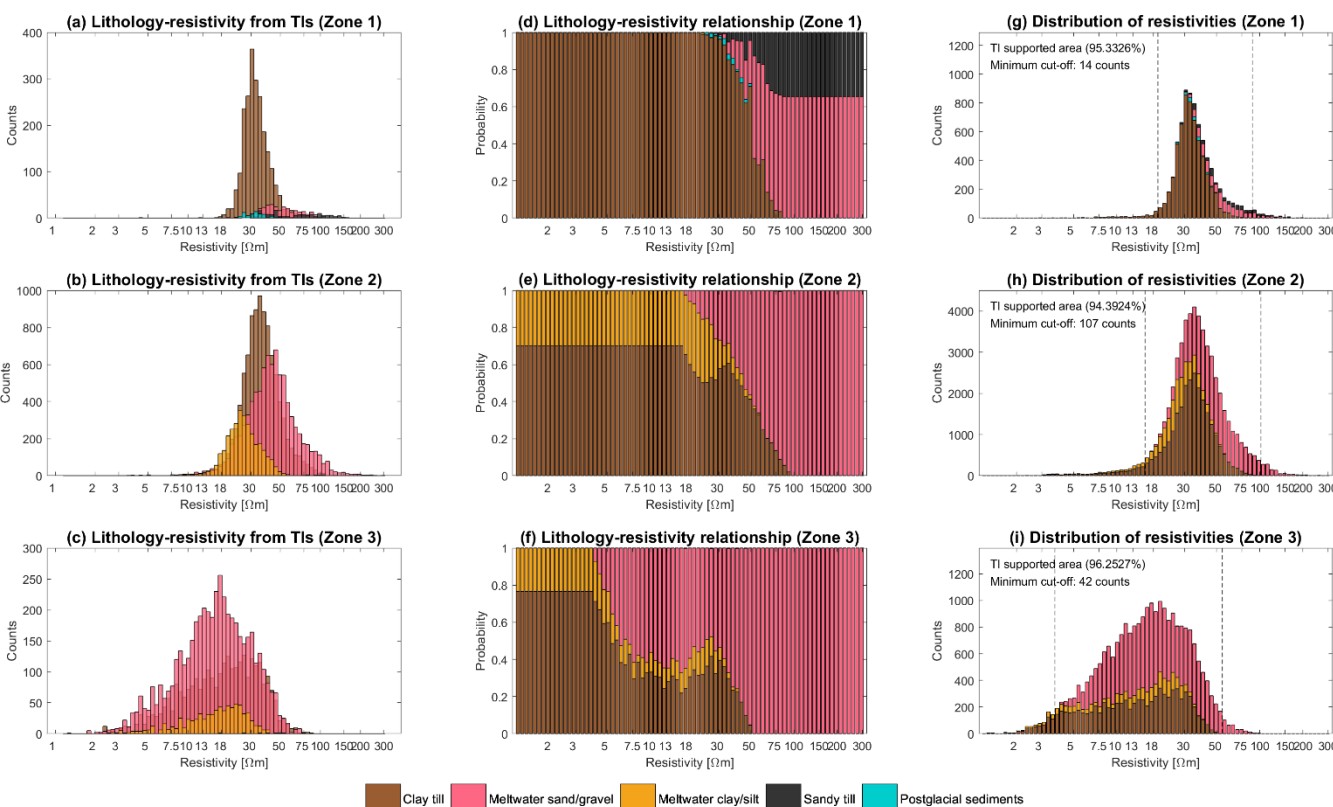

**Figure 8: Resistivity-lithology relationships illustrated as a)-c) histograms of resistivity for each lithological group based on the training images (Figure 4) and the corresponding values in the 3D resistivity grid (Figure 7), d)-f) TI-based cumulative distribution for all lithological groups for each bin and a priori relationship for rare resistivities g)-i) distribution of resistivity values in the corresponding zone in the simulation grid (Figure 9f) overprinted with the lithology-resistivity relationship established in d-f).**

Figure 9 presents the final soft data probabilities each of the $k$ lithology classes $p_{tot}(k)$. In zone 2 and 3 (Figure 9f) the inferred lithology-resistivity relationships from Figure 8e and Figure 8f are used to convert the resistivity grid from Figure 7 to soft data probabilities $p_{tTEM}(k)$. At the surface the soft data probabilities from the surface geology $p_{sg}(k)$ are combined with probabilities from the resistivity data to obtain the final soft data probabilities $p_{tot}(k)$:

$$p_{tot}(k) = \frac{p_{sg}(k)p_{tTEM}(k)}{\sum_{k=1}^{K} p_{sg}(k)p_{tTEM}(k)} \tag{2}$$


for each of the $K=5$ lithologies. The stronger colors at the surface represent the overall certainty level of 0.7 from the surface geology discussed previously. The tTEM data is largely more ambiguous in guiding the soft data probabilities as evident from the resistivity-lithology relationship in Figure 8 and is mostly within the color range of yellow and red in Figure 9a-c. The
dominance of the clay till and meltwater sand/gravel (Figure 9a-b) in the study area are apparent in the soft data probabilities when compared to e.g. meltwater clay/silt which is expected primarily in areas of lower resistivities. We do not expect much meltwater clay/silt at the boundary of the modeling domain as portrayed in the training images. The inferred relationship in zone 3 helps guide meltwater sand/silt to lower resistivities, but may not affect the results more than the general uncertainty in the boundary estimate, which depends largely on the tTEM resolution. Due to the low count of sandy till and postglacial
sediments (Figure 9d-e) in the TIs the probability for these lithology classes is considerably lower than the three main classes of the study area.

Based on these soft data probabilities a mode and entropy is calculated and shown in Figure 10. The entropy is generally low at the surface where the soft information from the surface geological map is present. Similarly the mode is dominated by the
soft information from the surface geological map. Due to the overlapping relationship in the resistiviy domain (Figure 8), the soft data based on the tTEM data is not as informative at the surface and does not help to lower the entropy much further. In general the entropy of the tTEM data ranges between 0.8 (yellow color in Figure 10b) and 0.3 (red color). In areas of particularly high resistivity the entropy drops even lower (black color), implying that the tTEM data provides high certainty on the lithology group. The overall pattern in the mode model (Figure 10a) reveals a slight tendency to form coherent layers,
especially seen in the buried valley. However, in many places the mode of the soft data is also raher patchy and changes between small clusters of either meltwater sand or clay till. These clusters simultaneously show high entropy (Figure 10b), which imply a wide distribtution of possible outcomes. Thus, these patchy structures can be consistent with information of more coherent layers.

If a single lithology group has a soft data probability greater than or equal to 0.5, a small fraction of this soft data is converted into hard data. This makes sure that soft data are not underrepresented in MPS simulations which is a recurring problem in MPS simulation (e.g. Hansen et al., 2018). The conversion rates based on the soft data probabilities are shown in Table 4.

**Table 4: Conversion rates for soft data in the conditional realization.**

| Soft data probability for a single lithology group | 0.5 to 0.6 | 0.6 to 0.7 | 0.7 to 0.8 | 0.8 to 0.9 | 0.9 to 1 |
|---|---|---|---|---|---|
| Conversion rate of soft to hard data | 2% | 3% | 4% | 5% | 6% |


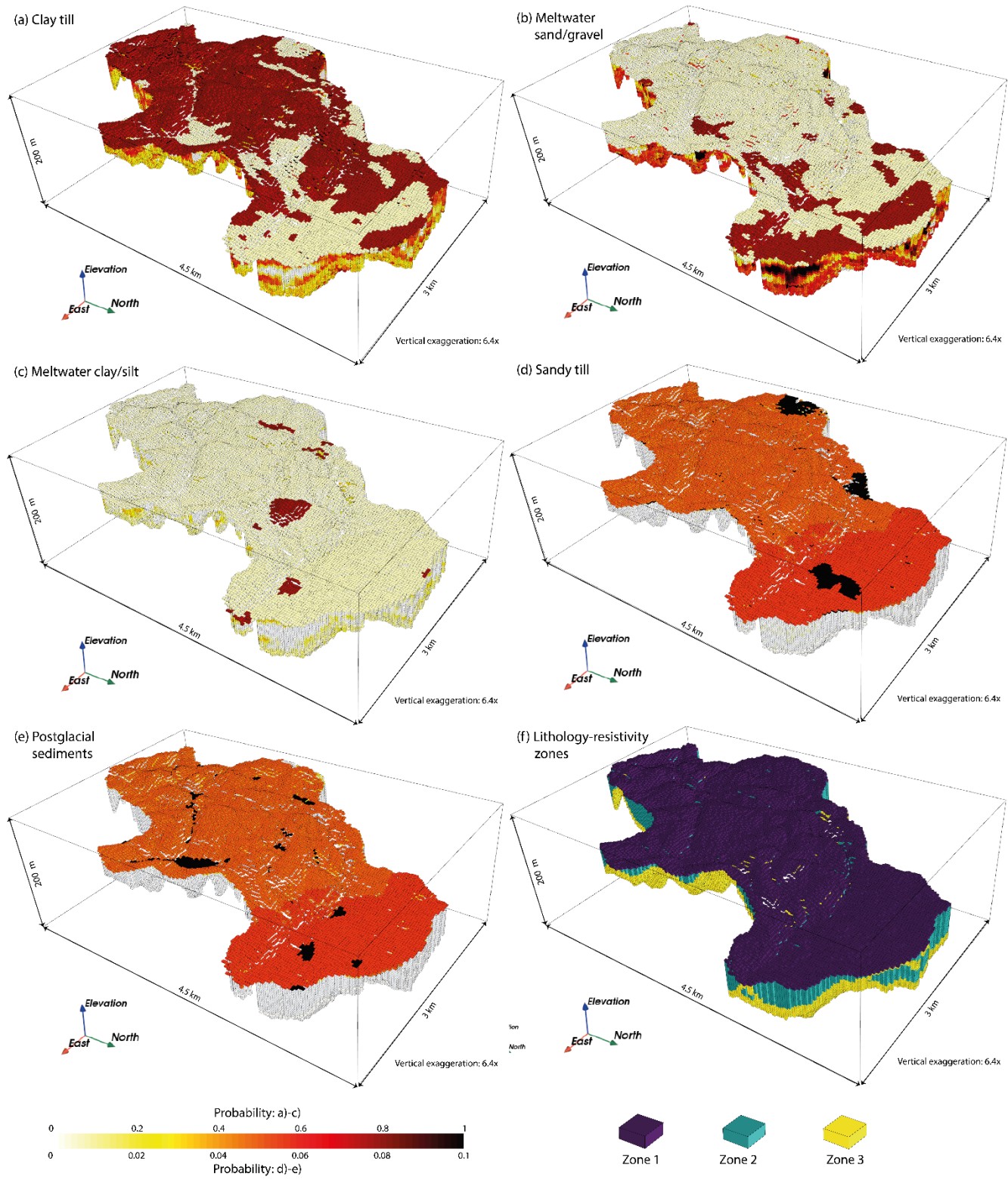

**Figure 9: a-e) Soft data probabilities of geology for the buried valley element. Soft data probabilities calculated from the surface geology and tTEM data available. Note the smaller range in the color scale of the sandy till (d) and postglacial sediments (e).**

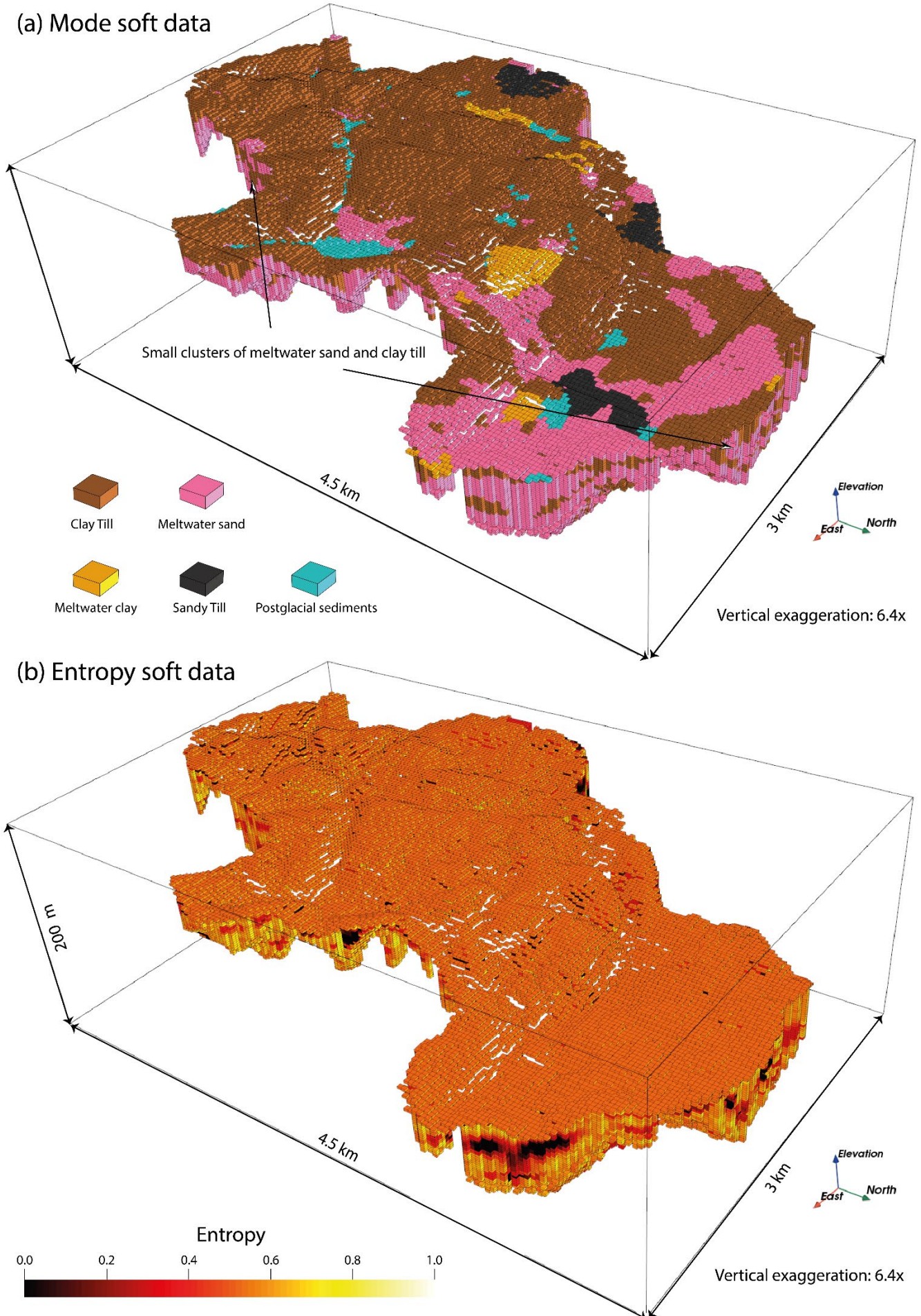

**Figure 10: a) Mode and b) entropy for soft data from Figure 9. Low entropy (certainty) is marked with black color, while white colors represent high entropy (uncertainty).**

## 5.5 Parameterization of the simulation algorithm

In direct sampling, the nodes in the simulation grid are visited sequentially. The training image is consulted at each iteration to find a suitable candidate at each visited node based on already simulated (conditional) nodes. To specify how this procedure is performed, several fundamental parameters need to be set in direct sampling (Mariethoz et al., 2010):

- The number of conditional data to consider when searching the TI, which influences the variability. Here, a maximum of 20 neighboring nodes are used preventing verbose copying from the TIs happening too often.

- The distance measure determining how well the candidate value match the conditional nodes in the simulation grid. Because both geology and redox are categorical variables we use the number of mismatching nodes as distance measure with a tolerance of 10% mismatch. For 20 neighboring nodes, we hence allow 2 conditional nodes to differ between the TI and the simulation grid to accept the currently proposed value.

- The maximum number of iterations allowed to find a suitable match within the TI. Because the TIs in current study are of a reasonable size, we allow a scan of the entire TI to find a suitable match. This alleviates some of the problems with the non-stationarity mentioned earlier. If a match is still impossible to obtain, the candidate providing the lowest misfit is retrieved and "flagged". During post-processing the flagged cells are simulated again using the same simulation setup and TIs. Because the larger structures are placed during the initial simulation, the flagged cells in postprocessing have a higher probability of finding a matching event in the training image, which minimizes the appearance of simulation artifacts.

- The path at which the simulation grid nodes are visited needs to be selected. We choose a random path as is often used in MPS simulation. When combined with conditional hard data, the random path preferentially first visits nodes that are in the vicinity of hard data. This is achieved by calculating distances to hard data and then randomly drawing nodes according to these distances to create the visitation path (Straubhaar, 2019). This ensures that especially hard data from the surface have a higher impact on the final realizations.

As pointed out by Tahmasebi (2018), a quantitative evaluation of the performance of MPS is still unresolved and the effect of the simulation algorithm parameterization remains an area of active research (Juda et al., 2020). To ensure that the combination of TI and MPS algorithm produce the sought-after spatial variability, we simulate 10 independent realizations without including the conditioning data, i.e. two realizations from the prior model. We adopt the heuristic strategy of Høyer et al., (2017), making sure that the realizations from the prior model are in accordance with and represent our expectations of both redox and geology. Two unconditional realizations from the prior model are shown in Figure 11. The spatial variability and patterns seen in the TIs (Figure 4) are generally represented for both redox and geology. As expected in the TI and conceptual model for geology, the prior realizations show primarily horizontal stratification. In the buried valley infill the extent of geological layers and redox structures is more limited than in the Quaternary sequence, which is also in accordance with our conceptual understanding. In the Quaternary sequence, the geological layer order is correct with clay till predominantly found near terrain while meltwater deposits are the main constituent of the deeper parts. Both sandy till (black) and postglacial sediments (blue) only occur near the surface in accordance with the TIs, much more infrequently than portrayed.

For redox, the layer order from the TI is likewise preserved in the unconditional realizations such that oxic conditions are found primarily at the surface with increasing N-reducing and reduced conditions at lower depths. The prior model also captures the possibility of secondary redox zones from geological windows that are portrayed in TI1. N-reducing conditions are found adjacent to oxic conditions at the surface and not in the bottom of the simulation domain in the unconditional realizations. The overall redox conditions can be visualized by plotting the accumulative probability for redox conditions as a function of depth, constructed by summarizing over both realizations, which hence provides the 1D marginal distribution in all voxels. This marginal distribution is accumulated with depth as shown in Figure 12. Less oxic and N-reducing conditions

(orange and green) are simulated in the prior model at the surface and does not stretch as far down as portrayed the TIs (Figure 12a), which can also be visually confirmed comparing Figure 4b and Figure 11b and f.

Due to the strict vertical layer ordering in the TI, the non-stationary characteristics are preserved in the unconditional realizations despite the expectation of a stationary training image in MPS. We suspect that the full scan of the training image
helps to provide the necessary configurations to enable a more non-stationary output in the prior realization. However, the MPS algorithm cannot fully capture all the non-stationarity of the TIs as there is a tendency to simulate less oxic conditions at the surface along with sandy till and postglacial sediments being underrepresented. Furthermore, the size of the TIs may hinder the reproduction of large-scale connected structures such as the oxic conditions at the surface (de Vries et al., 2009). This tendency is hence beyond immediate remediation by changing any of the fundamental parameters in direct sampling but can
instead be guided by the incorporation of conditioning data in the posterior model (Barfod et al., 2018). In summary, we conclude that the current parameterization of the direct sampling algorithm provides the spatial variability that fits our understanding of the system, albeit with some slight caveats. With the current simulation setup flagging occurs for approximately 8 % of the cells during initial simulation and 4 % after post-processing. We emphasize that the unconditional realizations represent the prior information of the system, not the TIs nor the exact parameters chosen in the DEESSE
algorithm.

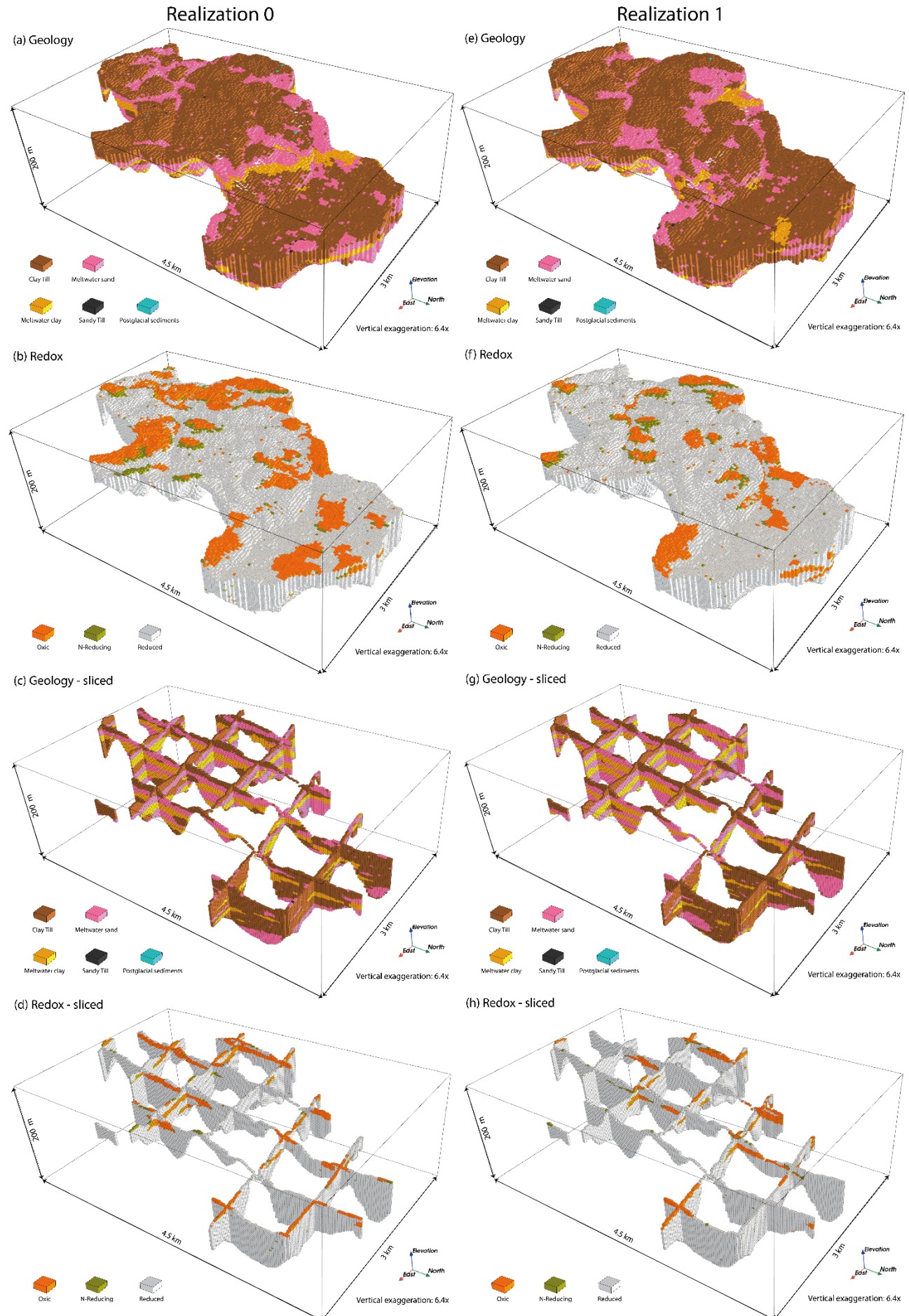

**Figure 11: Two unconditional realizations from the prior model. a-b) One realization of jointly simulated geology and redox c-d) The same realization sliced in the X and Y direction. e-h) Same figure configuration as in a-d but for a different realization.**

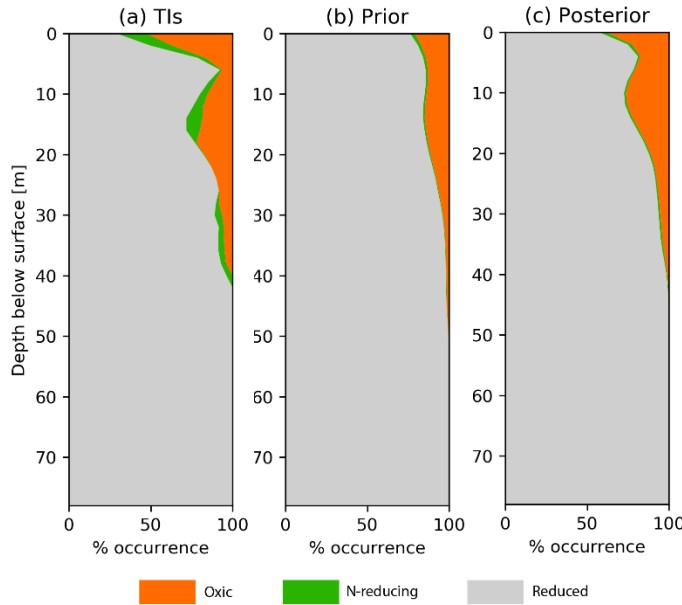

**Figure 12: Accumulative probability profiles of redox conditions in the study area for a) TIs, b) prior distribution and c) posterior distribution**

## 6 Modeling results

In this section, we present the modeling results from the set of posterior realizations of both geology and redox where the information from the prior model is conditioned to the data. We condition the simulation to the hard and soft data presented in section 5.4.

Figure 13 shows two conditioned realizations from the posterior model. The impact of introducing the conditioning data is immediately seen at the surface of the geology simulations (Figure 13a,e), which is guided to a large degree by the information from the surface geology map. The architecture stays relatively fixed between the realizations, and variability is predominantly small-scale. Given the high amount of conditioning data, this is not unexpected. The main part of the Quaternary sequence element is covered by an approximately 8-10 m (sometimes reaching more than 20 m) thick clay till, followed by meltwater deposits. These meltwater deposits exhibit a shorter correlation length than in the prior model as seen in Figure 13g. The lateral extent of layers in the buried valley is less than in the Quaternary sequence, but not as significant as in the prior model. In general, the amount of meltwater clay/silt in the posterior model is lower than in prior model and the realizations consist mostly of either clay till or meltwater sand/gravel. This change is due to information from the geology soft data which is heavily dominated by clay till and meltwater sand/gravel (Figure 9). In fact, in zones of high resistivity, the soft data is the dominant constraint on the realizations with meltwater sand/gravel causing low variability between the two realizations as seen in e.g. Figure 13a,e. Just northwest of the high resistive zone in the buried valley is an area with more ambiguous resistivities which leads to greater variability and more dependency on the prior model. The bottom of the simulation domain is mainly made up of meltwater sand/gravel which is likely information stemming from the prior model.

Due to the joint simulation of geology and redox in the current setup, the overall redox architecture in the realizations is coherent with the geology as outlined in the TI. For example, postglacial sediments are attributed to reducing conditions and meltwater sand/gravel is likely oxic at the surface. This consistency explains the predominantly oxic conditions at the surface seen in the sandy part of the buried valley (Figure 13b,f). In the Quaternary sequence, the clay till at the surface show both oxic and reduced conditions as indicated in the TIs (Figure 13d). Oxic conditions are clearly more present at the surface of the posterior model than in the prior realizations. The oxic conditions are distributed in the low gradient parts of the simulation domain, whereas reduced conditions are found along depressions in the landscape such as valleys and streams (Figure 13b), which is in good accordance with our geochemical understanding of the system. The entire posterior redox probability profile in Figure 12c also resembles the TI profile better than the prior model. Because there is no soft data aiding the occurrence of N-reducing conditions in the posterior model, it inherits the capacity of simulating N-reducing conditions from the prior model and is simulated less than in the TI profile from Figure 12. Thus, N-reducing conditions are also simulated adjacent to oxic conditions as in the prior model. The overall redox architecture is in place with planar type redox conditions in the buried valley and geological window type conditions in the Quaternary sequence (Figure 13d). However, sole voxels of oxic conditions in the deeper parts of the realizations appear as unwanted simulation artifacts (Figure 13h). Because these artifacts happen infrequently, are tiny and are surrounded by reduced conditions, we argue that for N-retention simulations these artifacts may be negligible.

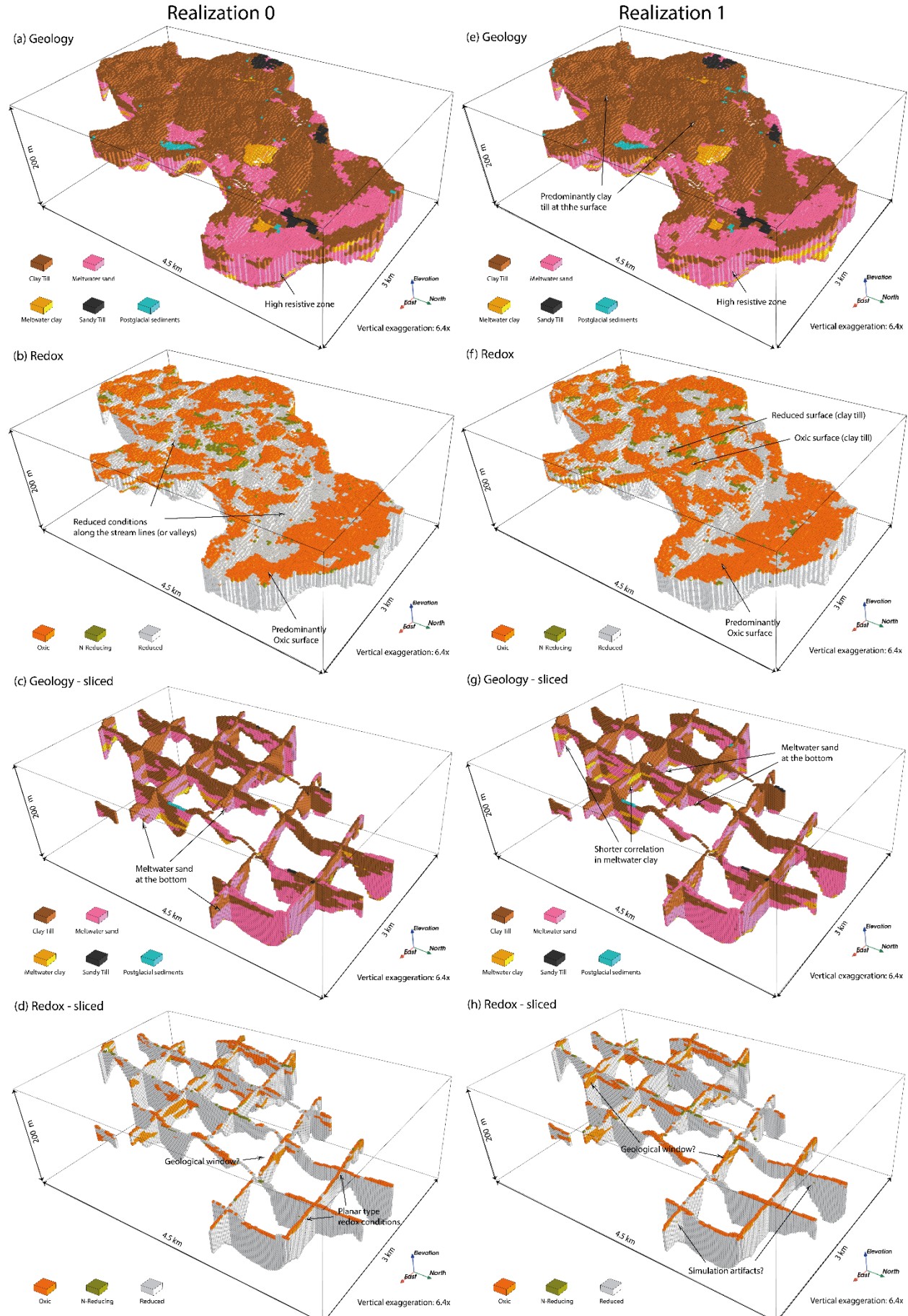

**Figure 13: Two conditioned realizations from the posterior model. a-b) One realization of jointly simulated geology and redox c-d) The same realization sliced in the X and Y direction. e-h) Same figure configuration as in a-d but for a different realization.**

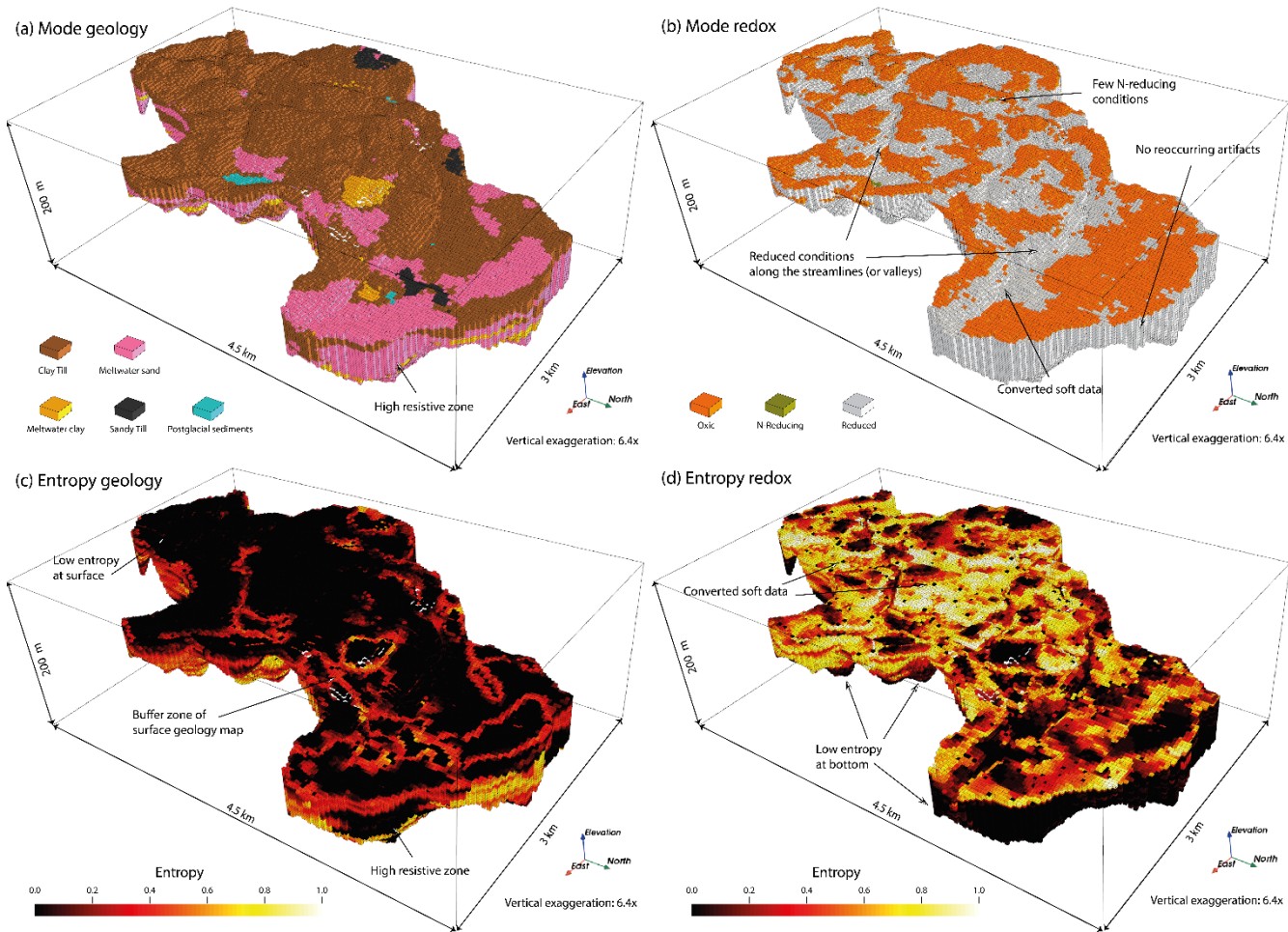

**Figure 14: Mode and entropy of 100 geology and redox realizations. Low entropy (certainty) is marked with black color, while white colours represent high entropy (uncertainty).**

In total, we simulate 100 realizations like those presented in Figure 13, which together are used to represent the full posterior model. To summarize the posterior model, we present ensemble statistics (from section 4.2) in Figure 14. The entropy of geology (Figure 14c) at the surface is 0 at most locations due to the hard data provided from the surface geology map. The more uninformed parts of the surface (lighter colors) correspond to the buffer zone in the surface geology map. The entropy is usually around 0.2-0.3, indicating that most realizations of the posterior model provide the same outcome in the buffer zone. In few places along the buffer zone an entropy level of 0.8 is reached, indicating that these voxels have a near uniform distribution among several different categories to the one shown in the mode model. This confirms the qualitative results from inspecting the individual realizations, that the effect of introducing prior information and hard data increase the information content (lowers entropy) of the final models drastically compared to the soft data only (Figure 10). For the lower part of the simulation domain, the posterior model shows higher entropy than at the surface. This means that the mode found in this region is usually more uncertain. In some areas of high resistivity, we also see very low entropy at depth, where the soft data provides the main architectural input and the posterior mode model resembles the soft data mode. The effects of introducing the prior information for the architecture is clearly seen in the coherent structures produced in the posterior mode in contrast to the patchiness of the mode in Figure 10a.

The redox mode does not display some of the minor simulation artifacts seen in the individual realizations, because these are averaged out over many realizations. Instead, we do see remnants of the converted oxic soft data at the surface of the mode model (Figure 14b) with zero entropy (Figure 14d). This is clearly a side-effect of the soft data conversion, since these sole

oxic voxels are not in accordance with the overall pattern of reduced conditions along the streamlines and valleys. The redox mode shows few N-reducing conditions, in accordance with the redox depth profiles shown in Figure 12c, demonstrating the inclination to simulate either oxic or reduced conditions in the posterior model.

Overall, the entropy of redox is showing a reverse pattern to that of geology. The redox entropy is highest near the surface and decreases with increasing depth (Figure 14d). One could at first expect the highest redox uncertainty at deeper depth because the density of the hard data is much higher near the surface and for the deeper part of the architecture, the geochemical data is rarely available. However, the entropy sharply decreases in the reduced zone beneath a certain depth. This pattern instead fits well with the conceptual understanding of the redox structure evolution: oxic conditions are developed as oxidants (e.g., 680 oxygen and nitrate) infiltrate from the root zone to the subsurface where reduced layers are present. Therefore, a redox front propagates downward and under homogeneous conditions with vertical flow of water, it would be unlikely to develop oxic conditions below the redox front. While the spatial heterogeneity of the geological settings of the near surface environments at various scales (pore scale to landscape) has been well documented (e.g. Baveye et al., 2018; Groffman et al., 2009; Sexstone et al., 1985), implying highly heterogeneous redox conditions in the shallower depth. The sharp decrease in entropy of the 685 buried valley take place due to the planar redox type domain in the buried valley, whereas the possibilities of geological windows in the Quaternary sequence makes the high entropy section develop further down. Some of the voxels at the surface of the Quaternary sequence element depart from the overall pattern by having a very low entropy. This trend is likely aided by the soft data giving high probabilities of oxic conditions at the surface. The high entropy at the surface is likely also aided by the soft data. At the surface and down to about 10-20 m of the buried valley, we generally do not know much about the redox 690 conditions as indicated by the white yellowish colors in Figure 14. The more evenly distributed redox soft data probabilities (Table 3) could explain some of this high entropy.

## 7 Discussion

To our knowledge, examples of simulating redox conditions with sediment-texture distributions using multiple-point geostatistical methods have not been done before. This study sets out with the aim of proposing and reviewing a methodology for modeling both redox architecture and geology simultaneously in high-resolution 3D using MPS.

### 7.1 Simulation artifacts

The spatial variability of the TIs is represented in the prior realizations and conditioning data guides our expectations of the system to a posterior model. In some cases, due to the limited size of the TI, inconsistencies between conditioning data and the prior model exist. In some cases, these inconsistencies lead to simulation artifacts in the realizations, but are rare since they are largely corrected during MPS post-processing. For the posterior model realizations, flagging is decreased to only about 5-6% after post-processing while happening 19-22% of the time during the initial run of the algorithm. Some simulation artifacts also occur in the prior model itself and therefore cannot alone be attributed to inconsistencies between the prior model and the conditioning data, which is underlined by the decrease in flagging that also happens during post-processing of the prior model realizations. These inconsistencies are associated with a lack of matching events in the TI. To remedy such simulation artifacts one either needs larger TIs to allow more spatial variability or artificially enhance the variability by lowering the amount of conditional data to consider when searching for a match in the MPS algorithm. In the latter case, this will happen at the cost of reproduction of the actual spatial variability portrayed in the TI. We argue, that in the current nitrate simulation at the catchment scale, these artifacts do not affect the overall architecture (Figure 14) and redox trends with depth (Figure 12). It is also expected to have negligible impact on hydrological modeling as the overall architecture allows groundwater flow pass by such artifacts. Nevertheless, future studies are required to reduce artifacts of this kind or, at least, downplay their significance. One solution could be to allow rotation during simulation could offer more configurations during simulation. But testing with a setup allowing 360 degrees rotation in the horizontal plane did not enable a substantial improvement on this issue. The flexibility of the current methodology also allows the inclusion of soft data probability maps through equation 2 indicating spatial restrictions on certain lithologies or redox conditions, which could potentially remedy some of the deeper lying artifacts.

### 7.2 The role of soft data

The random path have a tendency to underestimate soft data and provide less resolution in the results compared to other path types (Hansen et al., 2018). In the current study the amount of soft data coverage was high (more than 43.5% of the simulation grid). To utilize the abundant soft data, we randomly converted a fraction of the soft data into hard data to compensate the underestimation from the path. This helped transferring more weight towards the soft data during simulation, with the caveat of introducing converted soft data in unwanted positions, such as oxic in an overall reduced environment. This problem is however mostly encountered at the redox mode (Figure 14b) and does seemingly not pose as big of a problem for the individual realizations in Figure 13b,f. By further processing the realizations by removing any sole voxels that differ from the neighboring voxels, this problem can be removed entirely, but at the risk of removing actual sole voxels. One could also randomly select a new set at each iteration, although this is not directly implemented in the DEESSE software and still would not make sure that soft data in general are handled correctly. For instance, the current remediation only handles lithology groups with probability >= 50% and thereby cannot help improving the information content for any categories with probability < 50%. This affects e.g. N-reducing conditions at the surface where soft data probabilities are substantially lower (Table 3). Thus, N-reducing conditions are bound to be underrepresented since they are not converted from soft data, which is the tendency shown in the posterior redox profile compared with the TI (Figure 12). Despite the clear advantages of converting some soft data to provide more emphasis on them, the current simulation results could most likely be improved by better incorporating the soft data

information in general. However, neither a preferential path that visits voxels with soft data information before other voxels, nor the use of non-collocated soft data is currently implemented in most state-of-the-art MPS algorithms. The problem of how to best incorporate soft data information hence reaches beyond the current study. We encourage that this remains an active area of research to make MPS simulation relevant for practitioners without the need for too much ad-hoc remediation.

## 7.3 Resistivity-lithology relationship

The established resistivity-lithology relationship allows us to map the prior probabilities of each lithological group based on the tTEM in the simulation area. Utilizing tTEM as soft data information ensures that it does not have too much influence over the final results. Here, the relationship is inferred from the resistivity grid and training images. When simulating, the general mismatch between the training image patterns (based on interpreted geology) and the tTEM data is thus minimized. Methods exist for establishing a relationship between resistivity and clay content (Christiansen et al., 2014; Foged et al., 2014). Unfortunately, this is not directly applicable for the lithological groups used here as they are not defined on the basis of the clay content. Alternatively, this relationship could be inferred using boreholes near the study site. Similar to the approach in this study, inferring the resistivity-lithology relationship from boreholes is typically based on deriving probabilities from histograms (Barfod et al., 2016; Gunnink and Siemon, 2015; He et al., 2014a). In accordance with the present results, these studies also show a significant overlap between different lithologies and as such using nearby boreholes for inferring the resistivity-lithology relationship would mainly minimize the reuse of data and avoid subjectivity carried over from the TIs.

## 7.4 Geological modeling subjectivity and data reuse

The inclusion of geological mapping experts in the creation of TIs introduces modeling subjectivity. Thus, the final realizations could include unverifiable modeling choices following the interpretation procedure in cognitive modeling. Through experiments with geological interpretation of the uncertainty in boreholes, Randle et al. (2019) argued that expert elicitations do not result in accurate predictions of interpretation error. Schaff and Bond (2019) propose the quantification of interpretation uncertainty for inclusion in geostatistical simulation, while efforts have been made to make TI generators (Pyrcz et al., 2008) and data-driven TIs without the need for expert knowledge (Vilhelmsen et al., 2019). However, our approach of process-based TI generation from expert elicitation is a common approach in MPS applications (Mariethoz and Caers, 2015). A possible explanation for this is the benefit of bringing in prior expert knowledge, which is otherwise difficult to quantify. This ensures that results are in accordance with as much information as possible (Curtis, 2012; Tarantola, 2005) and realizations are not in clear conflict with geological concepts (Jessell et al., 2010; Wellmann and Caumon, 2018).

Despite the potential subjectivity in the geological modeling of the study area these modeling choices are primarily guided by data. The tTEM data collected in this study has e.g. contributed to a good correlation between the terrain and the subsurface architectures in the geological interpretations. These observations fit well with the current knowledge of the latest geological events in the area, thus providing good possibilities of making robust geological correlations between the geological and geophysical data.

It might be difficult to quantify the effect of the apparent loss in degrees of freedom that follows from using the same data for establishing the prior information and as condition data during simulation. In the current study, the problem of reusing data for outlining geological elements, is most likely not critical as only large-scale structural information is partly interpreted from the resistivity data, such as the top of the Paleogene clay layer. The degrees of freedom loss for reusing the resistivity information in the TIs and as conditioning data in simulation is undoubtly larger. Although the small size of the TIs may pose

a problem for reproducing the intended varibility, in this instance it acts to limit the effect of reusing data. This issue persist for approximately 33% of the total voxels (Table 1).

## 7.5 Training images and geological elements

If possible, the TI should provide all possible dimensions and shapes of the geological features in the subsurface (Strebelle,
2012). However, sizes of the TIs in the current setup are relatively small compared to the simulation grid and hence do not contain that many configurations. In general, the smaller the TI, the fewer possible structures can be represented (Mariethoz and Caers, 2015). We consider two remedying factors. Firstly, the simplicity of the TI. In the study area, we expect a geology with continuous clay and sand units partly restrained by incised valley structures in the Paleogene clays as seen in Figure 3. Even though the TI is small and simple, it conveys the general pattern to be expected in geological features throughout the
simulation domain. The simplicity should alleviate some of this issue, although in an area with more expected heterogeneity, a more diverse and larger TI would be needed. Secondly, if the geological variability provided in the TI is not sufficient, algorithmically induced variability measures such as scaling and rotation of features is possible with direct simulation (Mariethoz et al., 2010).

The non-stationarity of both sets of TIs is evident. This is a common problem when designing training images directly based on, and mimicking geology, which is inherently non-stationary. This might pose a problem, as only a certain number of the configurations in the TIs will produce a match during the direct simulation. Consequently, we might risk reproducing larger parts of the TI in the realizations. Such verbose copying is partially remedied by the addition of conditioning data and choosing a smaller search radius as argued in Vilhelmsen et al. (2019). However, a smaller search radius comes at a price of not
reproducing the features in the TI and adding variability more related to algorithmic choices than geological variability. Luckily, plenty of conditioning data is available for the simulations to remedy some of the shortcomings of the training images. As argued in de Vries et al. (2009), subdividing the TIs and simulation domain into different areas is another possibility to handle non-stationarity. To some degree, the geological elements represent such a subdivision of the entire modeling domain in the study area.


In the current study, we considered the boundaries between the geological elements fixed. In reality, there is some interpretation uncertainty related to these boundaries especially in data scarce areas. Future studies may be able to quantify this uncertainty. If this uncertainty is sufficiently large such that it affects the simulation results significantly, we put forward the idea of re-simulating boundaries between geological elements as part of the simulation.


Because TIs are attributed to a specific geological element, these TIs may be reused in other simulation studies with comparable geological elements and we therefore strongly recommend building a TI library. This approach would alleviate the most fundamental of the issues in the current setups. Information between TI and data becomes independent when using a generalized TI. Specifically, the reuse of data (in constructing the TI and implicitly when inferring the resistivity-lithology
relationship) is eradicated. For a smaller geological element, the TIs developed in the study area may also represent a proportionally larger portion of the expected variability. An additional bonus would be a reduction in labor/time since TIs are pre-existing or maybe only need slight alteration.

Conceptual TIs or based on data from another study area would most likely be preferable from a geostatistical point of view
as it would ensure independence of information. However, in the case of a TI based on nearby data, the TI should be close enough to the study area such that the depositional and redox setting are comparable. Furthermore, the study of Barfod et al.

(2018) suggests that TIs become secondary given a high amount of conditioning data. In future studies and if a similar approach of TI creation within the simulation domain is chosen, we recommend collaborative efforts between geologists and geochemists in securing the best possible location for representative TIs. We also suggest that the level of detail in the TI should be case-specific involving a trade-off between the time to construct the TI, the level of support in the available data, the background knowledge, limitations due to size and how well the features can be reproduced with the chosen parameterization.

## 7.6 Computationally attractive stochastic simulations

In the current setup, simulations are computationally feasible. 100 realizations of both elements are generated in less than 2.5 hours on a high-end personal laptop (Intel(R) Core(TM) i7-8850H CPU @ 2.60GHz, 6 cores (12 threads) with 10 threads allocated to DeeSse. The average simulation time for a single realization is hence just over 80 seconds. Several factors contribute to this: 1) The relatively small TIs making the number of possible combinations limited, 2) The restriction on maximum 20 conditioning points and 3) the subdivision of the simulation grid into geological elements. Some of the abovementioned factors are algorithm tuning parameters, while others are added bonuses of understanding the geology in question (e.g. the ability of breaking the problem up into smaller bits and choosing an acceptable level of simplicity in the models). In this case, bringing expert field knowledge to the modeling setup is advantageous.

## 7.7 Multi-purpose modeling results through uncertainties

The proposed workflow allows incorporation of quantified uncertainties in the input data and structural uncertainties in the subsurface models. This is a major advantage over e.g. static models. We specifically dealt with prior uncertainty in the geological and redox conditions as portrayed in the TIs and geological map and resistivity data (soft data). Other sources of error (e.g. modeling and measurement errors) in the input data can also be explored, as MPS offers a flexible setup for treating data with uncertainties. Additionally, it is clearly shown in the comparison between mode and entropy of posterior and soft data that MPS adds additional valuable information through the TIs that enable geologically viable architecture. Especially in cases where soft data is too weak to provide significant support. The quantitative description of uncertainties as portrayed by the final ensemble of realizations also has many useful properties for additional analysis. For instance, the ability to produce redox profiles as in Figure 12 is trivial once the simulation is completed. These redox profiles make comparisons with previous studies possible, while offering many other possibilities for summary statistics and quantifying uncertainty. This flexibility in the final analysis is one of the main benefits of applying geostatistical mapping of redox conditions (and geology). With the current methodology, depth profiles can also be calculated for specific sets of x- and y-coordinates to investigate some of the spatial variation in redox. Another example would be to investigate the distribution of redox conditions in the geological groups, which allows assessing new hypotheses on the coupling between geology and redox. It may also reveal insights to the spatial dependencies of such couplings and showcase potential geological windows for oxic conditions at depth. Entropy gives insight into the nature of information content and therefore it would be an active tool in finding the best spot for further investigation, i.e. showing where information is lacking. For instance, in the case of redox, entropy might be suited for assisting a focused field campaign in retrieving more information of redox in the buried valley element. In the current case, the Quaternary sequence many places showed a lack of information in the first 10-20 meters that is typically critical to model.

From the study area, it seems that it is possible to create a computationally feasible joint stochastic 3D high-resolution model of redox and geology with the current setup. However, these findings cannot be extrapolated directly to other study areas. Future research includes testing the method in other catchment areas to assess the robustness and general applicability. Many improvements, besides fine-tuning algorithm parameters, also exist. We e.g. expect improvements and minor changes to the

overall setup, as different study areas will contain site-specific challenges that should be addressed. As mentioned, one of the current issues that need to be adressed is how best to quantify and integrate soft data. Besides the resitivity-lithology

relationship, we also recognize the need for an extensive study on the quantification of uncertainty in geological maps such as the geological surface map presented here, but it is beyond the scope of the current study.

**8 Conclusion**

This study sets out to model both redox architecture and geology simultaneously in high-resolution 3D due to the dependency of the evolution of the subsurface redox conditions on the hydrogeological pathways. This is achieved using a bivariate MPS simulation. MPS modeling with a bivariate TI of geology and redox presents some important features compared to previous mapping studies: 1) MPS simulation effectively produces geology and redox following expectations and 2) TIs provide an intuitive and easy collaboration across different fields of expertise. Valuable expert information, otherwise difficult to quantify, is seamlessly integrated within MPS. This ensures in our case that there is a correspondence between geology and redox conditions, which is one of the key strengths of the proposed methodology. Although challenges in the current approach exist, we conclude that the proposed methodology offers improvements to existing methods for mapping geology and redox by producing consistent realizations of both variables. The flexibility of the geostatistical results as represented by the ensemble of realizations allows comparisons with traditional mapping techniques. We interpret and model individual sedimentary layers into coherent volumes ('geological elements') that greatly help to guide our simulation results and reduce computation costs. This new mapping technique should aid our understanding of the uncertainties and limitations of our knowledge and data. High-resolution 3D understanding of both redox and geological architecture will likely improve predictions of N-retention and water pathways in the subsurface. The generalizability of these results is subject to certain limitations as the proposed workflow is only tested on a single study site. This study lays the groundwork for future research into coupled understanding of geology and redox using MPS simulation. Despite its exploratory nature, this study offers valuable insights into the feasibility of joint geostatistical modeling of redox and geology. Several questions remain to be answered regarding interdependence between different sets of quantified information and integration of soft data. The geological and redox architecture simulations might be incorporated in hydrological modeling with N-transport to be used for N-retention mapping of the subsurface important for future more targeted N-regulation of agriculture.

**Author contribution**

RBM, HK and AJK have been primarily in charge of methodology development. The idea of using geological elements at the site was conceptualized by PS. Geological modeling was carried out by AJK with assistance of PS, while HK carried out the geochemical analysis under the assistance of BH. IM and AVC provided valuable insights to geophysical data and models. TMH and TNV provided crucial sparring on geostatistical modeling from an applied and theoretical perspective. RBM developed the model setup and performed the simulations, with quality control on the final realizations by HK, PS and AJK. RBM prepared the first draft of the manuscript with contributions from all co-authors.

**Competing interests**

The authors declare that they have no conflict of interest.

**Acknowledgements**

The authors would like to thank Innovation Fund Denmark (8855-00025B) for sponsoring the Mapfield project (www.mapfield.dk) from which this study emerges.

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
