# Peer review of "3D multiple point geostatistical simulation of joint subsurface redox and geological architectures"

_Hydrology and Earth System Sciences, 2020_

## Referee Comment (RC1) · Donald A. Keefer (Referee) · 24 Oct 2020

This manuscript is included for consideration of a special issue: Frontiers in the application of Bayesian approaches in water quality modeling. It is important to note that the research described in this manuscript does not use Bayesian methods, and so does not directly relate to the topic of the special issue.

Leaching of agriculturally-applied nitrogen is a significant threat to global groundwater resources. Success in reducing nitrate contamination in groundwaters by modification of farming practices and reductions in fertilizer input has been only occasionally realized and more work is definitely needed. Prediction of the fate and transport of

fertilizer-based nitrate is a complicated problem that is confounding successful remediations to the groundwater contamination problem. Modeling success is significantly limited by the natural heterogeneities and complexities of the subsurface geologic systems and by our limitations in characterizing and modeling those complexities. The complex nature of the heterogeneities has led to the treatment of these systems as stochastic which has subsequently led to a large interest in probabilistic modeling approaches.

Multiple Point Statistics (MPS) has proven to be an innovative and successful probabilistic approach to modeling inter-facies relationships, primarily within distinct, lithostratigraphic units. As typically used, MPS is not a method that can be used with stratigraphic units or with units that are genetically distinct (e.g., from subglacial environments and glaciofluvial environments), unless each genetically-distinct succession is modeled within separate portions of the model domain. To my knowledge, a modified MPS approach has not been successfully applied to modeling of larger stratigraphic assemblages of rocks or sediments. It would be particularly relevant and innovative if the authors were to present a method for adapting MPS to stratigraphic unit modeling. They do not suggest they are doing that here.

The authors have done a great job in selecting a globally-significant problem to study. They are also commended for their innovation in wanting to pair joint modeling of geology and redox conditions through MPS methods. In terms of formal review criteria, the scientific significance of this manuscript is 'good' to 'excellent'. The overall presentation is inconsistent, making the presentation quality 'fair'. Some parts are well written and logically argued, but there several important places in the text need more clarity or better explanations. There are a few key places where critical description of methods are missing. The graphics are of good quality and well chosen. However, given the small size and high complexity of the models, the color differences in the geologic models are difficult to interpret. The captions often need to be improved; they need clearer descriptions of what is actually shown in each figure. Importantly, however, I have significant concerns about the scientific quality of key aspects of the study design. Specifically, decisions about the representation of the geology, application of the geologic representation to the MPS modeling approach, and the discussion of the modeling results are all 'fair' to 'poor'. These scientific quality issues are addressed in detail, below. Overall, I believe this manuscript requires significant revisions prior to publication, but that if these concerns are addressed, it clearly merits another review opportunity.

The primary problem I see with the scientific quality in this manuscript is that the geologic deposits are represented as simply a succession of distinct textures (i.e., facies), when the depositional origin, size, distribution, and description, suggest the deposits being modeled are a succession of stratigraphic units, with similar textures, that were deposited through an unspecified number of distinct ice events. MPS would be a clearly suitable approach if the authors were modeling textural distributions within the meltwater sand/gravel and clay/silt assemblages, or modeling the distribution of textural facies or inclusions of sorted sediments within the till deposits. However, in that situation, they would have to model the main deposit boundaries separately (using some other approach) and later insert the textural simulation results into each main deposit. Instead, the authors are trying to apply MPS to a collection of sediments from two or three distinct depositional environments (i.e., subglacial, proglacial fluvial, and maybe proglacial lacustrine), and from 2-4 distinct ice events.

This is more than just a conceptual problem. The MPS algorithms use the training image to guide the location of textures, but every texture has a non-zero probability of occurring anywhere in the model domain. This is not problematic when the facies are all generated within one, single depositional environment. It is problematic when the facies are from multiple depositional environments (and from multiple ice events). Operationally, this matters because under this latter scenario, the realizations can be expected to have sediments from depositional environments and ice events that are randomly out of sequence. Post-glacial sediments can be expected in many subsurface locations. Too much meltwater sediment should be expected within the upper clay till unit. Too much clay and sandy tills would be modeled within the meltwater succession. Since till units are typically associated with distinct ice events, the clay till will be modeled in too many (and maybe occasionally too few) sedimentologic positions to be consistent with the geologic history. This also means that your geologic framework will lead to parameterized groundwater models with predictably-incorrect patterns of hydraulic heterogeneity – one of the main things MPS is trying to avoid. While the distributions of units in the valley-fill (Northern buried valley) setting are a bit more complex, for the same reasons this setting appears to be better described as a succession of stratigraphic units. As with the upland succession, modeling subglacial and proglacial deposits as facies within a single zone is particularly problematic with MPS. In a complex valley fill succession that is composed of multiple erosional/depositional events, it can be very difficult to see the resulting biases – you are expecting complex assemblages and complex assemblages is what you get. The logic of the model doesn't change, however, and the errors for the upland will inevitably be carried over to the valley deposits.

I noted early in my comments that the presentation quality of this manuscript is not ideal. It is possible that I have misunderstood the geologic setting and the approach taken to configure the MPS in this study. However, the description of the geologic deposits does not provide enough clarity to understand the geologic history of the model domain, or the consequent distribution of stratigraphic units and sediment textures. The authors also do not acknowledge this important geologic constraint to the successful application of MPS, nor do they provide discussions on their rationale for using MPS in this setting or on the design targets for texture proportion and zonation. This prevents a clear understanding of what the geologic history is known to be, and how this modeling approach is being used to reliably model the sediment while using that knowledge as a necessary constraint.

My last comment regarding the poor scientific quality addresses the poor quality of sim-

ulation results for the geologic deposits. Based on the figures of the model realizations and from one or two oblique comments from the authors, these results appear to have an unacceptable fit with the data and training images. The authors barely note the quality of fit in their discussion and (except for a reference to unspecified algorithmic artifacts that generate occasional small errors) they do not provide sufficient technical explanation of why the simulation results fit so poorly with the geology training images. More importantly, the authors suggest that they could have made the solutions better fit the training images, but stopped with the simulations so they could present the method as a viable option. This needs to be fixed prior to submitting a methods manuscript for publication. Until an author can demonstrate clearly and objectively that they can reliably meet the stated modeling goals using the proposed method, the method is not ready for publication.

Acknowledging these limitations within the manuscript, the authors are encouraged to fix these issues, remodel the area, and revise the manuscript. If these issues are corrected, this manuscript would be a worthwhile contribution to the literature.

---

## Referee Comment (RC2) · Anonymous Referee #2 · 27 Oct 2020

Overview: This paper presents an approach for joint stochastic simulation of subsurface geological and redox architectures in 3D using multiple point geostatistics (MPS). The method is demonstrated on a small catchment in Denmark, where simulations are conditioned on observed resistivities of the subsurface from towed transient electromagnetic measurements (tTEM) as well as on soil maps and borehole observations of lithology, sediment colours and water chemistry.

The paper is interesting and addresses an important topic which fits the scope of HESS. To predict the transport and fate of nitrate and to improve the understanding and management of nitrate contaminated aquifers, detailed knowledge of subsurface

geology and redox conditions is required. The characterisation of the subsurface is usually largely based on borehole data (sometimes together with other ancillary data, knowledge etc.), which subsequently are interpolated to cover the domain of interest. The heterogeneity of the subsurface means that such interpolations can be associated with large uncertainty. Stochastic geostatistical approaches, like the one used by the authors here, are therefore widely used to account for such uncertainties in subsurface mapping and modelling. I'm no expert in MPS or geologist (this is therefore likely to be reflected in my comments below), but to me, it seems appropriate to use the MPS method for the context here, although other methods exist. There is therefore a lot to like about this paper. However, I have number comments and issues that I think should be addressed before this paper can be published. Below are my comments which I hope the authors will find useful. Held og lykke!

General comments:

1. I'm not very familiar with the MPS method and how it works exactly, so I can't really comment on the chosen parametrisation or even on whether the results produced are good/acceptable. However, I do wonder why the simulation artifacts occur? I think this should be elaborated on. It is stated that the geological artifacts are likely due to inconsistencies between TI and conditioning data (L565), but what does that mean exactly? Why do the realisations deviate from the TI? The non-stationarity of the TI is mentioned but I don't understand what this means. It is suggested that artifacts could potentially be removed through better parametrization, but why would that be, and if so, shouldn't you have attempted to do some parameter 'tuning' as part of this work? I understand that the artifacts are basically being averaged out over multiple realisations, and I think if the main aim was to produce best estimate and associated uncertainty (entropy) of the geology and redox (Figure 11), then the occurrence of artifacts would be less of an issue. However, if the aim is also to provide realisations as input for transport and fate modelling of nitrate, then I would think the artifacts could become much more of an issue, especially as flow and solute transport will depend non-linearly on those real-

isations. It is argued that for hydrological modelling at catchment scale, the geological artifacts will have limited effect, which may be true, but I'm less convinced the same is the case for nitrate transport, especially given the redox artifacts.

2. The joint simulation requires a bivariate training image (TI) for geology and redox to be developed. I understand how the geological TIs were developed, but I find it less clear how the redox TIs were produced, who produced them and how the bivariate/joint nature of the TIs are specified. I think this could be better explained.

3. For the case study, two independent bivariate TIs are developed, one representing the Quaternary sequence and one representing the buried valley. It is assumed that the delineation of these geological elements is known in the domain (i.e. no uncertainty as to whether a voxel is Quaternary or buried valley). But what is this delineation based on and with what certainty can this be done?

4. I can't really work out what the influence and significance of the TIs and the conditioning data are for the results. I think it would be relevant to include a discussion on the value of the information used. Given the large amount of conditioning data, does the TI become less important? Would the results be significantly different if you just used the same TI for both the geological elements? What is the effect of the soft data on the result? It would be interesting to see what the results look like if you excluded these. Except for the surface geology, the soft data looks quite uninformative.

5. It is not clear to me exactly how topography is used to inform soil moisture when deriving soft data for redox conditions. L365 just states how slope/topography affect moisture in general (it sounds like a TOPMODEL type of approach?). I think further explanation is required here.

6. Figures: The 3D figures look impressive, but there is a lot of them, and I think they are quite complex, and the colour choices make them not so easy to 'read' (especially the small ones). I wonder if some of the figures would perhaps be more illustrative in 2D rather than 3D. I don't think Fig 5 works very well as a 3D figure, and in Fig. 6 it

is difficult to see that tTEM actually only covers part of the simulation grid. In Figure 1a it took me a while before I realised that the blue, black and red lines where actually the TEM and ERT data points. In Figure 7 the colours are difficult to distinguish in the histograms and it looks like there is a shade of blue in Fig 7a that is not in the legend.

7. Overall, I think the paper is well-written, but there are places where I find the presentation of the material unclear and confusing and where more explanation would be helpful (particularly section 6 and 7). I appreciate that this is not helped by the fact that I'm not so familiar with the MPS method.

Minor specific comments:

Abstract: I struggled to understand L16-22 when reading the abstract the first time around (and the similar paragraph in the introduction, L106-110). It makes more sense to me after reading the paper, but I would encourage the authors to sharpen the text here.

Figure 1c: Are the white areas unknown soils?

L167: Rephrase. It sounds like you are doing hydrological simulations as part of this work, which you are not.

L169: There are 7 classes Table 1.

L199: Not sure I understand why they provide independent measurements of redox.

L255: consists of. . .

L306: From figure 1 It looks like the data availability is better in the southernmost part of the domain?

L310-311: I do not follow this. Why is sandy till not included in the TI? How does that affect simulations for the buried valley and can simulations here then meaningfully be conditioned to observations of sandy till?

L313: I'm not sure I understand what you mean by translating the geological TI to a redox TI by integrating geochemical and water chemistry data. Is the redox TI mainly based on the geological TI, e.g. redox conditions are generally reducing in (upper) areas with meltwater sand, oxic at surface near sediments etc.? And then just making some adjustments based on measured data? That's almost how it looks from Figure 4.

L321: I don't understand how its eases the construction of the TI to include parts outside the domain. Please clarify.

P16 and Fig 7: I find it difficult to follow the text and observations that go with Fig 7. It is stated that sandy till is associated with some of the highest resistivities, but I can't see this from Fig 7. I don't follow how the general relationship /distribution of resistivities have been derived (Fig 7c). I think this whole section could be clearer.

Equation 2: I'm not sure I follow the equation. Is the soil map (surface geology data) used to inform the lithology below the surface layer as well in a similar way as in the buffer zones as described in section 5.4.2? Or what is the probability of surface geology below the surface layer (intuitively I would think zero probability in which case the equation collapses)?

L435: ...overlapping relationship...

Soft data: do you derive soft probabilities for geology for all voxels in the simulation grid, even where hard data are present? It looks so from Fig 8 and 9.

L457: To what extent did you experience "flagging" as part of the simulations you generated?

L485-500: I found some of the described observations here difficult to see in Figure 10. Maybe it would be easier to follow if there was a close-up on a relevant part of the transects instead?

L555: I'm not sure I understand why and when your simulations would not honour TI and data.

L574-585: I think the text here reads so well and it could be improved. This is also where terms like stationary TI and rotation in simulation are mentioned, which I find difficult to really follow not having a MPS background.
* * *

---

## Author Comment (AC1) · 5 Nov 2020

We would like to thank Donald Keefer for his referee comments. In the following, citations from the interactive comment are written with RC1 in front of them. From reading the comments, we have not been able to describe and state the work frame and purpose of the project thoroughly. We think that a lot of the comments are based on a misconception of the modeling purpose and detail. Before commenting, we therefore start by stating the purpose of the modeling.

The work we present is part of a project (The Mapfield project) that aims at developing a concept for targeted N regulation. This concept should be consistent with various

data sources available but should also be cost-effective such that it might be suited for commercial use afterwards. The evaluation of nitrate vulnerability should be based on a standardized workflow and should still be able to be customized to the regional challenges in specific areas.

The modeling purpose of the current study is to create subsurface 3D models of geological architecture and redox architecture to be used in hydrological modelling. Via particle tracking, the reduction capacity for the subsurface can be assessed taking some of the 3D architectural heterogeneities into account. The modeling scale is partly determined by the overall purpose of the project and partly by the typical resolution of EM data. We are fully aware that it is not a perfect description of the subsurface, but reasonable within the scale of investigation. An obvious advantage of using MPS is that the realizations are consistent with the information and inputs selected. If there is some variability that is not geologically correct in the realizations it is not necessarily an issue of MPS modeling but means that this variability is not present in the training data or conditional data. Irrespective of this, we still claim that it is better to integrate this information than not to.

This work is an important step towards solving the aim of the Mapfield project. Additionally, this work is not only of importance in the specific problem of targeted N regulations, but any water quality issues under heterogeneous settings.

Generally, we have clearly not been able to transfer this message in the manuscript and apologize for this. We will work on improving the introduction to the modeling and our justification for using MPS specifically to model spatial patterns in the subsurface.

RC1: "This manuscript is included for consideration of a special issue: Frontiers in the application of Bayesian approaches in water quality modeling. It is important to note that the research described in this manuscript does not use Bayesian methods, and so does not directly relate to the topic of the special issue. "

The manuscript is not considered for the special issue anymore.

RC1: "Leaching of agriculturally-applied nitrogen is a significant threat to global groundwater resources. Success in reducing nitrate contamination in groundwaters by modification of farming practices and reductions in fertilizer input has been only occasionally realized and more work is definitely needed. Prediction of the fate and transport of fertilizer-based nitrate is a complicated problem that is confounding successful remediations to the groundwater contamination problem. Modeling success is significantly limited by the natural heterogeneities and complexities of the subsurface geologic systems and by our limitations in characterizing and modeling those complexities. The complex nature of the heterogeneities has led to the treatment of these systems as stochastic which has subsequently led to a large interest in probabilistic modeling approaches."

We completely agree with the reviewer that not only is pollution of groundwater resources from surface leaching a significant problem but also that this is a complex problem to solve/model. Especially, given heterogeneities in the subsurface. Regarding the last line. Do you mean that these systems are modeled stochastically in our case specifically or in general? To our knowledge most studies focusing on mapping heterogeneities in geology and redox conditions are not treated stochastically. If you have knowledge of references where redox conditions and geology as a combined system are treated stochastically, we would be glad to know.

RC1: "Multiple Point Statistics (MPS) has proven to be an innovative and successful probabilistic approach to modeling inter-facies relationships, primarily within distinct, lithostratigraphic units. As typically used, MPS is not a method that can be used with stratigraphic units or with units that are genetically distinct (e.g., from subglacial environments and glaciofluvial environments), unless each genetically-distinct succession is modeled within separate portions of the model domain. To my knowledge, a modified MPS approach has not been successfully applied to modeling of larger stratigraphic assemblages of rocks or sediments. It would be particularly relevant and innovative if the authors were to present a method for adapting MPS to stratigraphic unit modeling.

They do not suggest they are doing that here. "

We would like to argue that MPS has not only been successfully applied to model inter-facies relationships. Since the inception of MPS the key feature of the method is the ability to model spatial continuity over longer distances. Mariethoz and Caers (2015) describe their comprehensive book on the application of MPS as: "This book is therefore a book about spatial and spatiotemporal modeling in the physical sciences (sedimentology, mineralogy, climate, environment, etc.)". In other words, MPS is not scale or problem dependent. Tahmasebi (2018) also showcase different applications of MPS in his state-of-the-art review of the method. Several studies have shown the applicability of MPS for modeling the geological architecture of major lithological units (Jørgensen et al. 2015; Barfod et al. 2018; Vilhelmsen et al. 2019), although we agree that the application of MPS for real world scenarios is not overwhelming. Høyer et al. (2017) are simulating the architecture of a thick succession of Miocene sand and clay units within a confined area of the subsurface. In accordance with this approach the geological elements in the current study confines a volume in which a TI represent the different deposition histories and the expected differences in spatial variability.. Instead of simulating merely sand and clay, in the current study we use 5 lithological groups thereby extending the concept proposed in Høyer et al. (2017). The reason we choose major lithological groups to model in the current work is based on the vertical resolution of both geophysical data and well data. The data does not allow simulation on the scale within lithostratigraphic units. There are several aspects to this.

1) the current discretization of the simulation grid (25m x 25m x 2m) is not able to accommodate internal sediment structures in the specific lithostratigraphic units.

2) It would require an additional MPS simulation step – a sequential process – where the results would predominantly rely on a TI and perhaps a few wells that contain information about grainsize in the lithological description to enable a distinction between fine-grained and coarse-grained material.

3) a more detailed grid than 25m x 25m x 2m would make it difficult to handle subsequent flow and transport modelling and would ultimately have to be re-gridded to coarser grid to a grid approximating the current. The current grid was chosen in collaboration with a hydrologist to determine the smallest discretization that would still allow computationally feasible hydrological modeling at catchment scale

4) We have doubts that this extra complexity of the problem of adding internal structures would have a significant impact on flow and transport models.

RC1: "The authors have done a great job in selecting a globally-significant problem to study. They are also commended for their innovation in wanting to pair joint modeling of geology and redox conditions through MPS methods. In terms of formal review criteria, the scientific significance of this manuscript is 'good' to 'excellent'. The overall presentation is inconsistent, making the presentation quality 'fair'. Some parts are well written and logically argued, but there several important places in the text need more clarity or better explanations. There are a few key places where critical description of methods are missing. "

We very much appreciate the overall positive feedback on our choice of subject and our presentation of the problem at hand. We fully agree that there is an urgent need for methods to enable N-targeted regulations for protection of groundwater resources in countries with extensive agriculture. We intend to review the manuscript carefully for places in lack of clarity and look at places that might need clarification on method. We encourage the reviewer to participate in this process.

RC1: "The graphics are of good quality and well chosen. However, given the small size and high complexity of the models, the color differences in the geologic models are difficult to interpret. The captions often need to be improved; they need clearer descriptions of what is actually shown in each figure. "

Thank you. We will work on improving the colormap and captions of the figures.

RC1: "Importantly, however, I have significant concerns about the scientific quality of key aspects of the study design. Specifically, decisions about the representation of the geology, application of the geologic representation to the MPS modeling approach, and the discussion of the modeling results are all 'fair' to 'poor'. These scientific quality issues are addressed in detail, below. Overall, I believe this manuscript requires significant revisions prior to publication, but that if these concerns are addressed, it clearly merits another review opportunity. The primary problem I see with the scientific quality in this manuscript is that the geologic deposits are represented as simply a succession of distinct textures (i.e., facies), when the depositional origin, size, distribution, and description, suggest the deposits being modeled are a succession of stratigraphic units, with similar textures, that were deposited through an unspecified number of distinct ice events. "

We refer to our modeling purpose stated in the beginning and our responses above.

RC1: "MPS would be a clearly suitable approach if the authors were modeling textural distributions within the meltwater sand/gravel and clay/silt assemblages, or modeling the distribution of textural facies or inclusions of sorted sediments within the till deposits. However, in that situation, they would have to model the main deposit boundaries separately (using some other approach) and later insert the textural simulation results into each main deposit. Instead, the authors are trying to apply MPS to a collection of sediments from two or three distinct depositional environments (i.e., subglacial, proglacial fluvial, and maybe proglacial lacustrine), and from 2-4 distinct ice events. "

We recognize the issue described. Rarely, however is data collected or analyzes made (e.g., grain analyzes) that would provide sufficient knowledge for such a detailed mapping. Especially not at the scale of the catchment area presented. We have therefore decided to split the area into larger geological elements that consists of multiple depositional events. We also refer back to our previous comment on the data resolution.

If there are some geological rules that would fit better with the modeling area the whole

idea is that this information can be quantified and added to the model. It would also be possible to combine the current realizations with these explicit rules/information as a sort of inverse problem. Other options include coming up with extra probability grids that could help guide the results. The question is where to obtain the information for these. It is beyond the scope of the current paper but is worth exploring in future studies. We hope that Donald Keefer and others appreciate that this work is a step towards better and more consistent models that integrate information from several sources.

RC1: "This is more than just a conceptual problem. The MPS algorithms use the training image to guide the location of textures, but every texture has a non-zero probability of occurring anywhere in the model domain. This is not problematic when the facies are all generated within one, single depositional environment. It is problematic when the facies are from multiple depositional environments (and from multiple ice events). "

MPS uses training images as analogs of the spatial variability to generate realizations of possible models of the subsurface. The conditional data (wells, geophysics) guide the location of the architecture. Once again, we have probably not been clear enough in delivering the modeling aim and the chosen method. This is unfortunate.

RC1: "Operationally, this matters because under this latter scenario, the realizations can be expected to have sediments from depositional environments and ice events that are randomly out of sequence. Post-glacial sediments can be expected in many subsurface locations. Too much meltwater sediment should be expected within the upper clay till unit. Too much clay and sandy tills would be modeled within the meltwater succession. Since till units are typically associated with distinct ice events, the clay till will be modeled in too many (and maybe occasionally too few) sedimentologic positions to be consistent with the geologic history. "

The current modeling is focused on the architecture of lithology and hence does not account for geological history in the sense of multiple ice events and depositional events. The geological history and processes need to be conveyed (quantified) through the

training image. In our case the subdivision of the simulation grid into geological elements also provide a way of conveying geological knowledge into the system. This modeling is not the conclusive description of the subsurface but fits the purpose of the modeling. The hard and soft data available will help guide the sand and moraine clay layers with realistic distributions in the realizations.

RC1: "This also means that your geologic framework will lead to parameterized groundwater models with predictably-incorrect patterns of hydraulic heterogeneity – one of the main things MPS is trying to avoid. "

In which sense is MPS trying to avoid incorrect patterns of hydraulic heterogeneity? The realizations are consistent with the patterns provided in the training image. Then the problem is rather whether the training image is conveying the expected kind of information about the spatial variability. MPS is not trying to do anything except it helps solve the problem of lacking connectivity of features encountered in classical 2-point statistics.

RC1: "While the distributions of units in the valley-fill (Northern buried valley) setting are a bit more complex, for the same reasons this setting appears to be better described as a succession of stratigraphic units. As with the upland succession, modeling subglacial and proglacial deposits as facies within a single zone is particularly problematic with MPS. In a complex valley fill succession that is composed of multiple erosional/depositional events, it can be very difficult to see the resulting biases – you are expecting complex assemblages and complex assemblages is what you get. The logic of the model doesn't change, however, and the errors for the upland will inevitably be carried over to the valley deposits. "

The delineation of geological elements is carried out being fully aware that each element can compose layer successions from several depositional events. Cognitive interpretation and modelling of specific stratigraphical units it not feasible or to strive for within the project frame, instead our key line out is to: a) Interpret independent

and important geological elements/successions with each study area, b) within each element 3D training images are set up with the purpose of representing the geological trends of the structural architecture.

We argue that MPS can be used as an important tool in the context of applying dense EM datasets, where the geological elements and the training images are valuable input to guide the realizations of subsurface structures characterized by the main site-specific lithology classes.

RC1: "I noted early in my comments that the presentation quality of this manuscript is not ideal. It is possible that I have misunderstood the geologic setting and the approach taken to configure the MPS in this study. However, the description of the geologic deposits does not provide enough clarity to understand the geologic history of the model domain, or the consequent distribution of stratigraphic units and sediment textures. The authors also do not acknowledge this important geologic constraint to the successful application of MPS, nor do they provide discussions on their rationale for using MPS in this setting or on the design targets for texture proportion and zonation. This prevents a clear understanding of what the geologic history is known to be, and how this modeling approach is being used to reliably model the sediment while using that knowledge as a necessary constraint. "

Here is probably the crux of our disagreement. As mentioned earlier, we have simply not been able to convey our modeling objectives clearly throughout the manuscript. We do not seek to provide a full description of the geological history of the area. This is not the aim of the paper. Our scientific contribution is to jointly simulate and determine the distribution of redox, geology (here lithological groups). We agree that our presentation of the geological modeling choices have not been sufficient and that we should provide more justification for the way the training images have been constructed. Herein lies most of the geological modeling choices.

RC1: "My last comment regarding the poor scientific quality addresses the poor quality

of simulation results for the geologic deposits. Based on the figures of the model realizations and from one or two oblique comments from the authors, these results appear to have an unacceptable fit with the data and training images. The authors barely note the quality of fit in their discussion and (except for a reference to unspecified algorithmic artifacts that generate occasional small errors) they do not provide sufficient technical explanation of why the simulation results fit so poorly with the geology training images. "

We are not quite sure what is meant here? MPS precisely creates a "fit with the data and training images". It is supposed to create conditional simulation based on the training images and the selected parameterization (prior information). Realization are forced to abide the conditional data which ensures that models are consistent with both data and prior information.

RC1: "More importantly, the authors suggest that they could have made the solutions better fit the training images, but stopped with the simulations so they could present the method as a viable option. This needs to be fixed prior to submitting a methods manuscript for publication. Until an author can demonstrate clearly and objectively that they can reliably meet the stated modeling goals using the proposed method, the method is not ready for publication. Acknowledging these limitations within the manuscript, the authors are encouraged to fix these issues, remodel the area, and revise the manuscript. If these issues are corrected, this manuscript would be a worthwhile contribution to the literature. "

The solutions (realizations) are not supposed to fit the training image. Conditional simulation (of any geostatistical simulation methods) generates realizations consistent with the assumed geostatistical model (here in form of the TI and choice of simulation algorithm) and the conditional data. Apparent 'artifacts' simply illustrate that (over longer distances) it is difficult to integrate the two types of information, as they provide different types of information. We have also adapted postprocessing procedures to remedy most of these artifacts.

This is less of a problem using 2-point based Gaussian simulation, in which one will have less of these 'inconsistencies' due to the high entropy of a Gaussian model. In general, MPS is based on a limited sized training image and based on higher entropy models. This combination suggests that, in general, one rarely have perfect consistency between the training image and the conditional data. (the conditional data does not appear as is in the training image). This should be noted, as we do in the text, but we do not think an analysis of this challenge should be part of the present manuscript but would warrant a separate independent work.

The solution should be consistent with the data, which is the case. We can try to come up with a way to quantify this statement to demonstrate it more objectively. We specifically wrote the following: "As pointed out by Tahmasebi (2018), a quantitative evaluation of the performance of MPS is still unresolved, and as previously mentioned, the parameterization could potentially be fine-tuned to produce even better results." What is meant here is that different parameterization of the problem provides different solutions (different variability given the same training image) and it is still difficult to find a universal measure for determining whether the current parameterization provides a better fit or not. We have chosen to avoid this discussion in detail since it could easily become a paper on its own as there is much to explore once you open this discussion. The topic of choosing a suitable parameterization of the sampling algorithm has drawn increased attention in the MPS community within the last years. Exemplified e.g., by the new study by Juda et al. (2020). Basically, there are three main targets for an algorithm parameterization. 1) Realizations should fit the data as good as possible 2) The realizations should convey the underlying patterns and structures in the training image and 3) The realizations should be feasible to compute. This is usually a trade-off situation. Here we focused primarily on making sure that realizations are consistent with our expectation of the variability: "However, we adapt the heuristic strategy of Høyer et al., (2017), making sure that the realizations are in accordance with our expectations and focus on presenting the methodology of simulating both geology and redox simultaneously in 3D high-resolution."

Kind Regards

Rasmus and co-authors

References

Barfod AS, Møller I, Christiansen AV, et al (2018) Hydrostratigraphic modeling using multiple-point statistics and airborne transient electromagnetic methods. Hydrol Earth Syst Sci 22:3351–3373. https://doi.org/10.5194/hess-22-3351-2018

Høyer A-S, Vignoli G, Hansen TM, Vu, Le Thanh, Keefer, Donald A., Jørgensen, Flemming (2017) Multiple-point statistical simulation for hydrogeological models: 3-D training image development and conditioning strategies. Hydrol Earth Syst Sci 21:6069–6089. https://doi.org/10.5194/hess-21-6069-2017

Jørgensen F, Høyer A-S, Sandersen PBE, et al (2015) Combining 3D geological modelling techniques to address variations in geology, data type and density - An example from Southern Denmark. Comput Geosci 81:53–63. https://doi.org/10.1016/j.cageo.2015.04.010

Juda P, Renard P, Straubhaar J (2020) A Framework for the Cross‐-Validation of Categorical Geostatistical Simulations. Earth Sp Sci 7:1–17. https://doi.org/10.1029/2020EA001152

Mariethoz G, Caers J (2015) Multiple-point geostatistics: Stochastic modeling with training images, 1st edn. John Wiley & Sons

Tahmasebi P (2018) Multiple Point Statistics: A Review. In: Handbook of Mathematical Geosciences. Springer International Publishing, Cham, pp 613–643

Vilhelmsen TN, Auken E, Christiansen AV, et al (2019) Combining Clustering Methods With MPS to Estimate Structural Uncertainty for Hydrological Models. Front Earth Sci 7:1–15. https://doi.org/10.3389/feart.2019.00181

---

## Author Comment (AC2) · 5 Nov 2020

We would like to thank the anonymous reviewer for his/her referee comments. In the following citations from the interactive comment is written with RC2 in front of them.

RC2: "Overview: This paper presents an approach for joint stochastic simulation of subsurface geological and redox architectures in 3D using multiple point geostatistics (MPS). The method is demonstrated on a small catchment in Denmark, where simulations are conditioned on observed resistivities of the subsurface from towed transient electromagnetic measurements (tTEM) as well as on soil maps and borehole observations of lithology, sediment colours and water chemistry. The paper is interesting

and addresses an important topic which fits the scope of HESS. To predict the transport and fate of nitrate and to improve the understanding and management of nitrate contaminated aquifers, detailed knowledge of subsurface geology and redox conditions is required. The characterisation of the subsurface is usually largely based on borehole data (sometimes together with other ancillary data, knowledge etc.), which subsequently are interpolated to cover the domain of interest. The heterogeneity of the subsurface means that such interpolations can be associated with large uncertainty. Stochastic geostatistical approaches, like the one used by the authors here, are therefore widely used to account for such uncertainties in subsurface mapping and modelling. I'm no expert in MPS or geologist (this is therefore likely to be reflected in my comments below), but to me, it seems appropriate to use the MPS method for the context here, although other methods exist. There is therefore a lot to like about this paper. However, I have number comments and issues that I think should be addressed before this paper can be published. Below are my comments which I hope the authors will find useful. Held og lykke!"

Thank you ("mange tak") for the fine review of the manuscript and the modeling objectives. We have found all the general comments provided much useful and will be answered in the following. We will specify what changes this bring about in the coming version of the manuscript.

RC2: "General comments: 1. I'm not very familiar with the MPS method and how it works exactly, so I can't really comment on the chosen parametrisation or even on whether the results produced are good/acceptable. However, I do wonder why the simulation artifacts occur? I think this should be elaborated on. It is stated that the geological artifacts are likely due to inconsistencies between TI and conditioning data (L565), but what does that mean exactly?"

The main idea behind MPS is that spatial variability is described through a training image. For most MPS algorithms a realization is generated sequentially one voxel at a time. Consider standing at some location in the simulation grid and wanting to know

what value to realize in this position. You then gather configuration of the surrounding already simulated nodes (conditional event). Specifically, in direct sampling, as used here, you then search the training image for a matching event. If such an event exist it is placed in the simulation grid and you can move to the next location and so forth until the entire simulation domain is filled.

Artifacts in the simulations might occur if it difficult to find a matching event in the TI. This is what is meant by line 565. If at some point it is impossible to find a suitable candidate, the next best thing must be selected. Sometimes this process leads to simulation artifacts. There are several ways this might be remedied but it all comes at a price. In all cases you are trying to broaden the number of possible matches in the training image (i.e., making sure the prior information is not in opposition to the conditioning data)

A simple way is to reduce the number of already simulated nodes to include in the conditional events. With smaller conditional events, more locations in the TI will be considered reasonable matches. This affects the ability to correctly represent the variability in the long-range structures from the TI. In general, it will lead to increased variability that is not necessarily related to geological variability as portrayed in the TI. Also, when simulating using fewer conditional data, the point being simulated might end up leading to inconsistencies for points to be simulated later.

These problems do not arise in 2-point based Gaussian simulation, because one can always compute the conditional distribution, while in MPS one must infer the conditional from a finite sized example (the TI) represent an outcome of the assumed underlying probability distribution.

An alternative could be to increase the size of TI and hence create additional possibilities for matching events in the TI. However, finding/generating a suitable TI is already the biggest challenge and strength of MPS simulation. The larger TI should still convey the same geology as before but with increased variability, which is not trivial.

We hope this sufficiently answers your question. We will try to rewrite the section to emphasize what exactly constitutes a simulation artifact.

RC2: "Why do the realisations deviate from the TI? "

Following from the above we hope it is clear that realization will always deviate from the TI. It should not be the same. The only place we might expect to have a huge overlap between the TI and the realizations is the area from which the TI is derived. Since the TIs are also in the simulation domain the correspondence between the two might be something to include in the revised manuscript.

The TI represent an outcome of the probability distribution we wish to sample from. The goal is to simulate new realizations that has the same multipoint statistics, but not the actual structures themselves.

RC2: "The non-stationarity of the TI is mentioned but I don't understand what this means."

The MPS simulation algorithm is expecting the same spatial variation in the whole training image. A training image that is stationary provides exactly that. For instance, postglacial sediments are only found at the top of TIs in the current study. When the algorithm searches for matching events containing postglacial sediment, matches can then only be found in some specific location. To be more technical, for a process to be stationary, all its statistical properties must be independent of spatial location. This is clearly not the case for a TI built on geology and redox for that matter. Both are extremely non-stationary by nature. Different processes shape the subsurface and it is not providing the same statistical properties at all locations. The issues of non-stationarity in MPS are discussed in detail in Mirowski et al. (2009). Another point is that issues with non-stationarity can be remedied with large amounts of conditional data (see e.g., Barford et al. (2018) or Vilhelmsen et al. (2019)).

Should we perhaps elaborate this within the manuscript? We would argue that for most

(geo)statisticians the term is well known. So, this will mostly be of value for people unaware of what exactly non-stationarity is. Another problem is the general length of the manuscript as we try to combine the work of several disciplines (geochemistry, geology, geophysics, geostatistics) we have chosen some areas to focus on while other are left for the reader to delve into.

RC2: "It is suggested that artifacts could potentially be removed through better parametrization, but why would that be, and if so, shouldn't you have attempted to do some parameter 'tuning' as part of this work? "

From our answers above we hope to lift the lid on why parameterization of MPS simulation algorithms is not straightforward. Parameterizing the algorithm depends on 1) the conditional data events available in the TI (its size, variability and stationarity), 2) computational demands and 3) ultimately if the realizations satisfy the variability that is expected within the training image. We focused more on later part from a qualitative perspective but can also include some quantitative measures also. By going too much into detail with parameter optimization we are afraid to divert the attention away from the core message of the paper.

RC2: "I understand that the artifacts are basically being averaged out over multiple realisations, and I think if the main aim was to produce best estimate and associated uncertainty (entropy) of the geology and redox (Figure 11), then the occurrence of artifacts would be less of an issue. However, if the aim is also to provide realisations as input for transport and fate modelling of nitrate, then I would think the artifacts could become much more of an issue, especially as flow and solute transport will depend non-linearly on those realisations. It is argued that for hydrological modelling at catchment scale, the geological artifacts will have limited effect, which may be true, but I'm less convinced the same is the case for nitrate transport, especially given the redox artifacts."

You are completely right here. This will remain an assumption on our part. We do not

know to which extent these artifacts affect subsequent hydrological models. The logical next step and what we are currently working on is to explore how these MPS models of the subsurface architecture affect hydrological models. If these redox artifacts are significant, it is possible and trivial within the current workflow to e.g., enforce that below a certain depth only reduced redox conditions would occur.

RC2: "2. The joint simulation requires a bivariate training image (TI) for geology and redox to be developed. I understand how the geological TIs were developed, but I find it less clear how the redox TIs were produced, who produced them and how the bivariate/joint nature of the TIs are specified. I think this could be better explained."

In the current study, the geological training images were developed primarily by a geologist and the redox training images by a geochemist. We agree that the description of the work carried out in constructing the redox training image is weakly documented. We will try to describe the process and choices made in more detail in the revised manuscript.

RC2: "3. For the case study, two independent bivariate TIs are developed, one representing the Quaternary sequence and one representing the buried valley. It is assumed that the delineation of these geological elements is known in the domain (i.e. no uncertainty as to whether a voxel is Quaternary or buried valley). But what is this delineation based on and with what certainty can this be done?"

The delineation is based on geological interpretation. At the ground surface this interpretation is primarily guided by topographic information combined with the surface geology maps. At depth borehole information and geophysical data (tTEM, SkyTEM etc.) provides spatial information about the boundaries between the geological elements. Knowing the geological processes and the depositional setting from neighboring areas aids this delineation as well. You raise an interesting question regarding the lack of uncertainty for this boundary. We have discussed this internally within the author group. Clearly there is some uncertainty attributed to the location of these boundaries.

The main question is how one can quantify this uncertainty. Also, one must find a way of incorporating these uncertainties in the simulation results. Finally, the effect on the realizations of having such an uncertain boundary must be assessed in comparison to a static boundary. To answer these questions in depth is beyond the scope of the current study. One idea that we would like to follow is to attribute the interpreted boundaries with a quantitative uncertainty based on the interpreter's qualitative uncertainty. By doing so we could simulate several boundary configurations prior to modeling. A set of simulation grids could then be produced that for which we generate one realization. The resulting set of realizations would then satisfy the independence between the geological elements and have a quantitative representation of the uncertainty of this boundary. Alternatively, there is also the possibility to, in direct sampling, simulate with an auxiliary probability grid covering the simulation grid telling which TI to use depending on spatial location. However, this would break the independence between the geological elements as conditional voxels from the buried valley would be included in conditional events for the Quaternary sequence.

RC2: "4. I can't really work out what the influence and significance of the TIs and the conditioning data are for the results. I think it would be relevant to include a discussion on the value of the information used. Given the large amount of conditioning data, does the TI become less important? Would the results be significantly different if you just used the same TI for both the geological elements? What is the effect of the soft data on the result? It would be interesting to see what the results look like if you excluded these. Except for the surface geology, the soft data looks quite uninformative."

Good point. It not straightforward to assess the value of different types of information. Also, because MPS tends to underestimate the importance of soft data compared to hard data. Clearly when looking at the entropy of the final models compared with the soft data, the TI provides a lot of information to the system. We discussed to include prior realizations (no conditioning data) to show the value of adding information to the system. Maybe the TI provides too much information in the current setup. We

will include a section discussing the value of information for the simulations. It is an important topic.

RC2: "5. It is not clear to me exactly how topography is used to inform soil moisture when deriving soft data for redox conditions. L365 just states how slope/topography affect moisture in general (it sounds like a TOPMODEL type of approach?). I think further explanation is required here. "

The slope is used as a surrogate for moisture. Not to inform soil moisture content. At the Quaternary sequence, we have wells that can assist in translating the surface geology map into probabilities for redox conditions. This is done simply by counting the occurrences of different redox conditions associated with the surface geology conditions. There are too few redox wells available in the buried valley to perform a similar translation. One solution would be to use the probabilities derived for the Quaternary sequence directly in the buried valley. However, we know that topography is coupled to soil moisture content. Since the slope of the buried valley is much flatter, we assume that soil moisture is different in this element and hence scale the probabilities obtained in the Quaternary sequence such that the buried valley has a lower probability of oxic conditions at the surface. This is our justification for changing the probabilities between the two elements.

RC2: "6. Figures: The 3D figures look impressive, but there is a lot of them, and I think they are quite complex, and the colour choices make them not so easy to 'read' (especially the small ones). I wonder if some of the figures would perhaps be more illustrative in 2D rather than 3D. I don't think Fig 5 works very well as a 3D figure, and in Fig. 6 it is difficult to see that tTEM actually only covers part of the simulation grid. In Figure 1a it took me a while before I realised that the blue, black and red lines where actually the TEM and ERT data points. In Figure 7 the colours are difficult to distinguish in the histograms and it looks like there is a shade of blue in Fig 7a that is not in the legend."

The other reviewer also had some remarks to the colormap. We intend to change the colormap to have more distinctive colors. We understand your point regarding figure 5. Conversely, we argue that for consistency between the figures it is better to keep figure 5 as an oblique 3D figure. In order to help better show the extent of the simulation grid, we intend to add an opaque version of the simulation grid to figure 6 similar to what is done in figure 5. The problems of figure 7 should be solved when another colormap is chosen. The apparent different shade of blue is probably due to transparency in the histogram. The colors should be identical. In figure 1a we intend to change the legend to better describe the geophysical data.

RC2: "7. Overall, I think the paper is well-written, but there are places where I find the presentation of the material unclear and confusing and where more explanation would be helpful (particularly section 6 and 7). I appreciate that this is not helped by the fact that I'm not so familiar with the MPS method."

Thank you very much. We will go through section 6 and 7 and try to rewrite them. As many readers of HESS would probably be unfamiliar with MPS we will try to elaborate on some of the concepts that might be confusing from an outsider.

Summary of intended changes for new manuscript version based on the above major comments:

- Calculate correspondence between TI and realizations in places where the TI corresponds is situated within the training images - Maybe introduce quantitative measures for realization performance if it does not divert the core message of the paper. - Better description of the process behind both geology and redox TI construction. Focusing on the specific choices during the process. - Include a section on the value of different types of information for the geostatistical simulations - Change colormap for geological realizations. - Revisit section 6 and 7 and rewrite passages that might requires too much background knowledge on MPS.

We intend not to elaborate on the minor comments in the discussion phase but will take

these into account in the revised version of the manuscript.

On behalf of myself and the co-authors

Rasmus Bødker Madsen

References

Mirowski, P.W., Tetzlaff, D.M., Davies, R.C. et al. (2009) Stationarity Scores on Training Images for Multipoint Geostatistics. Math Geosci 41, 447–474 . https://doi.org/10.1007/s11004-008-9194-0

Barfod AS, Møller I, Christiansen AV, et al (2018) Hydrostratigraphic modeling using multiple-point statistics and airborne transient electromagnetic methods. Hydrol Earth Syst Sci 22:3351–3373. https://doi.org/10.5194/hess-22-3351-2018

Vilhelmsen TN, Auken E, Christiansen AV, et al (2019) Combining Clustering Methods With MPS to Estimate Structural Uncertainty for Hydrological Models. Front Earth Sci 7:1–15. https://doi.org/10.3389/feart.2019.00181

———————————————

---

## Author Response (AR1)

**Author's response**

Date: 05/02-2021

Dear editor

I am writing this cover letter to present our resubmission of the manuscript: hess-2020-444. We have received some very constructive and helpful comments that have greatly improved on the old manuscript. Please do not hesitate to give our compliments in this regard.

Based on our answers in the initial discussion and the reviewer's comments we have updated the manuscript. Since the initial submission more wells containing geochemical information (from Mapfield) have been analyzed and added to the pre-existing pool of data. This has meant that the TIs for redox have been remade and the MPS modelling consequently has been redone. This was also done based on the suggestions from reviewer 1. We have also addressed the underrepresentation of the soft data in the final realizations by converting a small fraction into hard data and the issue of inconsistencies between prior and conditional data. We hope that these major changes address the issues raised by the reviewers to a satisfactory degree.

First, we respond to the two reviewers point by point, following on from our initial response in the discussion phase. Then we present a summary list of the major changes to the manuscript, which is followed by the changes made to the figures and tables.

We enclose a version of the manuscript with track changes to help showcase where changes are made.

All authors have approved the submission of the enclosed manuscript.

Kind Regards
Ph.D. Rasmus Bødker Madsen
Geological Survey of Greenland and Denmark
C.F. Møllers allé 8
8000 Aarhus C, Denmark

**Response to specific questions related to reviewers' comments**

**Reviewer 1:**

*The graphics are of good quality and well chosen. However, given the small size and high complexity of the models, the color differences in the geologic models are difficult to interpret. The captions often need to be improved; they need clearer descriptions of what is actually shown in each figure.*

We have made a new colormap more in line with color choices usually used to represent geological models. We have tried to ensure that color differences are clear in the new color map. We have also worked on improving the captions and legends. For instance, for 3D plots the legend now shows voxels instead of only colors. Furthermore, we have added arrows in the figures pointing towards relevant parts of the simulation results that are mentioned in the text. The viewing angle is also changed as to give a better overview of the models.

*This is more than just a conceptual problem. The MPS algorithms use the training image to guide the location of textures, but every texture has a non-zero probability of occurring anywhere in the model domain. This is not problematic when the facies are all generated within one, single depositional environment. It is problematic when the facies are from multiple depositional environments (and from multiple ice events).*

MPS uses training images as analogs of the spatial variability to generate realizations of possible models of the subsurface. The conditional data (wells, geophysics) guide the location of the architecture. We have not been clear enough in delivering the modeling aim and the chosen method in the first version of the manuscript. In the revised version we have tried to emphasize this and our modeling aim/contribution to science in the introduction.

*I noted early in my comments that the presentation quality of this manuscript is not ideal. It is possible that I have misunderstood the geologic setting and the approach taken to configure the MPS in this study. However, the description of the geologic deposits does not provide enough clarity to understand the geologic history of the model domain, or the consequent distribution of stratigraphic units and sediment textures. The authors also do not acknowledge this important geologic constraint to the successful application of MPS, nor do they provide discussions on their rationale for using MPS in this setting or on the design targets for texture proportion and zonation. This prevents a clear understanding of what the geologic history is known to be, and how this modeling approach is being used to reliably model the sediment while using that knowledge as a necessary constraint.*

The geological elements are supposed to capture major depositional trends in the study area. Within each geological element we convey the spatial variability through a TI. These TIs in essence contain the geological choices (and rules) for the given domain. We have altered the section which describe the training images, which should hopefully clarify this point further in the revised version.

*More importantly, the authors suggest that they could have made the solutions better fit the training images, but stopped with the simulations so they could present the method as a viable option. This needs to be fixed prior to submitting a methods manuscript for publication. Until an author can demonstrate clearly and objectively that they can reliably meet the stated modeling goals using the proposed method, the method is not ready for publication. Acknowledging these limitations within the manuscript, the authors are encouraged to fix these issues, remodel the area, and revise the manuscript. If these issues are corrected, this manuscript would be a worthwhile contribution to the literature.*

We have remodeled the area, focusing on creating a better fit between training images and conditional data. We also included a section discussing our prior model selection and our approach for parameterization of the MPS algorithm. We have also solved the problem of soft data representation by

converting a smaller fraction of the data into hard data and performed testing of different parameterization choices and whether rotation is effective in remedying inconsistencies between prior model and data, which is mentioned in the revised manuscript. We also quantify the effect of adding post-processing to the simulations through flagging percentages.

**Reviewer 2:**

*RC2: "General comments: 1. I'm not very familiar with the MPS method and how it works exactly, so I can't really comment on*
*the chosen parametrisation or even on whether the results produced are good/acceptable. However, I do wonder why the simulation artifacts occur? I think this should be elaborated on. It is stated that the geological artifacts are likely due to inconsistencies between TI and conditioning data (L565), but what does that mean exactly?"*

The following is taken from our initial response:
"The main idea behind MPS is that spatial variability is described through a training image. For most MPS algorithms a realization is generated sequentially one voxel at a time. Consider standing at some location in the simulation grid and wanting to know what value to realize in this position. You then gather configuration of the surrounding already simulated nodes (conditional event). Specifically, in direct sampling, as used here, you then search the training image for a matching event. If such an event exist it is placed in the simulation grid and you can move to the next location and so forth until the entire simulation domain is filled.

Artifacts in the simulations might occur if it difficult to find a matching event in the TI. This is what is meant by line 565. If at some point it is impossible to find a suitable candidate, the next best thing must be selected. Sometimes this process leads to simulation artifacts. There are several ways this might be remedied but it all comes at a price. In all cases you are trying to broaden the number of possible matches in the training image (i.e., making sure the prior information is not in opposition to the conditioning data)

A simple way is to reduce the number of already simulated nodes to include in the conditional events. With smaller conditional events, more locations in the TI will be considered reasonable matches. This affects the ability to correctly represent the variability in the long-range structures from the TI. In general, it will lead to increased variability that is not necessarily related to geological variability as portrayed in the TI. Also, when simulating using fewer conditional data, the point being simulated might end up leading to inconsistencies for points to be simulated later.

These problems do not arise in 2-point based Gaussian simulation, because one can always compute the conditional distribution, while in MPS one must infer the conditional from a finite sized example (the TI) represent an outcome of the assumed underlying probability distribution.

An alternative could be to increase the size of TI and hence create additional possibilities for matching events in the TI. However, finding/generating a suitable TI is already the biggest challenge and strength of MPS simulation. The larger TI should still convey the same geology as before but with increased variability, which is not trivial.

We hope this sufficiently answers your question."

We have tried to rewrite the section on simulation artifacts, to a hopefully sufficient degree. We did not want to divert the paper too much from the core message by explaining in the level of detail as the answer above. We have included some lines discussing why these inconsistencies relates to the number of conditionals in the searching event.

*RC2: "Why do the realisations deviate from the TI? "*

"Following from the above we hope it is clear that realization will always deviate from the TI. It should not be the same. The only place we might expect to have a huge overlap between the TI and the realizations is the area from which the TI is derived…. The goal is to simulate new realizations that has the same multipoint statistics, but not the actual structures themselves. "

*RC2: "The non-stationarity of the TI is mentioned but I don't understand what this means."*
The following is taken from our initial response:

"The MPS simulation algorithm is expecting the same spatial variation in the whole training image. A training image that is stationary provides exactly that. For instance, postglacial sediments are only found at the top of TIs in the current study. When the algorithm searches for matching events containing postglacial sediment, matches can then only be found in some specific location. To be more technical, for a process to be stationary, all its statistical properties must be independent of spatial location. This is clearly not the case for a TI built on geology and redox for that matter. Both are extremely non-stationary by nature. Different processes shape the subsurface and it is not providing the same statistical properties at all locations. The issues of non-stationarity in MPS are discussed in detail in Mirowski *et al.* (2009). Another point is that issues with non-stationarity can be remedied with large amounts of conditional data (see e.g., Barford *et al.* (2018) or Vilhelmsen *et al.* (2019))."

We have added a few lines describing the concept of non-stationarity in the TI section. We refer to this during the discussion to help the reader get a fuller perspective on the issue. We hope this makes sense.

*RC2: "It is suggested that artifacts could potentially be removed through better parametrization, but why would that be, and if so, shouldn't you have attempted to do some parameter 'tuning' as part of this work? "*
The following is taken from our initial response:

"From our answers above we hope to lift the lid on why parameterization of MPS simulation algorithms is not straightforward. Parameterizing the algorithm depends on 1) the conditional data events available in the TI (its size, variability and stationarity), 2) computational demands and 3) ultimately if the realizations satisfy the variability that is expected within the training image. We focused more on later part from a qualitative perspective but can also include some quantitative measures also. By going too much into detail with parameter optimization we are afraid to divert the attention away from the core message of the paper. "

We ended up remodeling the area to create a better fit between training images and conditional data. In doing so we tested different parameterizations of the algorithm. We also discuss and justify our prior model through a set of realizations + we quantify the effect of post-processing the simulations through flagging percentages. We have rewritten the discussion to accommodate these new results.

*RC2: "I understand that the artifacts are basically being averaged out over multiple realisations, and I think if the main aim was to produce best estimate and associated uncertainty (entropy) of the geology and redox (Figure 11), then the occurrence of artifacts would be less of an issue. However, if the aim is also to provide realisations as input for transport and fate modelling of nitrate, then I would think the artifacts could become much more of an issue, especially as flow and solute transport will depend non-linearly on those realisations. It is argued that for hydrological modelling at catchment scale,*

*the geological artifacts will have limited effect, which may be true, but I'm less convinced the same is the case for nitrate transport, especially given the redox artifacts."*
The following is taken from our initial response:

"You are completely right here. This will remain an assumption on our part. We do not know to which extent these artifacts affect subsequent hydrological models. The logical next step and what we are currently working on is to explore how these MPS models of the subsurface architecture affect hydrological models. If these redox artifacts are significant, it is possible and trivial within the current workflow to e.g., enforce that below a certain depth only reduced redox conditions would occur. "

With the new modelling setup, we have reduced the number of artifacts. The remaining artifacts are hence less relevant than before. These artifacts are typically oxic conditions that appear at depth within an otherwise reduced environment and should hopefully not influence our hydrological modelling too much. But it remains an assumption.

*RC2: "2. The joint simulation requires a bivariate training image (TI) for geology and redox to be developed. I understand how the geological TIs were developed, but I find it less clear how the redox TIs were produced, who produced them and how the bivariate/joint nature of the TIs are specified. I think this could be better explained."*
We have fully rewritten this section to better inform the reader on the specific modeling choices of the redox TI.

*RC2: "3. For the case study, two independent bivariate TIs are developed, one representing the Quaternary sequence and one representing the buried valley. It is assumed that the delineation of these geological elements is known in the domain (i.e. no uncertainty as to whether a voxel is Quaternary or buried valley). But what is this delineation based on and with what certainty can this be done?"*
The following is taken from our initial response:

"The delineation is based on geological interpretation. At the ground surface this interpretation is primarily guided by topographic information combined with the surface geology maps. At depth borehole information and geophysical data (tTEM, SkyTEM etc.) provides spatial information about the boundaries between the geological elements. Knowing the geological processes and the depositional setting from neighboring areas aids this delineation as well. You raise an interesting question regarding the lack of uncertainty for this boundary. We have discussed this internally within the author group. Clearly there is some uncertainty attributed to the location of these boundaries. The main question is how one can quantify this uncertainty. Also, one must find a way of incorporating these uncertainties in the simulation results. Finally, the effect on the realizations of having such an uncertain boundary must be assessed in comparison to a static boundary. To answer these questions in depth is beyond the scope of the current study. One idea that we would like to follow is to attribute the interpreted boundaries with a quantitative uncertainty based on the interpreter's qualitative uncertainty. By doing so we could simulate several boundary configurations prior to modeling. A set of simulation grids could then be produced that for which we generate one realization. The resulting set of realizations would then satisfy the independence between the geological elements and have a quantitative representation of the uncertainty of this boundary. Alternatively, there is also the possibility to, in direct sampling, simulate with an auxiliary probability grid covering the simulation grid telling which TI to use depending on spatial location. However, this would break the independence between the geological elements as conditional voxels from the buried valley would be included in conditional events for the Quaternary sequence. "

We put forth the idea of resimulating geological element boundaries in the revised manuscript, but have decided not to go into great detail as above within the manuscript.

*RC2: "4. I can't really work out what the influence and significance of the TIs and*
*the conditioning data are for the results. I think it would be relevant to include a discussion on the value of*
*the information used. Given the large amount of conditioning data, does the*
*TI become less important? Would the results be significantly different if you just used the same TI*
*for both the geological elements? What is the effect of the soft data on the result?*
*It would be interesting to see what the results look like if you excluded these. Except for the surface geology,*
*the soft data looks quite uninformative."*

We hope that the inclusion of the prior model now makes it more clear how much information comes from where. This also shows that if we were to use the same TI for both geological elements, we would get totally different results. Especially for redox, where the buried valley showcases planar type redox conditions whereas the Quaternaty sequence has the possibility of geological windows. Also, with the zonation of the lithology-resistivity relationship we have now made a much more informative transformation of the resistivity grid and hence more informative soft data. Furthermore, to make sure that soft data are not overruled in the realizations, some have been converted into hard data.

*RC2: "5. It is not clear to me exactly how topography is used to inform soil moisture when deriving soft data*
*for redox conditions. L365 just states how slope/topography affect moisture in general (it sounds like a*
*TOPMODEL type of approach?). I think further explanation is required here. "*

The whole section is deleted, because new data arrived since the last study and we did not need the topography to aid in deriving soft probabilities.

*RC2: "6. Figures: The 3D figures look impressive, but there is a lot of them, and*
*I think they are quite complex, and the colour choices make them not so easy to 'read' (especially the*
*small ones). I wonder if some of the figures would perhaps be more illustrative in 2D rather than 3D.*
*I don't think Fig 5 works very well as a 3D figure, and in Fig. 6 it*
*is difficult to see that tTEM actually only covers part of the simulation grid. In Figure 1a*
*it took me a while before I realised that the blue, black and red lines where actually the TEM and ERT*
*data points. In Figure 7 the colours are difficult to distinguish in the histograms and it looks like there is*
*a shade of blue in Fig 7a that is not in the legend."*

We have made a new colormap more in line with color choices usually used to represent geological models. The figures have remained in 3D but following the reasonable criticism raised here we changed the viewing angle to better illustrate that tTEM does not cover the entire simulation grid. Furthermore, we have added arrows in the figures pointing towards relevant parts of the simulation results that are mentioned in the text. Figure 1 is changed along with the legend. Figure 7 is redone with the new color scheme as well as the new zonation.

*RC2: "7. Overall, I think the paper is well-written, but there are places where I find the presentation of*
*the material unclear and confusing and where more explanation would be helpful (particularly section 6 and*
*7). I appreciate that this is not helped by the fact that I'm not so familiar with the MPS method."*

The following is taken from our initial response:

"Thank you very much. We will go through section 6 and 7 and try to rewrite them. As many readers of HESS would probably be unfamiliar with MPS we will try to elaborate on some of the concepts that might be confusing from an outsider. "

We hope this is now achieved.

**Minor specific comments from reviewer 2:**

*Abstract: I struggled to understand L16-22 when reading the abstract the first time around (and the similar paragraph in the introduction, L106-110). It makes more sense to me after reading the paper, but I would encourage the authors to sharpen the text here.*

We agree that this line is difficult to read without knowing the context of the paper. This is not ideal and has been changed to hopefully read better. In line 106-110 we deleted the part on how geological elements make the computations more feasible as this is covered in the discussion.

*Figure 1c: Are the white areas unknown soils?*

Yes, this is unmapped territory. The surface geology map is mapped over several field campaigns and some areas are still not mapped. We have updated the map and legend to convey this message better.

*L167: Rephrase. It sounds like you are doing hydrological simulations as part of this work, which you are not.*

Good point. This is not the case. Maybe this has happened because we are currently working on doing hydrological simulation on the realizations as a natural next step for assessing nitrate flowpaths. We have rephrased the sentence.

*L169: There are 7 classes Table 1.*

Yes, this is slightly confusing. The seven classes include Paleogene clay and unknown material, which is not part of the simulation. We have deleted the two classes that are not used in the geostatistical simulation. We hope this makes it less confusing.

*L199: Not sure I understand why they provide independent measurements of redox.*

This whole section has been rewritten.

*L255: consists of. . .*

Corrected

*L306: From figure 1 It looks like the data availability is better in the southernmost part of the domain?*

There are several criteria for selecting an appropriate site for TI construction. We fully acknowledge that the amount of geophysical data and boreholes tends to be slightly better in the southernmost part of the domain at surface level. However, the quality and depth of the boreholes in the chosen area is far superior to the southernmost part. There are only few wells that penetrate all the way down through the modelling domain in the area, and they are important to include in the TIs to make it more robust. We have tried to convey this message in the manuscript.

*L310-311: I do not follow this. Why is sandy till not included in the TI? How does that affect simulations for the buried valley and can simulations here then meaningfully be conditioned to observations of sandy till?*

No, they cannot. Sandy till is then only present at the surface where it is placed as hard data from the surface geology map. We have included some sandy till in the TI to accommodate this problem.

*L321: I don't understand how its eases the construction of the TI to include parts outside the domain. Please clarify.*

Because you get additional information about how the geology behaves that is independent information from whatever goes on in the simulation area. We have tried to convey this message in the revised manuscript.

*P16 and Fig 7: I find it difficult to follow the text and observations that go with Fig 7. It is stated that sandy till is associated with some of the highest resistivities, but I can't see this from Fig 7. I don't follow how the general relationship /distribution of resistivities have been derived (Fig 7c). I think this whole section could be clearer.*

To correct this, we have implemented a new colormap for resistivity as well as for the lithology classes. This makes it easier to see the sandy till as it is now shown in black. More importantly we have accounted for some of the non-stationarity in the resistivity-lithology calculation by introducing three zones to obtain the relationship. We have tried to clarify the way to obtain resistivity-lithology relationship in the text with this comment in mind.

*Equation 2: I'm not sure I follow the equation. Is the soil map (surface geology data) used to inform the lithology below the surface layer as well in a similar way as in the buffer zones as described in section 5.4.2? Or what is the probability of surface geology below the surface layer (intuitively I would think zero probability in which case the equation collapses)?*

No, the surface geology map is only used to inform our model at the surface. The sum of all probability distribution over all categories must sum to 1. Hence the model will never collapse. You are right that in the case where all categories had zero probability this would happen. In an uninformed case the total probability of 1 is hence uniformly distributed amongst each category.

*L435: . . .overlapping relationship. . .*
Corrected

*Soft data: do you derive soft probabilities for geology for all voxels in the simulation grid, even where hard data are present? It looks so from Fig 8 and 9.*
Yes. But in practice hard data overrules soft data

*457: To what extent did you experience "flagging" as part of the simulations you generated?*
We have added a few lines describing the flagging process and the percentage of "flagging" and the impact of postprocessing for both prior and posterior model.

*L485-500: I found some of the described observations here difficult to see in Figure 10. Maybe it would be easier to follow if there was a close-up on a relevant part of the transects instead?*
Good point. Again, we want to convey the message that these models are in 3D but at the same time these 3D plots are not always satisfying for showing the results. We have now introduced arrows pointing towards phenomena that are mentioned in the text. This we feel provides a better overview for the reader.

*L555: I'm not sure I understand why and when your simulations would not honour TI and data.*
The TI is honored if the spatial variability and patterns shown in it are represented to some degree in the realizations. The conditional data should be honored by the algorithm. In practice, when using many conditional data with MPS, there will be inconsistency between the conditional data and the TI: The conditional events will not exist in the TI. This is due to the limited size of the TI (one of mulitiple sources).

This sentence is moved to the newly introduced paragraph on the prior model, which should hopefully clarify what is meant.

*L574-585: I think the text here reads so well and it could be improved. This is also where terms like stationary TI and rotation in simulation are mentioned, which I find difficult to really follow not having a MPS background.*

We have rewritten this section to hopefully read better. Also, we introduce the concept of stationarity in the TI earlier in the text of the current manuscript. This should hopefully aid in the understanding of this paragraph.

**Summary of major changes to the manuscript**

- Parts of the abstract is rewritten to make it more accessible to readers not familiar with the technical details of the paper
- The view of all figures of the simulation grid is made more oblique such that it is easier to see details of the model and aspects of the surface.
- We have added a more in-depth explanation on the creation of the TIs as well as a figure showing the profiles of redox conditions based geochemical observations used to make the redox TIs.
- Redox TIs have been re-modelled with new data. A small patch of sandy till is added to TI2 to allow the simulation of sandy till at the surface, which is more in accordance with the surface geology map.
- Added a section on prior model and how it fits with our expectation of the spatial variability of the system. This is now used as justification for using the current parameterization of the sampling algorithm
- Added a new colormap to geology more in line with usual color palette of geological mapping which is also makes it easier to separate colors visually in the realizations.
- The slope factor in the conversion of surface geology map into soft redox probabilities is discarded to avoid any confusion and new values are introduced based on new geochemical observations. The paragraph is therefore rewritten.
- We have implemented a zonal system for inferring the lithology-resistivity relationship. This means that we can account for some of non-stationarity that governs this relationship with depth. In addition, this new relationship helped the undermining of postglacial sediments in the former relationship as well as remedying the effect of very low resistivities for meltwater sand/gravel in the whole domain. This had also a positive effect on inconsistencies between the conditional data and the prior model, leading to fewer simulation artifacts. Consequently, the paragraph is rewritten at large as well as a new figure has been produced.
- The new lithology-resistivity relationship meant that soft probabilities have been recalculated and the section is altered slightly
- In terms of parameterization of the MPS algorithm we have expanded this section to include the prior model, which should hopefully reveal more of our thoughts and considerations regarding parameterization tuning.

- We have section in the discussion dealing with the role of soft data and expanded our thought on inconsistencies between prior and conditional data.

**The following figures have been updated (old figure number in parathesis):**

- **Figure 1 (Figure 1):** The legend for geophysical data is updated to lines instead of dots. The legend for geochemical data has been changed for "this study". The lakes are now shown in blue in (c), while the unmapped territory is kept in white.
- **Figure 2 (Figure 2):** Because the colormap of both resistivity and geology has been changed, the input data in Figure 2 is changed to create a more coherent manuscript
- **Figure 3 (Figure 3):** Changes to the new camera position of the simulation grid.
- **Figure 4 (Figure 4):** The colormap of geology is changed and a new legend is presented. a subfigure c) has also been added to show the zones used for inferring the resistivity lithology relationship. The TIs have also been slightly altered.
- **Figure 5 (NEW):** A new figure on the redox conditions of the study area used for a more in-depth analysis and justification on the creation of the training images.
- **Figure 6 (Figure 5):** Updated with new colormap and legend
- **Figure 7 (Figure 6):** A new colormap for resistivity is introduced to better visualize the logarithmical nature of resistivity measurements and thereby see the structures provided by the tTEM data
- **Figure 8 (Figure 7):** The new implemented calculation of the resistivity-lithology relationship for the three zones displayed in figure 4 is now shown instead of the combined calculation. The new colormap is also adopted for this figure.
- **Figure 9 (Figure 8):** Figures updated with new soft probabilities and added (f) to show the spatial distribution of the lithology-resistivity zones in the simulation grid.
- **Figure 10 (Figure 9):** Mode and Entropy recalculated from soft probabilities in figure 9 and new colormap adopted for the mode.
- **Figure 11 (NEW):** New figure showing two unconditional realizations from the prior model
- **Figure 12 (Figure 12):** Reworked figure of accumulative probability of redox. Now shown for TIs, Prior and Posterior instead of only posterior model as before.
- **Figure 13 (Figure 10):** Figure updated with new modeling results and made with same configuration as figure 11 for easier comparability
- **Figure 14 (Figure 11)::** Figure updated with new modeling results summary and new legend + colormap

**The following tables have been updated:**

- **Table 1:** We have removed the last two lithological group names as they are not present in geostatistical simulation and hence create more confusion than clarification. We have also changed "soil" into "sediment" to be more consistent with the manuscript.
- **Table 3:** The soft probabilities of the if redox conditions at the surface are changed and are now based on redox measurements from wells in and around the study area instead of the arbitrary "slope factor" that was originally introduced. We also do not discern between the two geological elements.
- **Table 4:** New table created to show the fraction of conversion from soft to hard data.

---

## Referee Report (RR1)

Review comments for HESS manuscript 2020-444 by Donald A. Keefer
Bødker Madsen et al., **version hess-2020-444-ATC1.pdf**.

To the authors, well done. You've done a good job with this paper and made some very significant changes. I do still have a few concerns. Nothing insurmountable, but definitely issues that need to be addressed in some manner.

I also want to recommend a publication. It's a chapter by Mohan Srivastava in a book about geostatistical simulation. It's fantastic and has some perspectives about simulations that are really insightful. The whole book is very good, but some of the methods are a bit dated now. Srivastava's chapter is still as relevant as when he wrote it, and can be extended to MPS, which he and Guardiano had developed two years earlier.

Srivastava, R. M. (1994). An overview of stochastic methods for reservoir characterization. In J. M. Yarus & R. L. Chambers (Eds.), Stochastic Modeling and Geostatistics: Principles, Methods, and Case Studies (pp. 3–16). The American Association of Petroleum Geologists.

***Manuscript Evaluation* version hess-2020-444-ATC1.pdf**
Review Criteria:
Scientific Significance. I rate this paper as 1.5, Good-Excellent.
Scientific Quality. I rate this paper as 2.5, Fair-Good.
Presentation Quality. I rate this paper as 2, Good.

I suggest that this manuscript would be an excellent choice for publication in HESS, if the authors address my comments to the satisfaction of the Editor. I do not need to see it again.

Aspects for Consideration:
1. The paper does address relevant questions within the scope of HESS. Characterization of spatial distribution of geologic deposits and redox status.
2. The paper presents novel ideas and methods.
3. The conclusions are substantial: this method offers a way to effectively model the sediment distribution and associated redox conditions.
4. The methods and assumptions are generally valid and clearly outlined.
5. The results are generally sufficient to support interpretations and conclusions. However, some of the interpretations are not sufficiently established by this study. See comments below.
6. The description of experiments and calculations is sufficiently complete and precise to allow reproduction by fellow scientists.
7. The authors give proper credit to related work. They indicate their own contribution fairly well. There are a few places where they offer conclusions that I do not feel are substantiated by either this experimental design or by the results. I comment further on this below.
8. The title is a good reflection of the study.
9. The abstract is a good reflection of the study.
10. The overall presentation is very good and clear.
11. The language is generally fluent, but occasionally not sufficiently precise. I have comments to this below.
12. The formulae, symbols, abbreviations and units are correctly defined and used.
13. I believe that section 7.4 could be reduced or eliminated, as it is misleading and seems at odds with the bulk of the literature and even earlier passages in the paper. I address this more in my comments, below.
14. The number and quality of references are sufficient.
15. There is no supplementary material.

Major comments: **version hess-2020-444-ATC1.pdf**
The authors state unequivocally that geostatistical simulation can quantify uncertainty in the geological model. This capability is overstated. Geostatistics and geostatistical simulation rely on different statistical models that are, or should be, based on data and conceptual models of likely rock property distributions. While MPS can be more effective at simulating more-realistic sediment texture patterns than other geostatistical simulation techniques, these all are only approximations. And the statistical models they are based are only constrained estimates of the properties being modeled – to the degree that the selected models reflect aspects of the real system; and, that they sufficiently sample, with high reliability, the properties being simulated. I suggest the authors adopt this perspective, and add the word, 'estimate' or 'estimating', when talking about quantifying uncertainty. 'Quantitative estimate of uncertainty' is a good example.

The authors are inconsistent in their representation of the impacts of bias, subjectivity, and independence. They often suggest that subjectivity induces biases that are always undesirable. At other times, they advocate clearly for choices or methods that induce a specific bias, often based on the subjective justification that the choice better represents a preferred conceptual understanding. This inconsistency, and more importantly the erroneous representation, are significant in that they lead to confusion about the authors' advocacy regarding these concepts, and even to how the authors understand them. On lines 84-87 they state "In fact, subjective biases are accepted as one of the weak points of cognitive geological modeling…"  In this situation the authors cite Bond 2015 and Wycisk et al. 2007 for this statement. It appears, however, that the authors have misrepresented both of these papers. Bond does not make this statement in her paper, and frequently recognizes the importance of geologic knowledge for successful modeling. Wycisk et al. never use the words subjective or bias, and state that knowledge of the system is required to make any modeling successful. Biases are endemic to any form of modeling – cognitive or quantitative (numerical, probabilistic, geostatistical). Generally, you want to add a specific bias as a constraint to interpretation (e.g., texture distributions in different training images, or the proportion of specific sediment textures in a proximal proglacial sedimentary setting). The main down side of expert insight in cognitive modeling is the difficulty in elucidating the various biases that the experts use. However, expert biases can be evaluated for likely correctness, and these 'constrained' biases can be very helpful in ruling out improbable sediment distributions; both Bond (2015) and Wycisk et al. (2007) recognize this. The implication of the authors' phrasing suggests that quantitative modeling is always better and less biased than cognitive modeling. I suggest that the literature demonstrates that both can be very useful if done correctly and properly qualified – and both can be similarly erroneous and misleading if not done correctly or properly qualified. I recommend language like, 'Subjective biases are seen by some as one of the weak points of cognitive geological modeling (Bond, 2015; Wycisk et al., 2009). It can be difficult to identify and quantify uncertainties due to biases in subsurface predictions from cognitive modeling, and so these biases cannot be fully accounted for in subsequent analysis or process modeling efforts (i.e., hydrologic modeling).'

On line 1000 the authors also seem to misunderstand importance and need for independence, in a statistical sense, and how bias is intentionally inserted in MPS. The goal of a TI is to introduce a bias, subjectively generated based on a combination of geological knowledge of the area and depositional settings, and observational data of the location. The reason MPS is being recommended, rather than other categorical geostatistical methods, is that these other methods don't introduce the right biases. In fact, they impose biases that make the models more incorrect than is generally desired. MPS is a way to introduce biases of more spatial continuity and large- to medium-scale heterogeneity into simulations, in order to generate model realizations that are more biased towards patterns of heterogeneity that are observed in outcrop and modern depositional analogues. Assumptions (or requirements) of statistical independence are part of the theoretical underpinnings of the tests. However, these methods are being used as approximations of more

complex systems, and these assumptions can be (and usually must be) relaxed without making the MPS simulations worthwhile.

Figure 2: Input data…Soil Map. Should be Surficial Geology Map

Figure 8 revision is a great contribution. However, the Y axis label on 8a, 8b, c are wrong. Should be a frequency or count.  I'm a bit confused about how this was generated. See other comments in this section.

Figure 12, the X axes are incorrectly labeled. Should be 'percent' or maybe 'percent occurrence' or change x-axis values to be 0-1 for probability.
Figure 12b includes results of two realizations. This isn't sufficient estimator of the prior. You need to label it with some other label.

In general the figure revisions are much better. Good job on these. The patterns are much more readable and easier to follow with the discussions. A few more fence diagrams might be helpful in showing more of the internal architecture.

There seems to be a bit of misunderstanding about the relationships between the data used for modeling redox conditions and about what one of the data streams means. On lines 235-236 the authors state, "The benefit of using the two types of data is that they provide independent measurements of redox conditions." This is incorrect, these data are not independent. The redox colors of the sediments reflect long-term redox conditions which are consistent with some period of pore-water chemistry. Importantly, the authors do not mention that the redox sediment colors are indicators of long-term historic redox conditions and not necessarily reflective of the redox dynamics within the recent, human altered subsurface. This may be why they feel the two data streams are independent. This study is not structured to evaluate this independence, and prudence suggests it is better to assume some level of dependence. The benefits of using the redox colors are that there are more of those data values than there are of the water chemistry, and the authors may assume that the relatively smooth variations in lateral redox sediment colors allow more reliable interpolation of the colors to unsampled locations than can be assumed for the recent shallow water chemistry. It would be helpful if the authors could add information or interpretations on how the redox colors correlate to water chemistry. The results might show a good agreement, or not such a good agreement. Importantly, the water chemistry has a temporal dynamic that is not captured by the colors.

Also regarding the redox discussion, on lines 406-411, the authors make statements about inferring groundwater flow conditions from the redox colors of the sediments in the lowland (buried valley). However, the authors haven't presented any data that lets them make inferences about flow in the groundwater system. They are obviously able to cite other work in the area that did study gw flow, but no conclusions about flow can come from this work. Instead, the authors could say their work is, or is not, compatible with hydrologic observations and conceptual models documented by others (e.g., Kim et al., 2019). As a case in point, the authors suggest no horizontal flow because of clay-dominating conditions. However, a plausible alternate interpretation could be that soil development has provided preferential flow paths through the upper 3 meters of material across the landscape, that this network is exploited for groundwater flow, and the extended flowpaths from infiltrating up-gradient positions might provide sufficient residence time of nitrate-laden water for significantly more reduction. I am not advocating for either hypothesis, just noting that there are no data to prove or refute either of these ideas.

There is more clarity of explanation needed for the discussions about how soft data are used and processed. Importantly the authors made some very helpful changes from an earlier version. This is in lines 533-575 of the ATC1 version. I particularly like the clarification of the potential impacts of non-stationarity and how this was handled. I like the use of the three zones. That's all very good. However, I'm particularly concerned and

confused about the processes surrounding figure 8. It looks like the histograms were generated from assumptions of resistivity/lithology and the resistivity grid. The authors need to clarify the uncertainties or potential for error in the assumptions made and how these relationships were applied. There doesn't seem to be any calibration or validation around these resistivity/lithology relationships. This seems very subjective. Not necessarily wrong, but it seems to involve a several significant assumptions that warrant some recognition.

Table 4: I don't understand what the upper row signifies. I feel like I understand the text general discussion that supposedly addresses the table, but I don't understand what the values mean. The first row needs a better explanation.

On line 660 the authors cite Høyer et al. (2017) in their use of 'unconditional simulation'. In fact the simulations are conditioned by the Tis, but not on any data. However, Høyer et al. don't refer to this unconditional simulation as representing the prior distribution. They used this approach to evaluate how effectively the TI constrained the realizations. While the authors are using TI-conditioned simulation for this purpose as well, it is not correct to suggest that two realizations are able to provide sufficient representation of the whole prior distribution. This is important on Figure 12b, as you suggest it is the prior, but again, only two realizations (i.e., samples).

On line 791 the authors state, '…and hard data increase the information content (lowers entropy)…'. I believe this is incorrect. This should be written as, 'and hard data **decreases** the information content (lowers entropy)…' Remember, the equation is for Shannon's information entropy, not thermodynamic entropy. (cf. https://stats.stackexchange.com/questions/101351/entropy-and-information-content). Alternatively, the authors could think of entropy in terms of bits – a la Shannon. It takes more bits to describe more complexity, and fewer bits to describe less complexity. More bits = more information content. Fewer bits = less information content.

On lines 808-809 the authors state, 'This imply that simulation artifacts are not reoccurring in simulations…' I would strike this sentence. It's not phrased properly and is incorrect; analysis of the mode, as a statistic, doesn't prove or disprove either the frequency of artifacts or a bias. If the authors want to keep a reference to these figures, the text should be fixed so it doesn't erroneously say, 'no overall bias is found'. Maybe something like, 'The algorithmic biases that create these artifacts are generating a small number of artifacts that are not significantly affecting the posterior histograms.' However, this point was already made when the authors looked at the histograms earlier in the paper, so this sentence is not necessary or constructive. The statement, 'This is clearly an unwanted side-effect of the soft data conversion…' seems like an inference based on assumptions rather than an observation based on the algorithm. If that is the case, this statement is unsubstantiated and should be deleted. Consider a hypothetical example: any additions to the distribution would have no effect on the mode as long as they occurred less frequently than the modal value. These additions could be due to a systematic error (i.e., bias).

On lines 815-832, I think this passage is mistaken. Importantly, the redox scale only has three categories. The posterior distribution showed almost no occurrence of the middle, reducing, category. Don't forget the post-glacial seds were assigned a highly reducing value. And even in other textures, there was a non-negligible proportion of reduced occurrences. These all probably contribute to generate a higher entropy at land surface. The prior and TI for redox were strongly biased towards reduced conditions at a certain point in the subsurface. This is reflected in the low entropy. I don't see this as counter intuitive.

On line 959, the authors state, 'Although the small size of the TI may pose a problem for reproducing the intended variability…' They have been a bit inconsistent with how they refer to the Tis in terms of their level of detail, the intended goals with respect to the level of detail the authors are trying to represent, and with

the recommendations on how much detail to put into a TI. These decisions do involve a bit of professional judgement, more clarity is needed about the goals for the TIs, what the authors tried to do with the TIs, and how they are accommodating the limitations that the choices caused.

On line 1016 the authors state, '…allows the quantification of uncertainties in the input data…'  No it does not do this. It is not correct to use statistical models that were created by a data set to evaluate the uncertainties of that same data set. There isn't that much real data to begin with, so there really isn't enough data to prove any distributional assumptions or models. The manuscript details how the uncertainty in the geologic sediment and redox condition distributions have been quantitatively estimated.

Technical Concerns:
I present here a list of grammatical and minor technical concerns, listed by line number from the **version hess-2020-444-ATC1.pdf**.

Line Number: Comment
8: 'nitrogen (N) losses'…you are only dealing with one loss pathway, leaching. should use 'nitrogen (N) leaching' rather than 'losses'.
13-15: 'and 2) information of lithology and redox conditions…' this list is presented incorrectly. it should be: '2) lithologies from borehole observations, 3) redox conditions from colors reported in borehole observations, and 4) chemistry analyses from water samples.'
23: '…and conditional data'…should be: '…and conditioning data…' See Straubhaar, Renard, Mariethoz, 2016.
26: maybe 'consistent' rather than 'coherent'?
30: 'which may be fundamental to better understanding of the retention of the subsurface…" might read better as: which may lead to a better understanding of nitrogen (N) fate in the subsurface…"
34: maybe 'loss' instead of 'escape'…leaching is not ideal here because your citations also include runoff losses
50: questions: Do you mean nitrate concentrations, instead of conditions? If 'conditions' is correct, I'm not sure what that means. Can you clarify. Assuming concentrations is correct, shouldn't it also be 'leaching concentrations'?
56: 'contaminants: 1)'   should be something like, "contaminants. Modeling approaches have included: 1)…"
66: I would suggest a better wording than, '…space thus requires…" might be, "…space would benefit from more-detailed…" Mostly, this seems to imply that your 25x25x2 is sufficient to capture all detailed heterogeneities in redox conditions, which isn't accurate (it's probably also not what you mean…it just seems implied in the current wording).
80: 'is attributed to uncertainties such as' should be 'contains uncertainties from sources that include'
84-87: "In fact, subjective biases are accepted as one of the weak points of cognitive geological modeling…"  The authors cite Bond 2015 and Wycisk et al. 2007 for this statement. The authors have misrepresented both papers. Bond does not make this statemente and recognizes the importance of geologic knowledge. Wycisk et al. never use the words subjective or bias and state that knowledge of the system is required to make any modeling successful. See comments above about the authors' use of subjectivity, bias, and uncertainty concepts.
105: the geostatistical simulation doesn't "quantify uncertainty in…data". It does allow you to quantitatively estimate uncertainty in the unsampled locations, but not the sampled ones.
106: possible realizations of the subsurface…not a quantitative measurement of uncertainty?
235-236: "The benefit of using the two types of data is that they provide independent measurements of redox conditions." No that is not correct. The colors clearly reflect long-term redox conditions which are consistent with the pore water chemistry. So they aren't independent. See comments above.
324-325: '…based on geological event chronology.'  should it be '…based on sediment heterogeneity and geological event chronology.'  ??

406-411: You haven't presented data that lets you make any inferences about flow in the groundwater system. See comments above.

458: '5.4 Conditional data'   should be '5.4 Conditioning data'  There are conditional simulations and conditional realizations that are conditioned to the data. This means the data become condition**ing** data.

533-575. I like the revisions to this portion of the manuscript. See comments above.

586+ I'm a little concerned about the term, 'soft probability'. I realize Caers and Mariethoz may use it, but qualifying the word 'probabilities' with 'soft' seems inaccurate or at least unclear. 'Soft data probabilities' is more intuitive and seems more correct. I notice you use 'soft data probability' on page 10. Please check on the usage of this term and consider either using 'probabilities' or 'soft data probabilities'. I leave it to your discretion.

587: 'resistivity grid in Figure 7' probably makes more sense as, 'resistivity grid from Figure 7'

614-616: While the mode shows layers in the buried valley, which might be bad, these same voxels show a lot of entropy, which implies a wide distribution over these cells. Which is good. You might make this point in the text. I think it's helpful analysis discussion for readers.

623: Table 4. I don't understand the upper row. I understand from the text, what general topic it's addressing, and I think it is probably helpful. However, I don't understand what the values mean. The first row needs a better explanation.

633: 'In direct sampling, the nodes in the simulation grid…' You never discuss this term (direct sampling) nor contrast it with alternative MPS simulation modes. Maybe back up when you introduce MPS, put in a couple sentences about direct sampling and snesim and that your version uses direct sampling. Just for context here.

635: can you add a citation or citations that you used to generate this list of parameters?

660: Høyer et al. (2017) do not refer to this unconditional simulation as the prior distribution. See comments above

667:  'only occur near the surface in accordance with the TIs, but more infrequent than portrayed.' …with some corrections… 'only occur near the surface in accordance with the TI, much more infrequently than portrayed.'

686: '…incorporation of conditional data…' should be, '…incorporation of conditioning data…'

709: 'conditioned data' should be 'conditioning data'

718: '…the soft data is particularly dominating the realizations…' could be '…the soft data is the dominant constraint on the realizations…'

726: '…This coherency explains…' could be '…This consistency explains…'

740: question: why don't you more completely filter these oxic artifacts? even in post-simulation processing?

778-779: '…which together represent the full posterior model.' I think it would be more correct to say something like, '…which together are used to represent the posterior model.'

787: '…indicating that other categories are almost equally probable…' probably should be something like, '…indicating that these voxels have a near uniform distribution among several different categories.'

791: '…and hard data increase the information content (lowers entropy)…' should be, 'and hard data decreases the information content (lowers entropy)…' See comments above.

808-809: I would strike this sentence. See comments above.

815-832: I think you're mistaken here. See comments above.

836: 'examples of mapping redox conditions with multiple-point geostatistical simulation…' I don't think the use of mapping is appropriate here. It's the wrong audience for this use. A better wording would be, 'examples of simulating redox conditions with sediment-texture distributions using multiple-point geostatistical methods…'

848: 'The spatial variability of the TIs is well represented in the prior realizations…" Not really. You only generated two realizations, which is insufficient to estimate the prior distribution, and the prior distribution that you list is not a good estimate of the redox TI - the green zone (reducing) is not well simulated at all.

848: 'conditional data'…'conditioning data'

854, 857: 'conditional data' …'conditioning data'

847-865: Nice job on this section. It's a good contribution.

868: 'The random path have a tendency to underestimate soft data…' ? You later suggest this is why the

877: 'randomly converted a fraction of the soft data…' can you specify the fraction? You are defining a novel method and this would probably be helpful to people.

939-961: 7.3 'The inclusion of geological mapping experts in the creation of ThIs introduces modeling subjectivity.' See comments above.

959: Although the small size of the TI may pose a problem for reproducing the intended variability…" earlier, you stated that you wanted a generalized model that didn't fully capture all of the possible geometries. can't then make poor fit a criteria.

1000 "…would ensure independence of information." See comments above.

1003: conditional >> conditioning data

1007: computationally feasible… 80 seconds per realization is trivial. Could have done many more than two realizations to estimate the prior.

1016 7.7 …allows the quantification of uncertainties in the input data…" No it does not. See comments above.

---

## Author Response (AR2)

The authors state unequivocally that geostatistical simulation can quantify uncertainty in the geological model. This capability is overstated. Geostatistics and geostatistical simulation rely on different statistical models that are, or should be, based on data and conceptual models of likely rock property distributions. While MPS can be more effective at simulating more-realistic sediment texture patterns than other geostatistical simulation techniques, these all are only approximations. And the statistical models they are based are only constrained estimates of the properties being modeled – to the degree that the selected models reflect aspects of the real system; and, that they sufficiently sample, with high reliability, the properties being simulated. I suggest the authors adopt this perspective, and add the word, 'estimate' or 'estimating', when talking about quantifying uncertainty. 'Quantitative estimate of uncertainty' is a good example.

**We agree that the uncertainty quantified here is a representation of the uncertainty of the specific probabilistic system and should only be viewed in this context. This was misleading on our part. We have changed accordingly in the manuscript.**

The authors are inconsistent in their representation of the impacts of bias, subjectivity, and independence. They often suggest that subjectivity induces biases that are always undesirable. At other times, they advocate clearly for choices or methods that induce a specific bias, often based on the subjective justification that the choice better represents a preferred conceptual understanding. This inconsistency, and more importantly the erroneous representation, are significant in that they lead to confusion about the authors' advocacy regarding these concepts, and even to how the authors understand them. On lines 84-87 they state "In fact, subjective biases are accepted as one of the weak points of cognitive geological modeling…" In this situation the authors cite Bond 2015 and Wycisk et al. 2007 for this statement. It appears, however, that the authors have misrepresented both of these papers. Bond does not make this statement in her paper, and frequently recognizes the importance of geologic knowledge for successful modeling. Wycisk et al. never use the words subjective or bias, and state that knowledge of the system is required to make any modeling successful. Biases are endemic to any form of modeling – cognitive or quantitative (numerical, probabilistic, geostatistical). Generally, you want to add a specific bias as a constraint to interpretation (e.g., texture distributions in different training images, or the proportion of specific sediment textures in a proximal proglacial sedimentary setting). The main down side of expert insight in cognitive modeling is the difficulty in elucidating the various biases that the experts use. However, expert biases can be evaluated for likely correctness, and these 'constrained' biases can be very helpful in ruling out improbable sediment distributions; both Bond (2015) and Wycisk et al. (2007) recognize this. The implication of the authors' phrasing suggests that quantitative modeling is always better and less biased than cognitive modeling. I suggest that the literature demonstrates that both can be very useful if done correctly and properly qualified – and both can be similarly erroneous and misleading if not done correctly or properly qualified. I recommend language like, 'Subjective biases are seen by some as one of the weak points of cognitive geological modeling (Bond, 2015; Wycisk et al., 2009). It can be difficult to identify and quantify uncertainties due to biases in subsurface predictions from cognitive modeling, and so these biases cannot be fully accounted for in subsequent analysis or process modeling efforts (i.e., hydrologic modeling).'

**An important point raised here. We fully agree with the presented argument and that our phrasing was unfortunate. This is corrected to the suggestion made by the reviewer.**

On line 1000 the authors also seem to misunderstand importance and need for independence, in a statistical sense, and how bias is intentionally inserted in MPS. The goal of a TI is to introduce a bias, subjectively generated based on a combination of geological knowledge of the area and depositional settings, and observational data of the location. The reason MPS is being recommended, rather than other categorical geostatistical methods, is that these other methods don't introduce the right biases. In fact, they impose biases that make the models more incorrect than is generally desired. MPS is a way to introduce biases of more spatial continuity and large- to medium-scale heterogeneity into simulations, in order to generate

model realizations that are more biased towards patterns of heterogeneity that are observed in outcrop and modern depositional analogues. Assumptions (or requirements) of statistical independence are part of the theoretical underpinnings of the tests. However, these methods are being used as approximations of more complex systems, and these assumptions can be (and usually must be) relaxed without making the MPS simulations worthwhile.

**We think that the word "bias" is used differently here by the reviewer than how it used in the manuscript, which could explain this confusion. The message we have tried to get across in this line is a general concept from geostatistics. The TIs in the current study are based to a certain degree on the conditioning data. When we generate realizations from such a model to represent the prior distribution it does carry some information that is also present in conditional data. In the combined posterior models, we thus lose some degrees of freedom, and therefore one could argue that it most likely would be preferable to use a more conceptual TI to ensure that no degrees of freedom are lost. Furthermore, MPS is a way to make geostatistical simulations based on a spatial variability quantified through a training image. That's it. It is not a way to introduce biases into simulations. The reviewer is right that the spatial variability described in the TI affects the outcome of the simulations. As they should. And also, that the TIs represent all our assumptions about the system within the investigation scale. We have been upfront about this in the manuscript as well.**

Figure 2: Input data…Soil Map. Should be Surficial Geology Map

**It doesn't say soil map in the figure, but instead Surface geology mapping. We have changed it to surficial geology mapping according to the suggestion of the reviewer.**

Figure 8 revision is a great contribution. However, the Y axis label on 8a, 8b, c are wrong. Should be a frequency or count. I'm a bit confused about how this was generated. See other comments in this section.
**This is correct. The Y axis was wrong in the figure and has been changed to counts instead.**

Figure 12, the X axes are incorrectly labeled. Should be 'percent' or maybe 'percent occurrence' or change xaxis values to be 0-1 for probability.
**Changed to "percent occurrence"**

Figure 12b includes results of two realizations. This isn't sufficient estimator of the prior. You need to label it with some other label.

**This is a valid point. The two realizations themselves are not sufficient to describe the prior distribution. We have now based this probability with depth upon more realizations of the prior.**

In general the figure revisions are much better. Good job on these. The patterns are much more readable and easier to follow with the discussions. A few more fence diagrams might be helpful in showing more of the internal architecture.

**Good point. Additional fences have been added to help convey some of the internal structure.**

There seems to be a bit of misunderstanding about the relationships between the data used for modeling redox conditions and about what one of the data streams means. On lines 235-236 the authors state, "The benefit of using the two types of data is that they provide independent measurements of redox conditions." This is incorrect, these data are not independent. The redox colors of the sediments reflect long-term redox conditions which are consistent with some period of pore-water chemistry. Importantly, the authors do not mention that the redox sediment colors are indicators of long-term historic redox conditions and not necessarily reflective of the redox dynamics within the recent, human altered subsurface. This may be why they feel the two data streams are independent. This study is not structured to evaluate this independence,

and prudence suggests it is better to assume some level of dependence. The benefits of using the redox colors are that there are more of those data values than there are of the water chemistry, and the authors may assume that the relatively smooth variations in lateral redox sediment colors allow more reliable interpolation of the colors to unsampled locations than can be assumed for the recent shallow water chemistry. It would be helpful if the authors could add information or interpretations on how the redox colors correlate to water chemistry. The results might show a good agreement, or not such a good agreement. Importantly, the water chemistry has a temporal dynamic that is not captured by the colors.

**We agree on the reviewer's comment about the differences in the redox interpretations between the sediment colors and water chemistry. Therefore, we revised the manuscript as follows:**

**Line 217-222: The sediment colors may be the resultant of the cumulative effects of the redox structure evolution since the deglaciation while the water chemistry may display a snapshot of the short-term redox chemistry, which may be temporally variable. Therefore, we postulate that the redox conditions interpreted from the sediment colors may be more coherent with the geological structure than that of the water chemistry. In addition, the sediment colors provide 1D profile information of the redox conditions and more data points are available compared to water chemistry which provide point scale information.**

Also regarding the redox discussion, on lines 406-411, the authors make statements about inferring groundwater flow conditions from the redox colors of the sediments in the lowland (buried valley). However, the authors haven't presented any data that lets them make inferences about flow in the groundwater system. They are obviously able to cite other work in the area that did study gw flow, but no conclusions about flow can come from this work. Instead, the authors could say their work is, or is not, compatible with hydrologic observations and conceptual models documented by others (e.g., Kim et al., 2019). As a case in point, the authors suggest no horizontal flow because of clay-dominating conditions. However, a plausible alternate interpretation could be that soil development has provided preferential flow paths through the upper 3 meters of material across the landscape, that this network is exploited for groundwater flow, and the extended flowpaths from infiltrating up-gradient positions might provide sufficient residence time of nitrate laden water for significantly more reduction. I am not advocating for either hypothesis, just noting that there are no data to prove or refute either of these ideas.

**We have presented the sediment colors and water chemistry data in Figure 5, which demonstrate no secondary oxic layers in the buried valley unit. The reviewer has a valid point regarding the hypotheses to explain the buried valley's redox structure. Currently, we could not rule out either explanation. Therefore, we revised the manuscript as follows:**

**Line 380-383: A secondary oxic layer below the first oxic layer is not expected, due to the clay-dominant conditions of the surface geology (mainly clay-till; Figure 1c) and subsurface structure. We concluded that in the buried valley, oxidants are delivered either vertically via water infiltration or gas diffusion or the top oxic layer (4-6 meters below the land surface) from the Quaternary sequence, resulting in the planar type of redox architecture (Kim et al. 2019).**

There is more clarity of explanation needed for the discussions about how soft data are used and processed. Importantly the authors made some very helpful changes from an earlier version. This is in lines 533-575 of the ATC1 version. I particularly like the clarification of the potential impacts of non-stationarity and how this was handled. I like the use of the three zones. That's all very good. However, I'm particularly concerned and confused about the processes surrounding figure 8. It looks like the histograms were generated from assumptions of resistivity/lithology and the resistivity grid. The authors need to clarify the uncertainties or potential for error in the assumptions made and how these relationships were applied. There doesn't seem to be any calibration or validation around these resistivity/lithology relationships. This seems very subjective.

Not necessarily wrong, but it seems to involve a several significant assumptions that warrant some recognition.

**We disagree that these choices are very subjective. They are subjective but are mainly based on the relationship inferred from the TI. The TIs on which this relationship is mainly built is subjective. If subjectivity is a concern, the TI construction is more the area to focus on.**

**The most subjective part of our estimation of the relationship comes from the weighting with the background relationship. The reason we do this is because we want to be true to our general understanding of the lithology-resistivity relationship. To this we argue that changing the weights and number of bins would only significantly alter the relationship in zone 3 as zone 1 is dominated by clay till and there is relatively good separation of the different lithologies in zone 2. For zone 3, we think it is better to make sure that meltwater sand does not occur for very low resistivities.**

**Since only a small portion of the cells are affected by our subjective weighting it will only have a limited impact on the TI-established lithology-resistivity relationship. We have adjusted our wording to be more precise about what is meant in the text.**

Table 4: I don't understand what the upper row signifies. I feel like I understand the text general discussion that supposedly addresses the table, but I don't understand what the values mean. The first row needs a better explanation.
**We have tried to rephrase the sentence explaining table four in the main text. We have also changed the wording in the table to hopefully make it more readable.**

On line 660 the authors cite Høyer et al. (2017) in their use of 'unconditional simulation'. In fact the simulations are conditioned by the Tis, but not on any data. However, Høyer et al. don't refer to this unconditional simulation as representing the prior distribution. They used this approach to evaluate how effectively the TI constrained the realizations. While the authors are using TI-conditioned simulation for this purpose as well, it is not correct to suggest that two realizations are able to provide sufficient representation of the whole prior distribution. This is important on Figure 12b, as you suggest it is the prior, but again, only two realizations (i.e., samples).

**Regardless of whether Høyer et al. specifically refers to unconditional realizations as representing the prior distribution or not, the fact remains that it is exactly what they the authors are proposing. We prefer the wording of 'unconditional' as this is more in line with geostatistical literature. To name a few examples:**

**Gravey M, Mariethoz G (2020) QuickSampling v1.0: A robust and simplified pixel-based multiple-point simulation approach. Geosci Model Dev 13:2611–2630. https://doi.org/10.5194/gmd-13-2611-2020**

**Juda P, Renard P, Straubhaar J (2020) A Framework for the Cross-Validation of Categorical Geostatistical Simulations. Earth Sp Sci 7:1–17. https://doi.org/10.1029/2020EA001152**

**Laloy E, Hérault R, Lee J, et al (2017) Inversion using a new low-dimensional representation of complex binary geological media based on a deep neural network. Adv Water Resour 110:387–405. https://doi.org/10.1016/j.advwatres.2017.09.029**

**Remy N, Boucher A, Wu J (2009) Applied Geostatistics with SGeMS, 1st edn. Cambridge University Press**

**Tahmasebi P, Hezarkhani A, Sahimi M (2012) Multiple-point geostatistical modeling based on the cross-correlation functions. Comput Geosci 16:779–797. https://doi.org/10.1007/s10596-012-9287-1**

**The idea from Høyer et al. (2017) is to evaluate whether the realizations that arise from the chosen MPS algorithm (and parameterization) + TI provide the structural input that is intended.**

**It is true that the two realizations are too few to depict the full spatial variability of the prior distribution. In our analysis we therefore used more than these two realizations to assess the prior distribution. However, we only showed two realizations in the manuscript to provide an easy visual insight to our reasoning for the reader. This was not clear in the manuscript. We have corrected this.**

On line 791 the authors state, '…and hard data increase the information content (lowers entropy)…'. I believe this is incorrect. This should be written as, 'and hard data **decreases** the information content (lowers entropy)…' Remember, the equation is for Shannon's information entropy, not thermodynamic entropy. (cf. https://stats.stackexchange.com/questions/101351/entropy-and-information-content). Alternatively, the authors could think of entropy in terms of bits – a la Shannon. It takes more bits to describe more complexity, and fewer bits to describe less complexity. More bits = more information content. Fewer bits = less information content.

**Here we must object. Entropy is a measure of disorder. Both in thermodynamics and information theory. Hard data _do_ increase the information content. It lowers the self-information (the level of surprise) though. In the past these two terms (information and self-information) were unfortunately used interchangeably which might still today provide some confusion. Consider the example provided in the attached link.**

**We can all agree that the maximum entropy for a bit happens when the underlying distribution is split 50/50 for each outcome. When a single bit is realized as either heads or tails the self-information is high because it is impossible for us to predict this event happening. I.e. we are surprised by its outcome. That conversely means that the information content in our underlying distribution is low.**

**Conversely, if we have hard data in a voxel the outcome is the same in each realization. There is then no disorder, low entropy, and low self-information because each bit does not provide any information or surprise. Instead, the underlying distribution is fully informed. Following the point from the link, it is far easier to compress information when the self-information is low. In the case with hard data, we basically can compress the information down to a single value (heads or tails). But this is because the underlying distribution is well-informed.**

**We refer the reviewer to this recent article of Hansen (2021) (https://link.springer.com/article/10.1007/s11004-020-09876-z) for a more in-depth description of entropy and information content of geostatistical models. We have also added this reference in the main text for the readers to find.**

On lines 808-809 the authors state, 'This imply that simulation artifacts are not reoccurring in simulations…' I would strike this sentence. It's not phrased properly and is incorrect; analysis of the mode, as a statistic, doesn't prove or disprove either the frequency of artifacts or a bias. If the authors want to keep a reference to these figures, the text should be fixed so it doesn't erroneously say, 'no overall bias is found'. Maybe something like, 'The algorithmic biases that create these artifacts are generating a small number of artifacts that are not significantly affecting the posterior histograms.' However, this point was already made when the authors looked at the histograms earlier in the paper, so this sentence is not necessary or constructive. The statement, 'This is clearly an unwanted side-effect of the soft data conversion…' seems like an inference

based on assumptions rather than an observation based on the algorithm. If that is the case, this statement is unsubstantiated and should be deleted. Consider a hypothetical example: any additions to the distribution would have no effect on the mode as long as they occurred less frequently than the modal value. These additions could be due to a systematic error (i.e., bias).

**This is a valid point. We have deleted the sentence accordingly. Regarding the second point raised about the unwanted side-effect, we disagree. We observe no bias as such. We just observe that some of the converted soft data are placed as sole voxels in the mode modal. That means that even though these conditioning data provide some of the first hard data to condition on, there is no preference in the simulation to simulate oxic conditions around these. This is just an observation. We have removed the word "unwanted" from the sentence to remove the negative loaded interpretation on our behalf.**

On lines 815-832, I think this passage is mistaken. Importantly, the redox scale only has three categories. The posterior distribution showed almost no occurrence of the middle, reducing, category. Don't forget the postglacial seds were assigned a highly reducing value. And even in other textures, there was a non-negligible proportion of reduced occurrences. These all probably contribute to generate a higher entropy at land surface. The prior and TI for redox were strongly biased towards reduced conditions at a certain point in the subsurface. This is reflected in the low entropy. I don't see this as counter intuitive.

**Fair point. Although we would argue that it at first may seem counter-intuitive knowing that the entropy of aligns well with decreasing information sources with depth. We have deleted the 'counter-intuitive' and added a line saying that "The high entropy at the surface is likely also aided by the soft data."**

On line 959, the authors state, 'Although the small size of the TI may pose a problem for reproducing the intended variability…' They have been a bit inconsistent with how they refer to the Tis in terms of their level of detail, the intended goals with respect to the level of detail the authors are trying to represent, and with the recommendations on how much detail to put into a TI. These decisions do involve a bit of professional judgement, more clarity is needed about the goals for the TIs, what the authors tried to do with the TIs, and how they are accommodating the limitations that the choices caused.

**We have now added a line in the end of subsection 7.5 Training images and geological elements addressing our recommendations for TI creation in the future.**

On line 1016 the authors state, '…allows the quantification of uncertainties in the input data…' No it does not do this. It is not correct to use statistical models that were created by a data set to evaluate the uncertainties of that same data set. There isn't that much real data to begin with, so there really isn't enough data to prove any distributional assumptions or models. The manuscript details how the uncertainty in the geologic sediment and redox condition distributions have been quantitatively estimated.

**We agree. It allows a representation of the quantified uncertainties of the input data. It cannot quantify the uncertainty of the input directly. We have changed the wording accordingly.**

Line Number: Comment
8: 'nitrogen (N) losses'…you are only dealing with one loss pathway, leaching. should use 'nitrogen (N) leaching' rather than 'losses'.
**Corrected**

13-15: 'and 2) information of lithology and redox conditions…' this list is presented incorrectly. it should be: '2) lithologies from borehole observations, 3) redox conditions from colors reported in borehole observations, and 4) chemistry analyses from water samples.'
**Corrected**

23: '...and conditional data'...should be: '...and conditioning data...' See Straubhaar, Renard, Mariethoz, 2016.
**Corrected**

26: maybe 'consistent' rather than 'coherent'?
**Changed to "consistent"**

30: 'which may be fundamental to better understanding of the retention of the subsurface..." might read better as: which may lead to a better understanding of nitrogen (N) fate in the subsurface..."
**Corrected**

34: maybe 'loss' instead of 'escape'...leaching is not ideal here because your citations also include runoff Losses
**Corrected**

50: questions: Do you mean nitrate concentrations, instead of conditions? If 'conditions' is correct, I'm not sure what that means. Can you clarify. Assuming concentrations is correct, shouldn't it also be 'leaching concentrations'?

**This should be concentrations. 'Conditions' was revised to concentrations.**

56: 'contaminants: 1)' should be something like, "contaminants. Modeling approaches have included: 1)..."
**Corrected**

66: I would suggest a better wording than, '...space thus requires..." might be, "...space would benefit from more-detailed..." Mostly, this seems to imply that your 25x25x2 is sufficient to capture all detailed heterogeneities in redox conditions, which isn't accurate (it's probably also not what you mean...it just seems implied in the current wording).
**Corrected to proposal**

80: 'is attributed to uncertainties such as' should be 'contains uncertainties from sources that include'
**Corrected to proposal**

84-87: "In fact, subjective biases are accepted as one of the weak points of cognitive geological modeling..." The authors cite Bond 2015 and Wycisk et al. 2007 for this statement. The authors have misrepresented both papers. Bond does not make this statemente and recognizes the importance of geologic knowledge. Wycisk et al. never use the words subjective or bias and state that knowledge of the system is required to make any modeling successful. See comments above about the authors' use of subjectivity, bias, and uncertainty concepts.
**We have rewritten this part according to the suggestions of the reviewer to not misquote the references.**

105: the geostatistical simulation doesn't "quantify uncertainty in...data". It does allow you to quantitatively estimate uncertainty in the unsampled locations, but not the sampled ones.
**This has been rewritten to be more precise.**

106: possible realizations of the subsurface...not a quantitative measurement of uncertainty?
**We have rewritten this sentence to be more precise.**

235-236: "The benefit of using the two types of data is that they provide independent measurements of redox conditions." No that is not correct. The colors clearly reflect long-term redox conditions which are

consistent with the pore water chemistry. So they aren't independent. See comments above.

**See our answer in the beginning**

324-325: '…based on geological event chronology.' should it be '…based on sediment heterogeneity and geological event chronology.' ??

**Corrected**

406-411: You haven't presented data that lets you make any inferences about flow in the groundwater system. See comments above.
**This is addressed in our earlier answer**

458: '5.4 Conditional data' should be '5.4 Conditioning data' There are conditional simulations and conditional realizations that are conditioned to the data. This means the data become condition**ing** data.
**Changed**

533-575. I like the revisions to this portion of the manuscript. See comments above.
**Thank you**

586+ I'm a little concerned about the term, 'soft probability'. I realize Caers and Mariethoz may use it, but qualifying the word 'probabilities' with 'soft' seems inaccurate or at least unclear. 'Soft data probabilities' is more intuitive and seems more correct. I notice you use 'soft data probability' on page 10. Please check on the usage of this term and consider either using 'probabilities' or 'soft data probabilities'. I leave it to your discretion.

**We agree that this was unclear. We have changed to 'soft data probabilities' throughout.**

587: 'resistivity grid in Figure 7' probably makes more sense as, 'resistivity grid from Figure 7'
**Corrected**

614-616: While the mode shows layers in the buried valley, which might be bad, these same voxels show a lot of entropy, which implies a wide distribution over these cells. Which is good. You might make this point in the text. I think it's helpful analysis discussion for readers.
**Agreed, we have now incorporated this point into our argumentation.**

623: Table 4. I don't understand the upper row. I understand from the text, what general topic it's addressing, and I think it is probably helpful. However, I don't understand what the values mean. The first row needs a better explanation.
**Addressed earlier.**

633: 'In direct sampling, the nodes in the simulation grid…' You never discuss this term (direct sampling) nor contrast it with alternative MPS simulation modes. Maybe back up when you introduce MPS, put in a couple sentences about direct sampling and snesim and that your version uses direct sampling. Just for context here.

**We stated in subsection 4.1 MPS modelling (second paragraph) that:**

**"In the current study we use direct sampling (Mariethoz et al., 2010) as implemented in the software package DeeSse (Straubhaar, 2019)."**

**Is this not sufficient? We are not sure that it will benefit the average reader to discuss differences between e.g. direct sampling and SNESIM, ENESIM or other types of MPS. We also stated our main reason for working with direct sampling:**

**"The main reason is its ability to utilize a bivariate training image that allows for joint simulation of geology and redox. "**

**We intend to keep it this way**

635: can you add a citation or citations that you used to generate this list of parameters?
**We have added a reference to the original paper from Mariethoz et al. 2010.**

660: Høyer et al. (2017) do not refer to this unconditional simulation as the prior distribution. See comments Above
**This is true, we have argued to keep the current formulation. See our response earlier.**

667: 'only occur near the surface in accordance with the TIs, but more infrequent than portrayed.' …with some corrections… 'only occur near the surface in accordance with the TI, much more infrequently than portrayed '
**Changed**

686: '…incorporation of conditional data…' should be, '…incorporation of conditioning data…'
**Corrected**

709: 'conditioned data' should be 'conditioning data'
**Corrected**

718: '…the soft data is particularly dominating the realizations…' could be '…the soft data is the dominant constraint on the realizations…'
**Changed to suggestion**

726: '…This coherency explains…' could be '…This consistency explains…'
**Changed to suggestion**

740: question: why don't you more completely filter these oxic artifacts? even in post-simulation processing?
**Because we do not want to focus on these artifacts specifically. We acknowledge their presence; discuss why they are there (due to inconsistencies between prior and conditioning data) and suggest how to handle such artifacts. By post-processing and filtering the simulation, one alters the posterior such that it does not represent the combination of the original input. This is a factor that needs consideration as well. Post-processing should be viewed as more of a necessary evil than the default for removing artifacts. This warrants a deeper analysis of how inconsistencies can be lowered in the first place, and hence alleviating the need for post-processing.**

**Furthermore, we argue that for further hydrological flow modeling these artifacts may be negligible. That does not alleviate the problem, but instead suggest that for a greater modeling workflow other issues needs to be prioritized.**

778-779: '…which together represent the full posterior model.' I think it would be more correct to say something like, '…which together are used to represent the posterior model.'
**Changed to suggestion**

787: '…indicating that other categories are almost equally probable…' probably should be something like, '…indicating that these voxels have a near uniform distribution among several different categories.'
**Changed to suggestion**

791: '…and hard data increase the information content (lowers entropy)…' should be, 'and hard data decreases the information content (lowers entropy)…' See comments above.
**We disagree and have kept the original wording. See our previous answer.**

808-809: I would strike this sentence. See comments above.
**We have done so. See earlier answer.**

815-832: I think you're mistaken here. See comments above.
**The section is slightly altered. See earlier answer.**

836: 'examples of mapping redox conditions with multiple-point geostatistical simulation…' I don't think the use of mapping is appropriate here. It's the wrong audience for this use. A better wording would be, 'examples of simulating redox conditions with sediment-texture distributions using multiple-point geostatistical methods…'
**Changed to suggestion**

848: 'The spatial variability of the TIs is well represented in the prior realizations…" Not really. You only generated two realizations, which is insufficient to estimate the prior distribution, and the prior distribution that you list is not a good estimate of the redox TI - the green zone (reducing) is not well simulated at all.
**We generated more than two realizations but showed only two in the manuscript. We agree that we overemphasized how well the spatial variability of the TI is represented in the prior distribution. We have removed the word 'well' form the sentence in question.**

848: 'conditional data'…'conditioning data'
**Corrected**

854, 857: 'conditional data' …'conditioning data'
**Corrected**

847-865: Nice job on this section. It's a good contribution.

**Thank you**

868: 'The random path have a tendency to underestimate soft data…' ? You later suggest this is why the
877: 'randomly converted a fraction of the soft data…' can you specify the fraction? You are defining a novel method and this would probably be helpful to people.

**The fraction is shown in table 4. Perhaps this was unclear before but should hopefully be clearer now.**

939-961: 7.3 'The inclusion of geological mapping experts in the creation of ThIs introduces modeling subjectivity.' See comments above.
**Yes, the inclusion of geological mapping experts introduces subjectivity modeling. This is not necessarily bad or good per default. We don't think there is any inconsistency in argumentation in this regard. We further proceed to argue that subjectivity is important to bring forward expert knowledge. We mention that others are working on data-driven approaches for generating TIs, but our reasoning for doing a manual interpreted TI is precisely in line with the reasoning of Curtis 2012 and Tarantola 2005.**

**In summary we do not try to avoid subjectivity in our study, but we highlight that it is present. We have tried to rephrase this section to emphasize the need for subjectivity in designing the training images**

959: Although the small size of the TI may pose a problem for reproducing the intended variability…" earlier, you stated that you wanted a generalized model that didn't fully capture all of the possible geometries. can't then make poor fit a criteria.

**We don't state that we want a generalized model. We explicitly say that we intend to make training images that are sufficiently large to capture the general patterns in each geological element. We do recognize that the small size might pose a problem when running the MPS algorithm as fewer matching configurations are possible. This is a trade-off between considering the amount of time it takes to construct a training image, the data availability, independence of information, how well the features can be reproduced (algorithm parameters) and the computation time. We recognize that there is a future study for investigating this trade-off, which is beyond the scope of the current study.**

1000 "…would ensure independence of information." See comments above.

**See our answer in the beginning**

1003: conditional >> conditioning data

**Corrected**

1007: computationally feasible… 80 seconds per realization is trivial. Could have done many more than two realizations to estimate the prior.

**We have also done so but chose to only show two realizations in manuscript as in our above-mentioned answers.**

1016 7.7 …allows the quantification of uncertainties in the input data…" No it does not. See comments above.

**We agree. It allows a representation of the quantified uncertainties of the input data. It cannot quantify the uncertainty of the input directly. We have changed the wording accordingly.**